# Learning Across the Gap: Hybrid Multi-armed Bandits with Heterogeneous Offline and Online Data

**Qijia He**
Southern University of Science and Technology
`heqj2021@mail.sustech.edu.cn`

**Minghan Wang**
Southern University of Science and Technology
`12532257@mail.sustech.edu.cn`

**Xutong Liu**
University of Washington
`xutongl@uw.edu`

**Zhiyong Wang**
The University of Edinburgh
`zhiyongwangwzy@gmail.com`

**Fang Kong**[*]
Southern University of Science and Technology
`kongf@sustech.edu.cn`

## Abstract

The multi-armed bandit (MAB) is a fundamental online decision-making framework that has been extensively studied over the past two decades. To mitigate the high cost and slow convergence of purely online learning, modern MAB approaches have explored *hybrid* paradigms that leverage offline data to warm-start online learning. However, existing approaches face a significant limitation by assuming that the offline and online data are homogeneous—they share the same feedback structure and are drawn from the same underlying distribution. This assumption is often violated in practice, where offline data often originate from diverse sources and evolving environments, resulting in feedback heterogeneity and distributional shifts. In this work, we tackle the challenge of learning across this offline-online gap by developing a general hybrid bandit framework that incorporates heterogeneous offline data to improve online performance. We study two hybrid settings: (1) using reward-based offline data to accelerate online learning in preference-based bandits (i.e., dueling bandits), and (2) using preference-based offline data to improve online standard MAB algorithms. For both settings, we design novel algorithms and derive tight regret bounds that match or improve upon existing benchmarks despite heterogeneity. Empirical evaluations on both synthetic and real-world datasets further show the superior performance of our proposed methods over baseline algorithms.

## 1 Introduction

The multi-armed bandit (MAB) problem is a fundamental online sequential decision-making framework that has been extensively studied over the past two decades (Lattimore and Szepesvári, 2020). In the standard MAB setting, a learning agent chooses one action (i.e., pulls one arm) at each round and receives a stochastic reward (absolute value) as feedback. The goal of the agent is to maximize the cumulative reward over time by balancing exploration and exploitation. This framework has been widely applied in domains such as online advertising, recommendation systems, and information retrieval (Bouneffouf and Rish, 2019).

---

[*]Corresponding author.

39th Conference on Neural Information Processing Systems (NeurIPS 2025).

To deal with the limitations of cold start problem (Bubeck et al., 2009) in online learning and the bias in absolute feedback (Wirth and Fürnkranz, 2013), two extensions of MAB have been proposed: The first is the warm-start bandits (Li et al., 2010), where the agent can access historical offline data before online deployment. The second is the dueling bandits framework (Yue et al., 2012), where feedback is provided as a stochastic preference between two selected arms rather than stochastic rewards.

However, existing warm-start bandit studies-whether under stochastic or dueling settings-typically assume that offline data and online feedback share the same data structure (Oetomo et al., 2023; Cheung and Lyu, 2024; Zhang et al., 2019) (e.g., both provide either stochastic rewards or preference comparisons). But in real world scenarios, two types of data usually coexist. For example: Recommendation systems may combine offline explicit ratings with online pairwise comparison feedback (Zhang et al., 2020), and reinforcement learning in Robotics frequently use offline preference-ranked trajectories with online reward signals from environmental interactions (Brown et al., 2019). Furthermore, existing methods often overlook the potential distributional shift between offline and online data (Bu et al., 2020; Hao et al., 2023). This raises a natural but fundamental research question:

*How can we effectively leverage heterogeneous offline data with potential bias to accelerate online bandit learning?*

To answer the above questions, we made the following key contributions:

**Problem Formulation**: This work investigates the MAB problem under both absolute and relative feedback settings, considering two transition scenarios: from relative to absolute feedback, and from absolute to relative feedback. The framework captures the offline-to-online transition, where offline data can be biased or distributionally shifted. Addressing this framework could significantly enhance the flexibility in the choice of offline data. To the best of our knowledge, this is the first work to propose and analyze such a hybrid MAB setting combining offline-online heterogeneous feedback.

**Algorithm Design**: The central challenge lies in effectively integrating these heterogeneous types of data while mitigating bias inherent in the offline dataset. To tackle this, we propose a novel hybrid estimator that leverages absolute feedback to enhance the learning efficiency of preference-based models, along with a valid bias bound that quantifies the extent of distributional shift. Under the two settings, we propose HybUCB-AR, which integrates offline absolute data with online dueling bandits, and HybElimUCB-RA, which combines offline preference data with online stochastic bandits. While the hybrid estimator enables effective combination of offline data, the pessimistic bias bound allows selective utilization of offline data, ensuring performance aligns with classical MAB algorithms when offline data is uninformative.

**Theoretical Analysis**: We provide a rigorous and comprehensive regret analysis for both algorithms, including instance dependent regret upper bound, instance independent regret upper bound, instance dependent regret lower bound and instance independent regret lower bound. To be specific, for HybUCB-AR, and HybElimUCB-RA, we derive a instance-dependent regret upper bound of

$$O\left(\sum_{i \leq j} \frac{(\Delta_i + \Delta_j) \log T}{\max\{\Delta_i^2, \Delta_j^2\}} - \text{Saving}\right), \quad \text{and} \quad O\left(\sum_i \frac{\log T}{\Delta_i} - \text{Saving}\right)$$

respectively, where the "Saving" term quantifies the benefit derived from offline data. We provided a tighter and precise analysis compared to Cheung and Lyu (2024), with a new discovered "intermediate phase" where the existence of "Saving" is both determined by offline sample size and distribution shift. A key insight in HybUCB-AR is that, by maximizing informative pairs within the confidence set, we improve the RUCB bound of Zoghi et al. (2014) by replacing the denominator $\min\{\Delta_i^2, \Delta_j^2\}$ with $\max\{\Delta_i^2, \Delta_j^2\}$.

**Empirical Validation**: Finally, we conduct extensive experiments on both synthetic and real-world datasets. HybUCB-AR outperforms RUCB by 15% to 40%. Both algorithms outperform state-of-the-art online baselines when leveraging offline heterogeneous data, demonstrating their effectiveness.

## 2 Related Work

Our work is closely related to research on multi-armed bandits, dueling bandits, and bandits with offline data. Below, we provide a brief overview of the relevant literature. A detailed comparison with closely related works is presented in Appendix B.

**Multi-armed Bandits and Dueling Bandits** The concept of Multi-armed Bandits (MAB) was first introduced in Robbins (1952), and has since been extensively studied (Bubeck et al., 2009; Karnin et al., 2013; Slivkins et al., 2019; Lattimore and Szepesvári, 2020). Building on this foundation, dueling bandits were proposed as a variant of MAB to model settings where only relative feedback between pairs of actions is available (Yue and Joachims, 2011; Yue et al., 2012). This framework has gained increasing attention due to the prevalence of preference-based feedback in practical applications such as recommendation systems and reinforcement learning from human feedback (RLHF) (Zhao et al., 2016; Christiano et al., 2017; Wirth et al., 2017). Early works on dueling bandits focused on fundamental algorithms and theoretical guarantees (Urvoy et al., 2013; Zoghi et al., 2015; Dudík et al., 2015; Komiyama et al., 2016; Chen and Frazier, 2017). More recent research has expanded the framework to various settings such as acceleration in multi-user settings Wang et al. (2025b), robustness to corrupted feedback (Agarwal et al., 2021), best-of-both-worlds guarantees (Saha and Gaillard, 2022), adversarial environments (Saha et al., 2021), and delayed or biased feedback (Yi et al., 2024). Despite extensive literature on multi-armed and dueling bandits, few studies have explored hybrid settings where agents leverage both absolute and relative feedback simultaneously. The closest work is by Wang et al. (2025a), who proposed a framework that allows the agent to access both types of feedback in each round. However, their setting differs from ours, as it does not consider either offline data or potential bias.

**Bandits with Offline Data** Multi-armed bandits (MAB) with offline data have been studied in several prior works. Early approaches, such as those by Shivaswamy and Joachims (2012) and Banerjee et al. (2022), primarily assumed that the offline and online environments share the same distributions, and assume homogeneity between offline and online data—either both are stochastic or relative (Bu et al., 2020; Hao et al., 2023; Gur and Momeni, 2022). Only recently have researchers started addressing the challenge of biased offline data (Zhang et al., 2019; Agnihotri et al., 2024; Cheung and Lyu, 2024; Qu et al., 2024). Other studies, such as Sentenac et al. (2025), examine the tradeoff between online exploration budget and offline sample size. But works of heterogeneous offline-online feedback remains understudied, with only recent works making preliminary progress. Xia et al. (2024) investigated hybrid data for linear bandits, albeit without theoretical guarantees, while Agnihotri et al. (2024) adopted a Bayesian posterior approach for the same setting. However, under the MAB framework, none of the previous works simultaneously address both heterogeneous offline data and potential distributional bias—a critical gap our work bridges for the first time.

## 3 Preliminaries

In this paper, we propose a hybrid multi-armed bandit (MAB) framework designed to enhance the efficiency of online learning by incorporating offline heterogeneous data with potential bias. The problem is formulated as a $K$-armed bandit model, where $K \in \mathbb{Z}_+$ is the number of arms, and the arm set $\mathcal{A}$ is $\{1, 2, ..., K\}$.

Initially, we focus on a scenario in which the agent explores its preferences using relative feedback by selecting pairs of arms during the online interaction process, while the offline data provides absolute feedback. Another scenario involving heterogeneous feedback will be provided later.

The learning agent sequentially interacts with the environment for $T \in \mathbb{Z}_+$ time steps. At each step $t \in [T]$, the agent selects a pair of arms $(A_1(t), A_2(t))$, and receives a pairwise comparison feedback. For simplicity, if the selected pair is $(a_i, a_j)$, the agent receives the feedback $Y_{i,j}(t)$ with 1 denoting $a_i$ wins $a_j$ and 0 vice versa. For the preference model, we follow the widely adopted Bradley-Terry (BT) model (Yue et al., 2012; Sun et al., 2024; Zhu et al., 2023; Dong et al., 2024) to characterize the winning probability. Specifically, let $\mu_i^{\text{on}}$ denote the expected reward of arm $i$ during the online

process. The winning probability of arm $a_i$ over $a_j$ is

$$p_{i,j}^{\mathrm{on}} = \sigma(\mu_i^{\mathrm{on}} - \mu_j^{\mathrm{on}}) = \frac{\exp\left(\mu_i^{\mathrm{on}}\right)}{\exp\left(\mu_i^{\mathrm{on}}\right) + \exp\left(\mu_j^{\mathrm{on}}\right)}, \tag{1}$$

where $\sigma(\cdot)$ denotes the logistic sigmoid function. The comparison result $Y_{i,j}(t)$ can be regarded as a Bernoulli random variable with expectation $p_{i,j}^{\mathrm{on}}$. Following prior works (Zoghi et al., 2014; Chen and Frazier, 2017; Saha and Gaillard, 2022), we assume the existence of a Condorcet winner - an arm that is preferred over all other arms. Without loss of generality, we assume this to be arm $a_1$, which satisfies $p_{1,j}^{\mathrm{on}} > 1/2$ for all $j \neq 1$.

The agent typically possesses a substantial amount of offline data, such as historical user click data in a recommendation system (Cheung and Lyu, 2024; Bu et al., 2020; Agnihotri et al., 2024). However, the format of these data may not be entirely consistent with the online tasks. For instance, offline data may exist in the form of absolute click feedback, but the online task aims to identify the relative preference of the users. Besides, the changes in user preferences may lead to differences in expected rewards between offline and online environments. To characterize this phenomenon, we consider that the offline data set contains heterogeneous feedback from the online task. Specifically, for each arm $i \in [K]$, let $N_i \in \mathbb{N}$ denote the number of offline observations associated with arm $i$. The offline data set $\mathcal{D} = \{(i, X_{i,k}) : i \in [K], k \in [N_i]\}$ consists of absolute feedback from the arms, where $X_{i,k}$ is assumed to be 1-subgaussian with expectation $\mu_i^{\mathrm{off}}$. Further define $\{p_{i,j}^{\mathrm{off}}\}_{i,j\in[K]}$ as the preference value derived from $\{\mu_i^{\mathrm{off}}\}_{i\in[K]}$ under the BT model.

The objective of the agent is to maximize its cumulative reward over a given horizon $T$ by effectively utilizing the heterogeneous feedback available in the offline dataset. Equivalently, this objective can be framed as minimizing the cumulative regret:

$$\mathrm{Reg}(T) = \mathbb{E}\left[\sum_{t=1}^{T}\left(\frac{\Delta_{A_1(t)} + \Delta_{A_2(t)}}{2}\right)\right],$$

where the expectation is taken over the randomness of the agent's policy, and $\Delta_i := p_{1,i}^{\mathrm{on}} - \frac{1}{2}$ denotes the sub-optimality gap of arm $i \in [K]$.

### 3.1 Alternative Setting: Stochastic Bandits with Offline Relative feedback

In some applications such as recommendation system (Silva et al., 2022), robotics (Nemlekar et al., 2023; Brown et al., 2019) and RLHF (Das et al., 2024), the learning agent is tasked with making decisions or providing recommendations and observes absolute feedback, such as whether the user accepts or rejects a given option. However, the offline dataset often consists of relative feedback because, during the "hot start" phase when a new user first arrives, many platforms gather information through relative preferences rather than absolute ratings. In this scenario, users are typically asked to express their preferences between pairs of options, such as choosing between two different alternatives. So we further consider the learning task where the online agent is faced with stochastic bandits but with access to an offline data set consisting of relative (pairwise) feedback.

In this setting, the online interaction process follows the standard stochastic bandit formulation. At each time step $t$, the agent selects an arm $A(t)$ and receives a stochastic reward $X_{A(t)}(t)$, which is assumed to be 1-subgaussian with mean $\mu_{A(t)}^{\mathrm{on}}$.

The agent additionally has access to an offline dataset $\mathcal{D} = \{(i, j, Y_{i,j,k}) : i, j \in [K], k \in N_{i,j}\}$, where $N_{i,j}$ denotes the number of recorded comparisons between arms $a_i$ and $a_j$, and $Y_{i,j,k}$ represents the $k$-th comparison outcome. Following prior works on relative feedback (Yue et al., 2012; Ji et al., 2023), we assume the comparisons are generated according to the BT model. That is, $Y_{i,j,k}$ is a Bernoulli random variable with mean $p_{i,j}^{\mathrm{off}} = \sigma(\mu_i^{\mathrm{off}} - \mu_j^{\mathrm{off}})$.

Without loss of generality, we assume arm $a_1$ is the optimal arm in the online setting, i.e., $\mu_1^{\mathrm{on}} > \mu_i^{\mathrm{on}}$ for all $i \in [K]$. To ensure consistency in our definitions throughout the paper, we define the sub-optimality gap for each arm $i$ as $\Delta_i = p_{1i} = \sigma(\mu_1 - \mu_i) - \frac{1}{2}$. The agent aims to minimize the cumulative regret over the online horizon by leveraging information from the offline dataset.

Specifically, the regret is defined as:

$$\text{Reg}(T) = \mathbb{E}\left[\sum_{t=1}^{T} \Delta_{A(t)}\right],$$

where the randomness is taken over the stochastic rewards of the arms.

For notational convenience, throughout the remainder of the paper, we continue to use the 'off' superscript to indicate quantities derived from offline data, and omit the 'on' superscript where no ambiguity arises.

# 4 Dueling Bandits with Offline Absolute Feedback

In this section, we present HybUCB-AR for the dueling bandits with an offline data set consisting of absolute feedback.

## 4.1 Algorithm

Constructing a reliable estimator for the utility of each arm is essential for effective learning in the hybrid bandit problem with heterogeneous feedback considered in this work. Since dueling feedback provides only relative information among arms, directly estimating the absolute utility of individual arms based on relative data remains a challenging task. To leverage the offline dataset containing absolute feedback, we transform its data format into a unified relative feedback type. For data collected during online interactions, it is natural to estimate the relative preference of arm $a_i$ over arm $a_j$ as $\hat{p}_{i,j} = \sum_{k=1}^{T_{i,j}} Y_{i,j,k}/T_{i,j}$. For the offline dataset, we apply the Bradley-Terry (BT) model to construct an estimation of $p_{i,j}$ as $\hat{p}_{i,j}^{\text{off}} = \sigma(\hat{\mu}_i^{\text{off}} - \hat{\mu}_j^{\text{off}})$, where $\hat{\mu}_i^{\text{off}} = \sum_{k=1}^{N_i} X_{i,k}/N_i$ is the empirical mean of the offline data set.

The key challenge is to integrate these two heterogeneous sources of feedback into a single estimator that maximizes information utility while maintaining unbiasedness. To this end, we construct the *Minimum Variance Unbiased Estimator (MVUE)* for $\hat{p}_{i,j}$ by optimally combining empirical observations from both sources. Specifically, since $\hat{p}_{i,j}$ is $1/\sqrt{4T_{i,j}}$-subgaussian with mean $p_{i,j}$, $\hat{p}_{i,j}^{\text{off}}$ is $\sqrt{N_i + N_j}/\sqrt{4(N_i N_j)}$-subguassian with mean $p_{i,j}^{\text{off}}$, and $\hat{p}_{i,j}, \hat{p}_{i,j}^{\text{off}}$ are independent of each other, we construct $\hat{p}_{i,j}^{\text{hyb}}$ as

$$\hat{p}_{i,j}^{\text{hyb}} = \frac{T_{i,j}}{T_{i,j} + \frac{N_i N_j}{N_i + N_j}}\hat{p}_{i,j} + \frac{\frac{N_i N_j}{N_i + N_j}}{T_{i,j} + \frac{N_i N_j}{N_i + N_j}}\hat{p}_{i,j}^{\text{off}}, \tag{2}$$

which is $1/\sqrt{4(T_{i,j} + N_i N_j/(N_i + N_j))}$ subgaussian with mean $p_{i,j}^{\text{hyb}}$. The proof of the sub-Gaussianity is deferred to Appendix C.1.

Due to the distributional mismatch in rewards between the offline data set and online setting, directly applying such a hybrid estimator may lead to the offline data bias adversely affecting the online learning process. To mitigate this issue, the reward estimator for each arm should explicitly account for the potential bias present in the offline data. Inspired by Cheung and Lyu (2024), we introduce a hyper-parameter $V_{i,j}$ to characterize the difference in the offline and online reward distributions. Specifically, $V_{i,j}$ plays the role of an upper bound for $p_{i,j}^{\text{off}} - p_{i,j}$, where

$$V_{i,j} \geq \left|p_{i,j}^{\text{off}} - p_{i,j}\right|, \quad \text{for each arm pair } (a_i, a_j) \in \mathcal{A} \times \mathcal{A}.$$

The quantity $V_{i,j}$ serves as an upper bound on the amount of distributional shift from offline to online. In the extreme case, setting $V_{i,j} \geq 1$ corresponds to having no additional knowledge about the shift. In contrast, $V_{i,j} < 1$ imply nontrivial prior knowledge regarding the difference $|p_{i,j}^{\text{off}} - p_{i,j}|$, a tighter bound leads to better utility of the offline dataset.

**Remark 1.** *In practice, directly computing $V_{i,j}$ is often challenging, as the probabilities $p_{i,j}$, $p_{i,k}$, and $p_{j,k}$ are typically dependent—especially when the number of arms $K$ is large. When $V_{i,j}$ is unavailable for all $i, j \in [K]$, we instead require that $V_i \geq |\mu_i^{off} - \mu_i|$, $\forall i \in [K]$, and define the pairwise term as $V_{i,j} = \sigma(V_i + V_j)$. The justification for this construction is provided in Appendix C.2.*

The construction of HybUCB-AR is based on the upper confidence bound (UCB) estimators. Specifically, we can construct two UCBs for each arm pair $(a_i, a_j)$ based on the pure online data and the hybrid offline and online data as below:

$$\text{UCB}(a_i, a_j) = \hat{p}_{i,j} + 2\sqrt{\frac{\log(1/\delta_t)}{2T_{i,j}}}, \tag{3}$$

$$\text{UCB}^{\text{hyb}}(a_i, a_j) = \hat{p}_{i,j}^{\text{hyb}} + \sqrt{\frac{\log(1/\delta_t)}{2(T_{i,j} + \frac{N_i N_j}{N_i + N_j})}} + \frac{\frac{N_i N_j}{N_i + N_j}}{T_{i,j} + \frac{N_i N_j}{N_i + N_j}} V_{i,j}. \tag{4}$$

We will later show in Appendix C.3 that both UCBs represent upper bound estimates on the real online reward $p_{i,j}$, i.e., $p_{i,j} \leq \min\{\text{UCB}(a_i, a_j), \text{UCB}^{\text{hyb}}(a_i, a_j)\}$ with high probability. However, they reflect different levels of reliance on the offline data. When the offline data is highly informative and the variance term $V_{i,j}$ is small, the hybrid UCB provides a tighter upper bound. Conversely, when the offline data is less informative, the online-only UCB offers a more reliable upper bound estimate.

---

**Algorithm 1** HybUCB-AR (Hybrid UCB: Offline Absolute to Online Relative)

---

**Require:** Arm set $\mathcal{A}$, offline dataset $\mathcal{D} = \{(i, X_{i,k}) : i \in [K], k \in [N_i]\}$, hyperparameter $\delta_t$ and estimated bias $V_{i,j}$ for all $(a_i, a_j)$ pairs.
    **Initialization:**
1: **for all** $a_i, a_j \in \mathcal{A}$ **do**
2:     $T_{i,j} = 0$.                            ▷ Number of comparisons between $a_i$ and $a_j$
3:     $\text{UCB}(a_i, a_j) = +\infty$.
4:     $\text{UCB}^{\text{hyb}}(a_i, a_j) = \sigma(\hat{\mu}_i^{\text{off}} - \hat{\mu}_j^{\text{off}}) + \sqrt{\frac{\log(1/\delta_t)}{2(N_i N_j/(N_i + N_j))}} + V_{i,j}$.
5: **end for**
6: **for** $t = 1$ to $T$ **do**
7:     Construct candidate sets:

$$\mathcal{C}_t^{\text{on}} = \left\{a_i \in \mathcal{A} \mid \text{UCB}(a_i, a_j) \geq \frac{1}{2}, \ \forall a_j \in \mathcal{A}\right\}, \tag{5}$$

$$\mathcal{C}_t^{\text{hyb}} = \left\{a_i \in \mathcal{A} \mid \text{UCB}^{\text{hyb}}(a_i, a_j) \geq \frac{1}{2}, \ \forall a_j \in \mathcal{A}\right\}. \tag{6}$$

8:     $\mathcal{C}_t = \mathcal{C}_t^{\text{on}} \cap \mathcal{C}_t^{\text{hyb}}$.
9:     **if** $C_t \neq \emptyset$ **then**
10:         $(A_1(t), A_2(t)) = \arg\max_{a_i, a_j \in \mathcal{C}_t} \min\{\text{UCB}(a_i, a_j), \text{UCB}^{\text{hyb}}(a_i, a_j)\}$.
11:     **else**
12:         $(A_1(t), A_2(t)) = \arg\max_{a_i, a_j \in \mathcal{A}} \min\{\text{UCB}(a_i, a_j), \text{UCB}^{\text{hyb}}(a_i, a_j)\}$.
13:     **end if**
14:     Select the arm pair $(A_1(t), A_2(t))$.
15:     Update the selection times $T_{A_1(t), A_2(t)} += 1, T_{A_2(t), A_1(t)} += 1$.
16:     Record the observation $Y_{i,j,k}$ and $Y_{j,i,k}$ for $i = A_1(t), j = A_2(t), k = T_{A_1(t), A_2(t)}$.
17:     For all pairs $i, j \in [K]$, update UCB, $\text{UCB}^{\text{hyb}}$ according to equation (3), (4).
18: **end for**

---

In general, the core idea of HybUCB-AR is to construct a candidate set of optimal arms and to explore the arms within this set that have the highest UCBs. To fully leverage the advantages of offline data while mitigating the impact of bias, we employ both a hybrid UCB and a purely online UCB for constructing the candidate set and selecting arms. Specifically, the candidate set includes all arms that could potentially be optimal. Therefore, once an arm is defeated by another—i.e., its corresponding hybrid UCB or pure online UCB falls below 1/2—it is removed from the candidate set (Line 7-8). After constructing the candidate set, the agent would select the arm pair with the highest $\min\{\text{UCB}(a_i, a_j), \text{UCB}^{\text{hyb}}(a_i, a_j)\}$ (Line 9-12).

## 4.2 Theoretical Results

In this section, we provide the regret upper bound for HybUCB-AR and the corresponding lower bound for the problem.

**Theorem 1.** *Choosing $\delta_t = \frac{1}{2K(K+1)t^2}$, the regret of HybUCB-AR satisfies:*
*(a) Instance-dependent bound:*

$$O\left(\sum_{i \leq j} \frac{\Delta_i + \Delta_j}{2} \cdot \left[\frac{\log T}{\max\{\Delta_i^2, \Delta_j^2\}} - \underbrace{\frac{N_i N_j}{N_i + N_j} \cdot \frac{\max\{\max\{\Delta_i, \Delta_j\} - 4\omega_{i,j},\, 0\}}{\max\{\Delta_i, \Delta_j\}}}_{\text{Saving}(a_i, a_j)}\right]\right),$$

*where $\omega_{i,j} := V_{i,j} + p_{i,j}^{off} - p_{i,j}$ with $V_{i,j} \geq |p_{i,j}^{off} - p_{i,j}|$.*

*(b) Instance-independent bounds:*

$$O\left(\min\left\{\sqrt{K^2 T \log T} - \sum_{i \leq j} \text{Saving}(a_i, a_j),\, \left(\sqrt{\frac{\log T}{\tau_*}} + \max_{i,j} V_{i,j}\right) \cdot T\right\}\right),$$

*where $\tau_*$ is the optimal solution to the following linear program:*

$$\max_{\tau,\, t_{i,j}} \quad \tau \tag{7}$$
$$subject\ to \quad \tau \leq \frac{N_i N_j}{N_i + N_j} + t_{i,j}, \quad \forall i, j \in [K],$$
$$\sum_{i \leq j} t_{i,j} = T,$$
$$\tau \geq 0, \quad t_{i,j} \geq 0.$$

**Remark 2.** *In Theorem 1, we have provided two versions of instance-independent bound. The first is an extension of instance-dependent bounds, while the second combines the saving term with the term proportional to $T$. According to the linear programming formulation in (7), the effect of offline data is maximized when the offline data are uniformly distributed and decreases as the offline data become highly skewed, especially when $T$ is relatively small. This achieves a similar result to Sentenac et al. (2025).*

**Remark 3** (Saving)**.** *The complete proof is deferred to Appendix C.5. From the instance dependent case, it is obvious that the effectiveness of offline data is summarized in the "Saving" term. First, note that the Saving is always non-negative. When the condition $4\omega_{i,j} \geq \max\{\Delta_i, \Delta_j\}$ holds for all $i, j \in \mathcal{A}$, the total Saving become zero, and the result degenerates to the pure online regret upper bound. In contrast, when $\max\{\Delta_i, \Delta_j\} > 4\omega_{i,j}$, the offline data becomes beneficial. Specifically, this implies either **(i) the offline and online distributions are sufficiently close** such that the bias is small, or **(ii)** $p_{i,j}^{off} < p_{i,j}$, so the **direction of the distributional shift is correctly estimated by** $V_{i,j}$. This aligns with our intuition that small distribution bias leads to greater utility of offline data, and the bias created by the distributional shift could be neutralized if the shift direction is well-predicted. Furthermore, under the second condition (that is, $\max\{\Delta_i, \Delta_j\} > 4\omega_{i,j,}$), the Saving is proportional to the harmonic mean of the offline sample sizes, namely $N_i N_j / (N_i + N_j)$. This value is maximized when the offline data for $N_i$ and $N_j$ is uniformly distributed, which is consistent with our discussion of the instance-independent bound in Remark 2. A more precise and comprehensive analysis of the Saving, including the "intermediate phase", is deferred to Appendix C.6.*

**Corollary 1.** *If $V_{i,j} = 0$ and the offline data are uniformly distributed such that $N_i = n$ for all $i \in [K]$, then the regret upper bound simplifies to:*

$$O\left(\sum_{i \leq j} \frac{\Delta_i + \Delta_j}{2} \left\lceil \max\left\{\frac{\log T}{\max\{\Delta_i^2, \Delta_j^2\}} - \frac{n}{2}, 0\right\}\right\rceil\right), \quad and \quad O\left(\sqrt{\frac{2K(K+1)T^2 \log T}{nK(K+1) + 4T}}\right)$$

*respectively. Under this setting, the contribution of offline data to regret reduction is captured by the terms $-\frac{n}{2}$ and $nK(K+1)$, respectively. As the amount of offline data becomes sufficiently large, the regret upper bound approaches a constant.*

**Remark 4** (Comparisons with homogeneous offline data). *Under this setting, a natural question arises:* **Compared to homogeneous offline data (absolute feedback), is heterogeneous offline data (preference feedback) less informative or harder to leverage?** *For HybUCB-AR, if we replace the offline stochastic data with preference feedback, the resulting hybrid estimator $\hat{p}_{i,j}^{hyb}$ would depend on a confidence radius of $\sqrt{1/2 \cdot \log(1/\delta_t)/(T_{i,j} + N_{i,j})}$, rather than $\sqrt{1/2 \cdot \log(1/\delta_t)/(T_{i,j} + N_i N_j/(N_i + N_j))}$ as in equation (4). This suggests that preference data can yield a tighter bound for a given pair $(i,j)$ under the same amount of data. However, this result critically relies on the Bradley–Terry model and the 1-subGaussian assumption. As shown in Appendix (C.1), transforming stochastic data into relative data via the BT model enlarges the estimation variance, and such variance misalignment directly impacts the utility of heterogeneous data. If these assumptions are altered, the subGaussian properties of the estimator $\hat{p}_{i,j}^{off}$ under stochastic feedback may also change, potentially leading to different concentration behaviors or confidence bounds. Despite this, as the number of arms $K$ grows, stochastic data will eventually provide more information, since each stochastic sample $X_i$ can be jointly utilized to estimate pairwise preferences across all arms, whereas preference feedback only provides information about the specific pair $(i,j)$ being compared.*

Finally, we establish the regret lower bound of the proposed algorithm, whose analysis relies on the definition of Cp-consistency, which is a generalization to the consistent policy in stochastic bandits (Lai and Robbins, 1985).

**Definition 1.** *For $C > 0$, $p \in (0,1)$ and a collection $\mathcal{I}$ of instances, an Algorithm is said to be Cp-consistent on $\mathcal{I}$, if for all $I \in \mathcal{I}$, it holds that $\mathrm{Reg}(T) \leq CT^p$.*

**Corollary 2.** *Define $\mathcal{I}_V$ be a collection of instance satisfies $V_i \geq |\mu_i^{off} - \mu_i|$ for all $i \in [K]$. By Theorem 1 and Definition 1, HybUCB-AR is Cp-consistent on $\mathcal{I}_V$.*

**Theorem 2** (Regret Lower Bounds). *(a) Instance-dependent bound: Let $V$ be an arbitrary bias bound, and let $\Delta \in (0, \frac{1}{2})$ be the gap between the optimal arm and all suboptimal arms. For all $\mathcal{I} \in \mathcal{I}_V$. the regret of HybUCB-AR satisfy the following lower bounds:*

$$\Omega \left( \frac{K}{4\Delta} \left( (1-p)\ln T + \ln \frac{\Delta}{64C} \right) - \frac{1}{4\Delta} \sum_{i \in [K]} \frac{\max\{2\Delta - \omega_i, 0\}^2}{2} N_i \right),$$

*where $C, p$ are constants from Definition 1, and $\omega_i = V_i + \mu^{off} - \mu_i$ with $V_i \geq |\mu_i^{off} - \mu_i|$.*

*(b) Instance-independent bound: Let $V_{\max}$ be fixed and arbitrary, and let $V_i = V_{\max}$ for all $i \in [K]$, then there exists a Gaussian instance $\mathcal{I} \in \mathcal{I}_V$ satisfying the following regret lower bound:*

$$\Omega \left( \min \left\{ \sqrt{KT}, \left( \sqrt{\frac{1}{\tau'_*}} + V_{\max} \right) \cdot T \right\} \right), \tag{8}$$

*where $\tau'_*$ is the optimal solution to the following Linear Programming problem:*

$$
\begin{aligned}
\max_{\tau', t_i} \quad & \tau' \\
\text{subject to} \quad & \tau' \leq t_i + N_i, \quad \forall i \in [K], \\
& \sum_{i \in [K]} t_i = 2T, \quad \tau' \geq 0, \, t_i \geq 0.
\end{aligned}
$$

The proof of Theorem 2 is deferred to Appendix D.1 and D.2. Both the gap-dependent and gap-independent lower bounds *closely match* the regret upper bound provided in Theorem 1. In particular, for the special case where $N_i = n$ and $\Delta_i = \Delta$, we show in Appendix D.1 that the upper and lower bounds differ by at most a factor of $K$. Notice that the term $\sqrt{KT}$ in Equation (8) does not contain "Saving", as under our construction the worst case could be that the Saving is zero.

## 5 Stochastic Bandits with Offline Relative Feedback

In contrast to the previous setting, we now consider a classical MAB problem augmented with offline preference data. Due to the switch between offline and online data, we redefine $\hat{p}_{i,j}^{off} =$

$\sum_{k=1}^{N_{i,j}} Y_{i,j,k}/N_{i,j}$, $\hat{p}_{i,j} = \sigma(\hat{\mu}_i - \hat{\mu}_j)$, where $\hat{\mu}_i = \sum_{i=1}^{T_i} X_{i,k}/T_i$, $\forall i \in [K]$. and $\hat{p}_{i,j}^{\text{hyb}} = \alpha \hat{p}_{i,j}^{\text{off}} + (1-\alpha)\hat{p}_{i,j}$, where $\alpha = N_{i,j}/(\frac{T_i T_j}{T_i + T_j} + N_{i,j})$. The upper confidence bound becomes:

$$\text{UCB}(a_i, a_j) = \hat{p}_{i,j} + 2\sqrt{\frac{\log(1/\delta_t)}{2} \cdot \frac{T_i + T_j}{T_i T_j}}, \tag{9}$$

$$\text{UCB}^{\text{hyb}}(a_i, a_j) = \hat{p}_{i,j}^{\text{hyb}} + \sqrt{\frac{\log(1/\delta_t)}{2\left(\frac{T_i T_j}{T_i + T_j} + N_{i,j}\right)}} + \frac{N_{i,j}}{\frac{T_i T_j}{T_i + T_j} + N_{i,j}} V_{i,j}. \tag{10}$$

---

**Algorithm 2** HybElimUCB-RA (Hybrid Elimination UCB: Offline Relative Online Absolute)

---

**Require:** an arm set $\mathcal{A}$, offline dataset $\mathcal{D} = \{(i, j, Y_{i,j,k}), i, j \in [K], k \in [N_{i,j}]\}$, hyperparameter $\delta_t$ and estimated bias $V_{i,j}$ for all $(a_i, a_j)$ pairs.
   **Initialization:** Let $\mathcal{C} = \mathcal{A}$, $T_i = 0, \forall i \in [K]$, and for all $i, j \in [K]$, let $\text{UCB}(a_i, a_j) = +\infty$, $\text{UCB}^{\text{hyb}}(a_i, a_j) = \hat{p}_{i,j}^{\text{off}} + \sqrt{\frac{\log 1/\delta_t}{N_{i,j}}}$.
1: **for** $t = 1, \cdots, T$ **do**
2:    Select Action $A(t) = \arg\min_{a_i \in \mathcal{C}} T_i(t)$.
3:    Update selection times $T_{A(t)} = T_{A(t)} + 1$.
4:    Record observation $X_{i,k}$ for $i = A(t)$, $k = T_{A(t)}$.
5:    For all pairs $i, j \in [K]$, update UCB, $\text{UCB}^{\text{hyb}}$ according to equation (9), (10).
6:    $\mathcal{C} = \mathcal{C} \setminus \{a_i \in \mathcal{C}, \exists a_j \in \mathcal{C}\setminus\{a_i\} \text{ s.t. } \min\{\text{UCB}(a_i, a_j), \text{UCB}^{\text{hyb}}(a_i, a_j)\} < \frac{1}{2}\}$.
7: **end for**

---

In Section 4, we used absolute offline data to estimate $p_{i,j}$ based on BT model. However, using preference data to estimate $\mu_i$ is hard, as it provides less information. in HybElimUCB-RA, we approach this problem by continuing using preference based estimation model, with an elimination approach. In each round, we select the arm that has been played the least, and eliminate suboptimal arm $a_i$ if $\exists a_j \in \mathcal{A}$ s.t. $\min\{\text{UCB}(a_i, a_j)\text{UCB}^{\text{hyb}}(a_i, a_j)\} < 1/2$. Under the worst case where the hybrid terms provides no additional information, it's performance matches the vanilla elimination UCB algorithm.

**Remark 5.** *One might question how to align with vanilla UCB when offline data cannot provide additional information. As mentioned, estimating $\mu$ from preference data is mathematically challenging due to the absence of explicit reward signals. An alternative approach is to maintain pure online UCB and hybrid UCB separately, which is similar to the ElimFusion Algorithm in Wang et al. (2025a). A detailed implementation of this approach is provided in Appendix F.*

**Theorem 3.** *Let $\delta_t = \frac{1}{2K(K+1)T^2}$. The instance dependent regret upper bound of HybElimUCB-RA satisfies:*

$$O\left(\sum_{i \in [K]} \frac{\log(T)}{\Delta_i} - \underbrace{2N_{i,1}\max\{\Delta_i - 2\omega_{i,1}, 0\}}_{\text{Saving}(a_i)}\right),$$

*where $\omega_{i,1} := V_{i,1} + p_{i,1}^{\text{off}} - p_{i,1}$ with $V_{i,1} \geq |p_{i,1}^{\text{off}} - p_{i,1}|$.*

**Remark 6.** *Due to space limits, we defer the instance-independent regret bound and lower bound to Appendix E.2 and E.3. The proof of Theorem 3 is provided in the Appendix E.1. The regret bound follows a form similar to that in Section 4: a larger offline dataset and smaller bias in the offline data lead to greater regret reduction. Notably, one key insight is that, under the relative to stochastic setting, uniformly distributed offline data no longer leads to good Saving. As shown in Theorem 3, the regret reduction for each suboptimal arm $a_i \in [K]\setminus\{1\}$ depends critically on the comparisons with the optimal arm $a_1$. Specifically, offline data concentrated on comparisons with $a_1$ leads to higher efficiency in the online phase.*

# 6 Experiments

We evaluate HybUCB-AR and HybElimUCB-RA on synthetic and real-world datasets to assess their performance in hybrid feedback bandit settings. Figure 1 presents four subplots comparing their performance against classic bandit baselines. Specifically, for online preference setting, we compared our result with IF (Yue et al., 2012) and RUCB (Zoghi et al., 2014), while for online stochastic setting we set the baseline as ETC, vanilla UCB (Lattimore and Szepesvári, 2020) and Thompson Samping (Agrawal and Goyal, 2013). Without offline data, HybUCB-AR, by maximizing the informative pair in the candidate set, outperforms RUCB by 15 to 40%. Both algorithms surpass baseline when heterogeneous offline data is included. These results highlight the robust performance of our algorithms across diverse scenarios. Details and supplemental experiments, including algorithm's performance across varying K, sensitivity to parameters, and efficacy in real-world datasets, are provided in Appendix G.

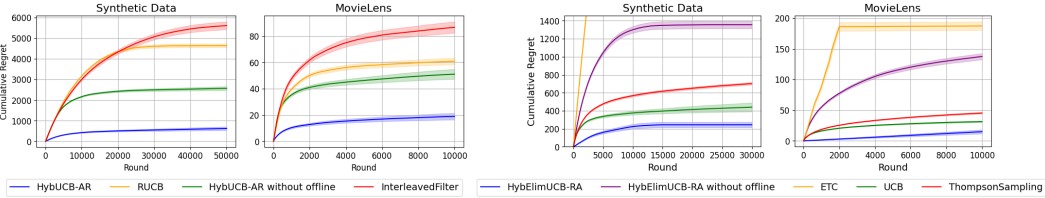

Figure 1: Cumulative regret comparison of HybUCB-AR (left) and HybElimUCB-RA (right) against baselines. Left two subplots: HybUCB-AR on synthetic data and the MovieLens dataset; right two subplots: HybElimUCB-RA on the same datasets.

# 7 Conclusion

This paper investigates multi-armed bandits with heterogeneous offline data, we propose a novel hybrid estimator that effectively leverages hybrid data to enhance online learning, with comprehensive theoretical guarantees. This significantly expands the practical applicability of bandit frameworks by offering greater flexibility in incorporating offline data. Future directions include generalization to linear bandits and adversarial environments, as well as more flexible settings where heterogeneity exists within offline or online data.

## Acknowledgement

The corresponding author Fang Kong is supported by the Guangdong Basic and Applied Basic Research Foundation 2025A1515011412.

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

# A  Notations and Models

| Notation | Meaning |
|---|---|
| $K$ | # of arms |
| $a_i$ | arm $i$ |
| $\hat{\mu}_i^{\text{off}}, \hat{\mu}_i$ | the estimated mean reward of $a_i$, offline and online respectively |
| $\mu_i^{\text{off}}, \mu_i$ | the ground truth mean reward of $a_i$, for offline and online respectively |
| $p_{i,j}, p_{i,j}^{\text{off}}$ | ground truth Ber(p) s.t. $a_i$ wins against $a_j$ for offline and online data respectively |
| $\hat{p}_{i,j}$ | the estimated value of $p_{i,j}$ only using online data |
| $\hat{p}_{i,j}^{\text{off}}$ | the estimated value of $p_{i,j}$ using offline data |
| $\hat{p}_{i,j}^{\text{hyb}}$ | the estimated value of $p_{i,j}$ using offline and online data(hybrid) |
| $\Delta_i$ | $p_{1,i} - \frac{1}{2}$, $a_1$ refers to the Condorcet winner |
| $T_{i,j}$ | # of times that played $(a_i, a_j)$ online, |
| | especially, we use $T_{i,j}(t)$ to refer the # of times that play $(a_i, a_j)$ at the start of t |
| $T_i(t)$ | # of times that $a_i$ is selected for online comparison |
| $N_i$ | # of times that played $a_i$ in the offline dataset |
| $X_{i,k}$ | the $k^{th}$ reward of $a_i$ |
| $Y_{i,j,k}$ | the k-th feedbcak of $(a_i, a_j)$ |
| $V_{i,j}$ | a input valid bound for pairwise data, where $V_{i,j} \geq |p_{i,j}^{\text{off}} - p_{i,j}|$ |
| $\mathcal{E}_t$ | good event, $\mathcal{E}_t = \bigcap_{i,j \in [K]} \left( \mathcal{E}_t(p_{i,j}) \cap \mathcal{E}_t^{\text{hyb}}(p_{i,j}) \right)$ |
| $\omega_{i,j}$ | $V_{i,j} + p_{i,j}^{\text{off}} - p_{i,j}$ |
| UCB, UCB$^{\text{hyb}}$ | the UCB using pure online data and hybrid data |
| $\sigma(\cdot)$ | sigmoid function |

# B  More Detail Comparisons with Existing Works

## B.1  Comparison with Cheung and Lyu (2024)

Cheung and Lyu (2024) proposed Policy MIN-UCB algorithm, which is the first to deal with bias dataset under MAB setting with input valid bias bound $V(a)$, our work extends it to heterogeneous data setting and refine its regret upper bound. According to Lemma 4.8 in Cheung and Lyu (2024),

$$N_t(a) \geq 32 \cdot \frac{\log(4Kt^4)}{\Delta(a)^2} - T_S(a) \cdot \max\left\{ 1 - \frac{\omega(a)}{\Delta(a)}, 0 \right\}^2,$$

where $N_t(a)$ is the number of times arm $a$ is selected online, and $T_S(a)$ denotes its offline sample size. Following the derivation in Appendix B.2 of Cheung and Lyu (2024), and adapting it to our setting, we simplify the above inequality into the following form, as used in our Lemma 5 and 6:

$$T' \geq 8\frac{\delta'}{\Delta'} - N' \cdot \max\left\{ 1 - \frac{\omega'}{\Delta'}, 0 \right\}^2.$$

One difference in our setting lies in the preference-based feedback model, which assumes a $1/2 \cdot \sqrt{1/T_{i,j}}$ sub-Gaussian distribution. This changes the confidence radius from $\sqrt{2\log(1/\delta_t)/T_i}$ to $\sqrt{\log(1/\delta_t)/(2T_{i,j})}$, leading to a different constant factor in the inequality.

However, the theoretical analysis in Cheung and Lyu (2024) is limited in that it only examines the Saving when there is no online data at all (Case 1a in Appendix B.2, where it directly scales online data $N_t(a) = 0$). To achieve a regret bound with a "Saving" term, their analysis doubles the coefficient of the standard $\log T/\Delta$ term of UCB. Specifically, traditional online UCB yields a regret of the form $C \log T/\Delta$ where $C$ is a constant term, but the regret upper bound for the hybrid setting in Cheung and Lyu (2024) is $2C \log T/\Delta - \text{Saving}$. This means that when the Saving term is small, their regret can even exceed that of pure online UCB. Our analysis resolves this issue from a different theoretical analysis perspective, and restores the regret bound back to $C \log T/\Delta - \text{Saving}$. Furthermore, the Case 1a in Cheung and Lyu (2024) is actually a subcase of our more general analysis, as the following chain of inequalities shows:

$$N'\left(1 - \frac{\omega'}{\Delta'}\right)^2 \geq \frac{8\delta'}{\Delta'^2}$$
$$\geq \frac{4\delta'}{\Delta'^2} \cdot \frac{2\Delta' - \omega'}{\Delta'}$$
$$= \frac{4\delta'(2\Delta' - \omega')}{\Delta'^3},$$

where the first inequality corresponds to the condition in Case 1a in Cheung and Lyu (2024), the second holds due to the assumption $\Delta' \geq \omega'$, and the resulting expression aligns with the condition stated in our Lemma 5.

## B.2 Comparison with Wang et al. (2025a)

Wang et al. (2025a) proposed an online hybrid MAB setting where the agent could observes both rewards and pairwise preferences per round. To be specific, Wang et al. (2025a) proposed two algorithms to address the hybrid feedback multi-armed bandit (MAB) problem: *Elimination Fusion* (ElIMFUSION) and *Decomposition Fusion* (DECOFUSION). ElIMFUSION eliminates an arm $a_j$ if either $\mathrm{UCB}(a_j) \leq \mathrm{LCB}(a_i)$ or $\mathrm{UCB}(a_j, a_i) < \frac{1}{2}$ holds, this method closely related to our HybElimUCB-RA. DECOFUSION employs a more complex and cooperative policy between dueling and reward-based feedback, randomly using one of the feedback to explore the arm space and the other to exploit. Under the regret definition $R_T = \alpha R_T^{\mathrm{R}} + (1 - \alpha) R_T^{\mathrm{D}}$, where D refers "dueling" and R refers "reward", and $\alpha$ is an input parameter. The regret upper bound of ElIMFUSION and DECOFUSION algorithms are:

$$\mathbb{E}[R_T] \leqslant O\left(\sum_{k \neq 1} \frac{(\alpha \Delta_k^{\mathrm{(R)}} + (1 - \alpha)\Delta_k^{\mathrm{(D)}}) \log T}{\max\{(\Delta_k^{\mathrm{(R)}})^2, (\Delta_k^{\mathrm{(D)}})^2/K\}}\right)$$

and

$$\mathbb{E}[R_T] \leqslant O\left(\sum_{k \neq 1} \frac{\log T}{\max\{\Delta_k^{\mathrm{(R)}}/\alpha, \Delta_k^{\mathrm{(D)}}/(1 - \alpha)\}}\right)$$

respectively.

In our setting, the fusion arises from the integration of offline and online data. Within each source (offline or online), the data is homogeneous; therefore, in the online learning phase, only one type of feedback is available. To effectively leverage both sources, we adopt the Bradley–Terry model and construct a unified estimator for $p_{i,j}$, capturing the pairwise preference probability. In contrast, the method in Wang et al. (2025a) treats the estimation of each arm or arm pair independently, the joint utility of heterogeneous data comes by constructing candidate set together.

## B.3 Comparison with Zoghi et al. (2014)

Zoghi et al. (2014) proposed the RUCB algorithm to address preference-based feedback without relying on the strong stochastic transitivity and stochastic triangle inequality assumptions required by earlier works such as Yue et al. (2012). However, RUCB does not fully exploit the information in its candidate set, as it selects the second arm $c$ (corresponding to $A_2(t)$ in HybUCB-AR) uniformly at random. As a result, it can only guarantee a regret bound of

$$O\left(\sum_{i \leq j} \frac{(\Delta_i + \Delta_j) \log T}{\min\{\Delta_i^2, \Delta_j^2\}}\right).$$

In this work, we improve upon this by maximizing informative pair in the candidate set:

$$(A_1(t), A_2(t)) = \underset{a_i, a_j \in \mathcal{C}_t \times \mathcal{C}_t}{\arg\max} \mathrm{UCB}(a_i, a_j).$$

This change allows us to better utilize candidate set information, leading to a tighter regret bound of

$$O\left(\sum_{i \leq j} \frac{(\Delta_i + \Delta_j) \log T}{\max\{\Delta_i^2, \Delta_j^2\}}\right).$$

## B.4 Comparison with Qu et al. (2024)

Qu et al. (2024) proposed a hybrid transfer reinforcement learning (HTRL) algorithm, HySRL, which selectively uses historical data exhibiting shifted dynamics to reduce the sample complexity of online reinforcement learning.

Similar to the findings in Cheung and Lyu (2024), it proved in general HTRL, when no addition knowledge or restriction in applied to historical dataset, the sample complexity could not be improved.

To address the distributional shift between offline and online samples, HySRL introduces the concept of $\beta$-separable shift, which classifies offline data as either distributionally identical to or different from the online environment. For offline samples deemed identical, the algorithm directly incorporates them into online estimation. In contrast, samples identified as distributionally different are entirely excluded during the online learning phase. Under this framework, the sample complexity of HySRL is:

$$\widetilde{O}\left(\min\left(\frac{H^3 SA}{\varepsilon^2}, \frac{H^3|\mathcal{B}|}{\varepsilon^2} + \frac{H^2 S^2 A}{(\sigma\beta)^2}\right)\right),$$

where $\alpha, \beta, \epsilon$ are their input parameters, $H$ is the length of episode, $S, A$ refer to state and action space respectively.

In Qu et al. (2024), the distribution shift region is detected online, and under the theoretical sample complexity bounds, the shifted region can be correctly identified with high probability. Despite its flexibility, this approach relies on strong assumptions regarding the $\beta$-separability definition and the choice of $\beta$. Moreover, it does not account for the sample size of the historical dataset. In contrast, our method assumes a valid bias bound as input, which eliminates the need for warm-start estimation of the shift region, but requires prior knowledge about the arms or the transition probability functions.

## C   Proof for Theorem 1

### C.1   Sub-Gaussian Properties

**Lemma 1** (sub-Gaussian Properties of Estimators). *Let $\hat{p}_{i,j}$ denote an estimator of the preference probability between arms $i$ and $j$. The sub-Gaussian parameter of $\hat{p}_{i,j}$ depends on the type of data used for estimation:*

1. *For relative preference data, where*

$$\hat{p}_{i,j} = \frac{1}{T_{i,j}} \sum_{k=1}^{T_{i,j}} Y_{i,j,k},$$

   *the estimator is $\frac{1}{2}\sqrt{\frac{1}{T_{i,j}}}$ sub-Gaussian.*

2. *For stochastic utility data, where*

$$\hat{p}_{i,j} = \sigma\left(\frac{1}{N_i}\sum_{k=1}^{N_i} X_{i,k} - \frac{1}{N_j}\sum_{k=1}^{N_j} X_{j,k}\right),$$

   *the estimator is $\frac{1}{2}\sqrt{\frac{N_i + N_j}{N_i N_j}}$ sub-Gaussian.*

3. *For a hybrid estimator combining both types of data:*

$$\hat{p}_{i,j}^{hyb} = \frac{1}{T_{i,j} + \frac{N_i N_j}{N_i + N_j}}\left(\sum_{k=1}^{T_{i,j}} Y_{i,j,k}\right) + \frac{\frac{N_i N_j}{N_i + N_j}}{T_{i,j} + \frac{N_i N_j}{N_i + N_j}}\sigma\left(\frac{1}{N_i}\sum_{k=1}^{N_i} X_{i,k} - \frac{1}{N_j}\sum_{k=1}^{N_j} X_{j,k}\right),$$

   *the sub-Gaussian parameter is $\frac{1}{2}\sqrt{\frac{1}{T_{i,j} + \frac{N_i N_j}{N_i + N_j}}}$.*

*Proof.* We prove each case separately.

1. **Pure Relative Preference Estimator.** Let $Y \sim \text{Bernoulli}(p)$ with values in $[0, 1]$. By Hoeffding's Lemma, for any $\lambda \in \mathbb{R}$,

$$\mathbb{E}\left[\exp\left(\lambda(Y - \mathbb{E}[Y])\right)\right] \leq \exp\left(\frac{\lambda^2(1-0)^2}{8}\right) = \exp\left(\frac{\lambda^2}{8}\right).$$

   This corresponds to a $\frac{1}{2}$ sub-Gaussian variable, since

$$\frac{\lambda^2}{8} = \frac{\lambda^2(1/2)^2}{2}.$$

   As $\hat{p}_{i,j}$ is the average of $T_{i,j}(t)$ independent Bernoulli samples, Lemma 5.4 in Lattimore and Szepesvári (2020) imply that $\hat{p}_{i,j}$ is $\frac{1}{2}\sqrt{\frac{1}{T_{i,j}(t)}}$ sub-Gaussian.

2. **Pure Stochastic Utility Estimator.** Assume that each $X_{i,k}$ and $X_{j,k}$ is 1 sub-Gaussian. Since sub-Gaussianity is preserved under averaging, we have:

$$\frac{1}{N_i}\sum_{k=1}^{N_i} X_{i,k} \sim \frac{1}{\sqrt{N_i}} \text{ sub-Gaussian}, \quad \frac{1}{N_j}\sum_{k=1}^{N_j} X_{j,k} \sim \frac{1}{\sqrt{N_j}} \text{ sub-Gaussian}.$$

Therefore, the difference of these averages is sub-Gaussian with parameter $\sqrt{1/N_i + 1/N_j}$ Since sigmoid function is $1/4$-Lipschitz continuous, applying Lemma 10 yields the final sub-Gaussian parameter:

$$2 \cdot \frac{1}{4} \cdot \sqrt{\frac{1}{N_i} + \frac{1}{N_j}} = \frac{1}{2}\sqrt{\frac{1}{N_i} + \frac{1}{N_j}}.$$

3. **Hybrid Estimator.** Let

$$\alpha := T_{i,j}, \quad \beta := \frac{N_i N_j}{N_i + N_j}, \quad Z := \sigma\left(\frac{1}{N_i}\sum_{k=1}^{N_i} X_{i,k} - \frac{1}{N_j}\sum_{k=1}^{N_j} X_{j,k}\right).$$

The estimator is a convex combination:

$$\hat{p}_{i,j}^{\text{hyb}} = \frac{1}{\alpha + \beta}\left(\sum_{k=1}^{\alpha} Y_{i,j,k} + \beta Z\right).$$

Since $\sum_{k=1}^{\alpha} Y_{i,j,k}$ is $\frac{1}{2}\sqrt{\alpha}$ sub-Gaussian and $Z$ is $\frac{1}{2}\sqrt{\frac{1}{N_i} + \frac{1}{N_j}}$ sub-Gaussian, by Lemma 5.4 Lattimore and Szepesvári (2020), the entire sum has a sub-Gaussian parameter:

$$\sigma = \frac{1}{\alpha + \beta}\sqrt{\alpha^2 \cdot \frac{1}{4\alpha} + \beta^2 \cdot \frac{1}{4}\left(\frac{1}{N_i} + \frac{1}{N_j}\right)} = \frac{1}{2\sqrt{\alpha + \beta}}.$$

Substituting back yields the desired result.

$\square$

## C.2 Justification of Valid Bias Bound

**Lemma 2.** *Given $V_i \geq |\mu_i^{\text{off}} - \mu_i|, \forall i \in [K]$, under the Bradley-Terry model, we could derive the valid bias upper bound for pairwise term as $V_{i,j} \geq |p_{i,j}^{\text{off}} - p_{i,j}|, \forall i, j \in [K]$.*

*Proof.* Since $V_i \geq \mu_i - \mu_i^{\text{off}}$ and $V_j \geq \mu_j^{\text{off}} - \mu_j$, we have $V_i + V_j \geq (\mu_i - \mu_j) - (\mu_i^{\text{off}} - \mu_j^{\text{off}})$, which further derives

$$\sigma(V_i + V_j) \geq \sigma((\mu_i - \mu_j) - (\mu_i^{\text{off}} - \mu_j^{\text{off}})) \geq \sigma(\mu_i - \mu_j) - \sigma(\mu_i^{\text{off}} - \mu_j^{\text{off}}) = p_{i,j} - p_{i,j}^{\text{off}}.$$

The first inequality is by the monotonically increasing property of sigmoid function, and the second inequality is by the property of sigmoid function where $\sigma(x + y) \leq \sigma(x) + \sigma(y), \forall x, y \in \mathbb{R}$.

Similarly, we could derive $\sigma(V_i + V_j) \geq p_{i,j}^{\text{off}} - p_{i,j}$, hence $\sigma(V_i + V_j) \geq |p_{i,j}^{\text{off}} - p_{i,j}|$, completes the proof. $\square$

## C.3 Good Event and High Probability Bound

Define good event $\mathcal{E}_t = \bigcap_{i,j \in [K]}\left(\mathcal{E}_t(p_{i,j}) \cap \mathcal{E}_t^{\text{hyb}}(p_{i,j})\right)$ where

$$\mathcal{E}_t(p_{i,j}) = \left\{p_{i,j} \leq \text{UCB}(a_i, a_j) \leq p_{i,j} + 2\sqrt{\frac{\log(1/\delta_t)}{2T_{i,j}}}\right\} \tag{11}$$

$$\mathcal{E}_t^{\text{hyb}}(p_{i,j}) = \left\{p_{i,j} \leq \text{UCB}^{\text{hyb}}(a_i, a_j) \leq p_{i,j} + 2\sqrt{\frac{\log(1/\delta_t)}{2(T_{i,j}(t) + \frac{N_i N_j}{N_i + N_j})}} + \frac{\frac{N_i N_j}{N_i + N_j}}{T_{i,j}(t) + \frac{N_i N_j}{N_i + N_j}}\omega_{i,j}\right\} \tag{12}$$

with $\omega_{i,j} := V_{i,j} + p_{i,j}^{\text{off}} - p_{i,j}$, we have the following lemma:

**Lemma 3.** *The good event satisfies the following lower bound:* $\Pr(\mathcal{E}_t) \geq 1 - 2K(K + 1)\delta_t$.

*Proof.* This proof largely follows the argument of Appendix B.1 in Cheung and Lyu (2024). We begin by applying the Chernoff bound for sub-Gaussian random variables. Consider the estimator $\hat{p}_{i,j}$ of the true pairwise comparison probability $p_{i,j}$. Since the difference $X = \hat{p}_{i,j} - p_{i,j}$ is sub-Gaussian with parameter $\frac{1}{2}\sqrt{1/T_{i,j}(t)}$ by Lemma 1, we can apply the Chernoff inequality (see Lemma 8) to obtain:

$$\Pr\left(|\hat{p}_{i,j} - p_{i,j}| \geq \sqrt{\frac{\log(1/\delta_t)}{2T_{i,j}(t)}}\right) \leq 2\delta_t.$$

This establishes the condition for the 'bad event' $\mathcal{E}_t^c(p_{i,j})$ by rearranging the inequality above.

Next, we analyze the 'bad event' $\mathcal{E}_t^{\mathrm{hyb},c}(p_{i,j})$, which occurs when either of the following two conditions is violated. The first case corresponds to a violation of the first inequality in Equation (12). For this case, we aim to derive an upper bound on the probability of violation:

$$\Pr\left(p_{i,j} \geq \mathrm{UCB}^{\mathrm{hyb}}(a_i, a_j)\right),$$

where

$$\mathrm{UCB}^{\mathrm{hyb}}(a_i, a_j) = \frac{T_{i,j}(t)\hat{p}_{i,j} + \frac{N_i N_j}{N_i + N_j}\sigma(\hat{X}_i - \hat{X}_j)}{T_{i,j}(t) + \frac{N_i N_j}{N_i + N_j}} + \sqrt{\frac{\log(1/\delta_t)}{2\left(T_{i,j}(t) + \frac{N_i N_j}{N_i + N_j}\right)}} + \frac{\frac{N_i N_j}{N_i + N_j}}{T_{i,j}(t) + \frac{N_i N_j}{N_i + N_j}}V_{i,j}.$$

This probability can be rewritten as:

$$\Pr\left(\frac{T_{i,j}(t)p_{i,j} + \frac{N_i N_j}{N_i + N_j}p_{i,j}^{\mathrm{off}}}{T_{i,j}(t) + \frac{N_i N_j}{N_i + N_j}} + \frac{\frac{N_i N_j}{N_i + N_j}(p_{i,j} - p_{i,j}^{\mathrm{off}})}{T_{i,j}(t) + \frac{N_i N_j}{N_i + N_j}} \geq \right.$$

$$\left. \frac{T_{i,j}(t)\hat{p}_{i,j} + \frac{N_i N_j}{N_i + N_j}\sigma(\hat{X}_i - \hat{X}_j)}{T_{i,j}(t) + \frac{N_i N_j}{N_i + N_j}} + \sqrt{\frac{\log(1/\delta_t)}{2\left(T_{i,j}(t) + \frac{N_i N_j}{N_i + N_j}\right)}} + \frac{\frac{N_i N_j}{N_i + N_j}}{T_{i,j}(t) + \frac{N_i N_j}{N_i + N_j}}V_{i,j}\right)$$

$$\leq \Pr\left(\frac{T_{i,j}(t)p_{i,j} + \frac{N_i N_j}{N_i + N_j}p_{i,j}^{\mathrm{off}}}{T_{i,j}(t) + \frac{N_i N_j}{N_i + N_j}} \geq \frac{T_{i,j}(t)\hat{p}_{i,j} + \frac{N_i N_j}{N_i + N_j}\sigma(\hat{X}_i - \hat{X}_j)}{T_{i,j}(t) + \frac{N_i N_j}{N_i + N_j}} + \sqrt{\frac{\log(1/\delta_t)}{2\left(T_{i,j}(t) + \frac{N_i N_j}{N_i + N_j}\right)}}\right)$$

$$\leq \delta_t,$$

where the penultimate inequality uses the fact that $V_{i,j} \geq |p_{i,j}^{\mathrm{off}} - p_{i,j}|$, and the last inequality comes by applying the Chernoff inequality to the hybrid estimator, which is $\frac{1}{2}\sqrt{\frac{1}{T_{i,j}(t) + \frac{N_i N_j}{N_i + N_j}}}$ sub-Gaussian (by Lemma 1).

For the second case, consider:

$$\Pr\left(\mathrm{UCB}^{\mathrm{hyb}}(a_i, a_j) \geq p_{i,j} + 2\sqrt{\frac{\log(1/\delta_t)}{2\left(T_{i,j}(t) + \frac{N_i N_j}{N_i + N_j}\right)}} + \frac{\frac{N_i N_j}{N_i + N_j}}{T_{i,j}(t) + \frac{N_i N_j}{N_i + N_j}}(V_{i,j} + p_{i,j}^{\mathrm{off}} - p_{i,j})\right).$$

This can be written as:

$$\Pr\left(\frac{T_{i,j}(t)\hat{p}_{i,j} + \frac{N_i N_j}{N_i + N_j}\sigma(\hat{X}_i - \hat{X}_j)}{T_{i,j}(t) + \frac{N_i N_j}{N_i + N_j}} \geq p_{i,j} + \sqrt{\frac{\log(1/\delta_t)}{2\left(T_{i,j}(t) + \frac{N_i N_j}{N_i + N_j}\right)}} + \frac{\frac{N_i N_j}{N_i + N_j}}{T_{i,j}(t) + \frac{N_i N_j}{N_i + N_j}}(p_{i,j}^{\mathrm{off}} - p_{i,j})\right)$$

$$= \Pr\left(\frac{T_{i,j}(t)\hat{p}_{i,j} + \frac{N_i N_j}{N_i + N_j}\sigma(\hat{X}_i - \hat{X}_j)}{T_{i,j}(t) + \frac{N_i N_j}{N_i + N_j}} \geq \frac{T_{i,j}(t)p_{i,j} + \frac{N_i N_j}{N_i + N_j}p_{i,j}^{\mathrm{off}}}{T_{i,j}(t) + \frac{N_i N_j}{N_i + N_j}} + \sqrt{\frac{\log(1/\delta_t)}{2\left(T_{i,j}(t) + \frac{N_i N_j}{N_i + N_j}\right)}}\right)$$

$$\leq \delta_t.$$

The final inequality is by applying the Chernoff inequality.

Since there are $\frac{K(K+1)}{2}$ distinct pairs of arms, and for each pair the probability of $\mathcal{E}_t$ is at most $4\delta_t$, applying this to all arms gives:

$$1 - \Pr(\mathcal{E}_t) = \Pr(\mathcal{E}_t^c)$$

$$\leq \sum_{i=1}^{K} \sum_{i \leq j} \Pr(\mathcal{E}_t^c(p_{i,j}) + \mathcal{E}_t^{\text{hyb},c}(p_{i,j})) \quad \text{(union bound)}$$

$$\leq 2K(K+1)\delta_t \quad \left( \text{since } \Pr(\mathcal{E}_t^c(p_{i,j}) + \mathcal{E}_t^{\text{hyb},c}(p_{i,j})) \leq 4\delta_t \right)$$

and hence,

$$\Pr(\mathcal{E}_t) \geq 1 - 2K(K+1)\delta_t,$$

which concludes the proof. $\qquad\square$

## C.4 Property of Algorithm 1

**Lemma 4.** *(HybUCB-AR's property) Conditioned on $\mathcal{E}_t$, the arm pair $(a_i, a_j)$ will **not** be selected in round $t$ if it satisfies the following Inequality:*

$$\max\{\Delta_i, \Delta_j\} > \min \left\{ 4\sqrt{\frac{\log(1/\delta_t)}{2T_{i,j}(t)}}, 4\sqrt{\frac{\log(1/\delta_t)}{2\left(T_{i,j}(t) + \frac{N_i N_j}{N_i + N_j}\right)}} + 2\frac{\frac{N_i N_j}{N_i + N_j}}{T_{i,j}(t) + \frac{N_i N_j}{N_i + N_j}} \omega_{i,j} \right\}.$$

*Proof.* Conditioned on $\mathcal{E}_t$, Suppose arm pair $(a_i, a_j)$ is selected, there are two possible cases for this selection:

1. $(A_1(t), A_2(t)) = (a_i, a_j)$,

2. $(A_1(t), A_2(t)) = (a_j, a_i)$.

We analyze the first case, while the second case follows symmetrically. By HybUCB-AR's selection criterion (Line-7), when $(a_i, a_j)$ is chosen, it must hold that:

$$1 - p_{i,j} + 2\sqrt{\frac{\log(1/\delta_t)}{2T_{i,j}(t)}} = p_{j,i} + 2\sqrt{\frac{\log(1/\delta_t)}{2T_{i,j}(t)}} \geq \text{UCB}(a_j, a_i) \geq \frac{1}{2}, \tag{13}$$

and

$$1 - p_{j,i} + 2\sqrt{\frac{\log(1/\delta_t)}{2T_{i,j}(t)}} = p_{i,j} + 2\sqrt{\frac{\log(1/\delta_t)}{2T_{i,j}(t)}} \geq \text{UCB}(a_i, a_j) \geq \frac{1}{2}. \tag{14}$$

Equation (13) and (14) implies that

$$p_{i,j} \leq \frac{1}{2} + 2\sqrt{\frac{\log(1/\delta_t)}{2T_{i,j}(t)}} \quad \text{and} \quad p_{j,i} \leq \frac{1}{2} + 2\sqrt{\frac{\log(1/\delta_t)}{2T_{i,j}(t)}}. \tag{15}$$

Given $\Delta_i > 4\sqrt{\frac{\log(1/\delta_t)}{2T_{i,j}(t)}}$, we have the following:

$$\text{UCB}(a_i, a_j) \leq p_{i,j} + 2\sqrt{\frac{\log(1/\delta_t)}{2T_{i,j}(t)}} \leq \frac{1}{2} + 4\sqrt{\frac{\log(1/\delta_t)}{2T_{i,j}(t)}} < p_{1,i} \leq \min\left\{\text{UCB}(a_1, a_i), \text{UCB}^{\text{byb}}(a_1, a_i)\right\}, \tag{16}$$

The first and fourth inequalities in Equation (16) follow from condition $\mathcal{E}_t$, while the second inequality is derived from the first inequality in Equation (15). The third inequality holds by the given condition $\Delta_i > 4\sqrt{\frac{\log(1/\delta_t)}{2T_{i,j}(t)}}$. Recalling that HybUCB-AR's selection criterion (Line 10) satisfies $(a_i, a_j) = \arg\max_{a_i, a_j \in \mathcal{C}_t} \min\{\text{UCB}(a_i, a_j), \text{UCB}^{\text{hyb}}(a_i, a_j)\}$, we observe that (16) leads to a contradiction.

Following the same reasoning, the condition $\Delta_j > 4\sqrt{\frac{\log(1/\delta_t)}{2T_{i,j}(t)}}$ would also lead to an analogous contradiction. This implies that the arm pair $(a_i, a_j)$ cannot be selected when the following holds:

$$\max\{\Delta_i, \Delta_j\} > 4\sqrt{\frac{\log(1/\delta_t)}{2T_{i,j}(t)}}. \tag{17}$$

Since the candidate set is constructed by $\mathcal{C}_t = \mathcal{C}_t^{\text{on}} \cap \mathcal{C}_t^{\text{hyb}}$, applying the same argument to $\mathcal{C}_t^{\text{hyb}}$ yields that $(a_i, a_j)$ will also not be selected if:

$$\max\{\Delta_i, \Delta_j\} > 4\sqrt{\frac{\log(1/\delta_t)}{2\left(T_{i,j} + \frac{N_i N_j}{N_i + N_j}\right)}} + 2\frac{\frac{N_i N_j}{N_i + N_j}}{T_{i,j} + \frac{N_i N_j}{N_i + N_j}} \omega_{i,j}. \tag{18}$$

Combining Equation (17) and (18) leads to the desired result. By symmetry, the analysis for the second case $(A_1(t), A_2(t)) = (a_j, a_i)$ mirrors that of $(a_i, a_j)$ and leads to the same conclusion, as desired. □

## C.5 Proof of Instance Dependent Regret Upper Bound

Before we provide the proof, we present a general lemma concerning the solution to the key inequalities derived in Lemma 4.

**Lemma 5.** *Consider the notation $N'$, $T'$, $\omega'$, $\delta'$, and $\Delta'$, these variables are defined to simplify the structure of the equations while preserving their form, enhancing clarity and generality. Let $T'$ be the smallest integer satisfying*

$$\Delta' > \min\left\{2\sqrt{\frac{\delta'}{T'}}, 2\sqrt{\frac{\delta'}{T'+N'}} + \frac{N'}{T'+N'}\omega'\right\}, \tag{19}$$

*then*

$$T' > \begin{cases} \max\left\{\frac{2\delta' + N'\Delta'(\omega'-\Delta') + \sqrt{4\delta'^2 + 4\delta'N'\omega'\Delta'}}{\Delta'^2}, 0\right\} & \text{if Condition Saving,} \\ \frac{4\delta'}{\Delta'^2} & \text{otherwise,} \end{cases}$$

*where:*

$$\text{Condition Saving} = (2\omega' \leq \Delta') \text{ or } \left(N' \geq \frac{4\delta'(2\omega' - \Delta')}{\Delta'(\omega' - \Delta')^2}\right).$$

*Proof.* As established, Equation (19) represent bounds derived from the HybUCB-AR's framework, corresponding to pure online and pure hybrid data scenarios, respectively. Our proposed algorithm requires that $T'$ satisfies at least one of these conditions. The use of offline data reduces regret when the smallest $T'$ satisfying

$$\Delta' > 2\sqrt{\frac{\delta'}{T'}} \tag{20}$$

is less than that required by

$$\Delta' > 2\sqrt{\frac{\delta'}{T'+N'}} + \frac{N'}{T'+N'}\omega'. \tag{21}$$

For Equation (20), direct manipulation yields:

$$T' > \frac{4\delta'}{\Delta'^2}. \tag{22}$$

For Equation (21), define $\sqrt{T'+N'} = x$, $A = 2\sqrt{\delta'}$, and $B = N'\omega'$. The inequality becomes:

$$\frac{A}{x} + \frac{B}{x^2} < \Delta'.$$

Solving this for $x \geq 0$, we obtain:

$$x > \frac{A + \sqrt{A^2 + 4B\Delta'}}{2\Delta'}$$

$$x^2 > \frac{A^2 + 2B\Delta' + \sqrt{A^4 + 4A^2B\Delta'}}{2\Delta'^2}.$$

This implies:

$$T' > \frac{2\delta' + N'\omega'\Delta' + \sqrt{4\delta'^2 + 4\delta'N'\omega'\Delta'}}{\Delta'^2} - N' = \frac{2\delta' + N'\Delta'(\omega' - \Delta') + \sqrt{4\delta'^2 + 4\delta'N'\omega'\Delta'}}{\Delta'^2}.$$

Since $T' \geq 0$, we have:

$$T' > \max\left\{\frac{2\delta' + N'\Delta'(\omega' - \Delta') + \sqrt{4\delta'^2 + 4\delta'N'\omega'\Delta'}}{\Delta'^2}, 0\right\}. \tag{23}$$

Combining these, the general bound is:

$$T' > \min\left\{\frac{4\delta'}{\Delta'^2}, \max\left\{\frac{2\delta' + N'\Delta'(\omega' - \Delta') + \sqrt{4\delta'^2 + 4\delta'N'\omega'\Delta'}}{\Delta'^2}, 0\right\}\right\}. \tag{24}$$

Next, we identify conditions under which Equation (21) yields a smaller $T'$ than Equation (20). Define the Saving as:

$$\text{Saving} = \frac{4\delta' - \left(2\delta' + N'\Delta'(\omega' - \Delta') + \sqrt{4\delta'^2 + 4\delta' N'\omega'\Delta'}\right)}{\Delta'^2}.$$

From Equation (24), Saving are positive only if $\Delta' > \omega'$, since $\omega' - \Delta' < 0$ is the sole negative contribution, potentially reducing $T'$. This condition also ensures the left-hand side of the subsequent second inequality (Equation (25)) remains non-negative.

We require Saving $\geq 0$:

$$\frac{4\delta' - 2\delta' - N'\Delta'(\omega' - \Delta') - \sqrt{4\delta'^2 + 4\delta' N'\omega'\Delta'}}{\Delta'^2} \geq 0$$

$$2\delta' - N'\Delta'(\omega' - \Delta') \geq \sqrt{4\delta'^2 + 4\delta' N'\omega'\Delta'} \quad (25)$$

$$N'^2\Delta'^2(\omega' - \Delta')^2 - 4\delta' N'\Delta'(\omega' - \Delta') \geq 4\delta' N'\omega'\Delta'$$

$$N'^2\Delta'^2(\omega' - \Delta')^2 \geq 4\delta' N'\Delta'(2\omega' - \Delta').$$

Hence the final inequality, guaranteeing that saving $\geq 0$, holds if either

(i)  $2\omega' \leq \Delta'$;

or (ii)  $2\Delta' > 2\omega' > \Delta'$  and  $N' \geq \dfrac{4\delta'(2\omega' - \Delta')}{\Delta'(\omega' - \Delta')^2}.$

These conditions together validate the case distinction stated in the lemma.     $\square$

*Proof.* (Complete Proof of Instance Dependent Regret Upper Bound in Theorem 1)

By Lemma 3 and Lemma 4, take $\Delta' = \frac{1}{2}\max\{\Delta_i, \Delta_j\}$, $\delta' = \frac{1}{2}\log(1/\delta_t)$, $\omega' = w_{i,j}$, $N' = \frac{N_i N_j}{N_i + N_j}$, $T' = T_{i,j}(t)$, under the good event $\mathcal{E}_t$, the number of times the arm pair $(a_i, a_j)$ is played satisfies:

If $2\max\{\Delta_i, \Delta_j\} > 2\omega_{i,j} > \max\{\Delta_i, \Delta_j\}$ and $\frac{N_i N_j}{N_i + N_j} \geq \frac{2\log(1/\delta_t)(2\omega_{i,j} - \max\{\Delta_i, \Delta_j\})}{\max\{\Delta_i, \Delta_j\}(\omega_{i,j} - \max\{\Delta_i, \Delta_j\})^2}$, then:

$$T_{i,j}(t) \leq \max\left\{\frac{4\log(1/\delta_t) + \frac{N_i N_j}{N_i + N_j}\max\{\Delta_i, \Delta_j\}(2\omega_{i,j} - \max\{\Delta_i, \Delta_j\}) + \sqrt{D}}{\max\{\Delta_i^2, \Delta_j^2\}}, 0\right\}, \quad (26)$$

where $D = \log^2(1/\delta_t) + \log(1/\delta_t)\cdot\frac{N_i N_j}{N_i + N_j}\omega_{i,j}\max\{\Delta_i, \Delta_j\}$,

otherwise:

$$T_{i,j}(t) \leq \frac{8\log(1/\delta_t)}{\max\{\Delta_i^2, \Delta_j^2\}}. \quad (27)$$

Define $g(a_i, a_j)$ and $f(a_i, a_j)$ as follows. Here, $g(a_i, a_j)$ denotes the upper bound of $T_{i,j}(t)$ in the presence of Saving, while $f(a_i, a_j)$ characterizes the condition under which Saving occur. Specifically, Saving exist if $f(a_i, a_j) < 0$.

$$g(a_i, a_j) = \max\left\{\frac{4\log(1/\delta_t)}{\max\{\Delta_i^2, \Delta_j^2\}} + \frac{\frac{N_i N_j}{N_i + N_j}\max\{\Delta_i, \Delta_j\}\cdot(2\omega_{i,j} - \max\{\Delta_i, \Delta_j\}) + \sqrt{D}}{\max\{\Delta_i^2, \Delta_j^2\}}, 0\right\},$$

$$f(a_i, a_j) = \begin{cases} -1 & \text{if } 4\omega_{i,j} \leq \max\{\Delta_i, \Delta_j\}, \\ \frac{4\log(1/\delta_t)(4\omega_{i,j} - \max\{\Delta_i, \Delta_j\})}{\max\{\Delta_i, \Delta_j\}(2\omega_{i,j} - \max\{\Delta_i, \Delta_j\})^2} - N_{i,j} & \text{if } 2\max\{\Delta_i, \Delta_j\} > 4\omega_{i,j} > \max\{\Delta_i, \Delta_j\}, \\ 1 & \text{otherwise.} \end{cases}$$

$$(28)$$

The expected regret is:

$$\text{Reg}(T) = \sum_{t=1}^{T} \mathbb{E}\left[\left(\frac{\Delta_{A_1(t)} + \Delta_{A_2(t)}}{2}\right) \cdot [\mathbf{1}(\mathcal{E}_t) + \mathbf{1}(\mathcal{E}_t^c)]\right]$$

$$\leq \sum_{t=1}^{T} \mathbb{E}\left[\left(\frac{\Delta_{A_1(t)} + \Delta_{A_2(t)}}{2}\right) \cdot \mathbf{1}(\mathcal{E}_t)\right] + \Delta_{\max} \sum_{t=1}^{T} \mathbb{E}\left[\mathbf{1}(\mathcal{E}_t^c)\right]$$

$$= \sum_{i \leq j} \frac{\Delta_i + \Delta_j}{2} \mathbb{E}[T_{i,j}(T) \cdot \mathbf{1}(\mathcal{E}_t)] + \Delta_{\max} \sum_{t=1}^{T} \mathbb{E}\left[\mathbf{1}(\mathcal{E}_t^c)\right],$$

Let $\delta_t = 1/(2K(1+K)t^2)$, under the bad event, the regret is:

$$\Delta_{\max} \sum_{t=1}^{T} \mathbb{E}\left[1(\mathcal{E}_t^c)\right] = \Delta_{\max} \sum_{t=1}^{T} \mathbb{E}\left[1(\mathcal{E}_t^c)\right] = \sum_{t=1}^{T} 2K(1+K)\delta_t \Delta_{\max} = \sum_{t=1}^{T} \Delta_{\max} \frac{1}{t^2} \leq \frac{\pi^2}{6} \Delta_{\max}, \quad (29)$$

under the good event, the regret is:

$$\sum_{i \leq j} \frac{\Delta_i + \Delta_j}{2} \mathbb{E}[T_{i,j}(T) \cdot \mathbf{1}(\mathcal{E}_t)] = \sum_{\substack{i \leq j, \\ f(a_i, a_j) < 0}} g(a_i, a_j) \frac{\Delta_i + \Delta_j}{2} + \sum_{\substack{i \leq j, \\ f(a_i, a_j) \geq 0}} \frac{8(\Delta_i + \Delta_j) \log(\sqrt{2K(1+K)}T)}{\max\{\Delta_i^2, \Delta_j^2\}}.$$
$$(30)$$

Combining Equation (29) and (30), we obtain the desired result. The simplified regret upper bound is obtained by applying the following inequality to $g(a_i, a_j)$ in Equation (30):

$$\sqrt{x^2 + 2ax} < a + x, \quad (31)$$

where $x = \log(1/\delta_t)$, $a = \frac{N_i N_j}{2(N_i + N_j)} \max\{\Delta_i, \Delta_j\}\omega_{i,j}$, we derive:

$$\text{Reg}(T) \leq \frac{\pi^2}{6}\Delta_{\max} + \sum_{i \leq j} \max\left\{\frac{\Delta_i + \Delta_j}{2}\left[\frac{16 \log(\sqrt{2K(K+1)}T)}{\max\{\Delta_i^2, \Delta_j^2\}} - \frac{N_i N_j}{N_i + N_j} \frac{\max\{\max\{\Delta_i, \Delta_j\} - 4\omega_{i,j}, 0\}}{\max\{\Delta_i, \Delta_j\}}\right], 0\right\}.$$
$$(32)$$

$\square$

## C.6   Saving: Further Analysis

**Remark 7.** *To facilitate concise notation, we denote by $T_{ij}(UCB)$ and $T_{ij}(UCB^{hyb})$ the upper bounds on $T_{ij}$ under the UCB and $UCB^{hyb}$ selection rules, respectively. Specifically,*

$$T_{ij}(UCB) = \frac{4\delta'}{\Delta'^2},$$

$$T_{ij}(UCB^{hyb}) = \max\left\{\frac{2\delta' + N'\Delta'(\omega' - \Delta') + \sqrt{4\delta'^2 + 4\delta'N'\omega'\Delta'}}{\Delta'^2}, 0\right\}.$$

*This is the result we derived from Equation (22) and (23) in Lemma 5. In the context of Theorem 1, $T_{ij}(UCB)$ and $T_{ij}(UCB^{hyb})$ becomes Equation (27) and (26) respectively.*

After establishing Theorem 1, we provided a brief analysis of the "Saving" term in its regret bound in Remark 3. Note that Theorem 1 is a simplified version derived by applying Equation (31) to Equations (29) and (30). Below, we present a more precise analysis of the conditions under which "Saving" exists. By Equation (28), we could divide the condition into three different cases:

1. **Significant Saving:** When $\max\{\Delta_i, \Delta_j\} \geq 4\omega_{i,j}$, offline data guarantees regret reduction.

2. **Data-Dependent Saving:** If $4\omega_{i,j} > \max\{\Delta_i, \Delta_j\} > 2\omega_{i,j}$, Saving exist only when the offline data size satisfies:
$$\frac{N_i N_j}{N_i + N_j} \geq \frac{4\log(1/\delta_t)(4\omega_{i,j} - \max\{\Delta_i, \Delta_j\})}{\max\{\Delta_i, \Delta_j\}(2\omega_{i,j} - \max\{\Delta_i, \Delta_j\})^2}.$$

3. **No Saving:** For $\max\{\Delta_i, \Delta_j\} \leq 2\omega_{i,j}$, offline data provides no benefit.

This result aligns with our Saving analysis in Section 3, which shows that Saving depends on the similarity between offline and online data distributions, with the magnitude of Saving influenced by the offline data size. But the detail analysis further shows there exist an intermediate phase: where the existence of Saving is data dependent. Here, we provide deeper insight into the three regimes. When the distributions are sufficiently close $(\max\{\Delta_i, \Delta_j\} \geq 4\omega_{i,j})$, the $T_{ij}(\text{UCB})$ is consistently at most that of $T_{ij}(\text{UCB}^{\text{hyb}})$, ensuring Saving regardless of the offline data size. As $\omega_{i,j}$ increases, indicating greater distributional divergence, the $T_{ij}(\text{UCB})$ grows, potentially exceeding that of $T_{ij}(\text{UCB}^{\text{hyb}})$. In this intermediate regime $(4\omega_{i,j} > \max\{\Delta_i, \Delta_j\} > 2\omega_{i,j})$, Saving is possible only if the offline data size $N_{i,j}$ is large enough to reduce $T_{ij}(\text{UCB}^{\text{hyb}})$, compensating for the increased $\omega_{i,j}$. However, when the distributions are too dissimilar $(\max\{\Delta_i, \Delta_j\} \leq 2\omega_{i,j})$, $T_{ij}(\text{UCB}^{\text{hyb}})$ is dominated by $\omega_{i,j}$, and no amount of offline data can yield Saving. These cases illustrate a spectrum of outcomes driven by distributional similarity and the size of offline data.

In Lemma 5, we established conditions for the existence of Saving, but their magnitude remains unanalyzed. For instance, when $\Delta' > 2\omega'$, Saving are guaranteed regardless of the offline dataset size $N'$. However, if $N'$ is small, the confidence radius may remain large, resulting in negligible Saving. To further quantify the saving term, we analyze a specific setting where inferior arms can be identified without online data, thereby maximizing Saving.

Given the following Lemma 6, this implies no online exploration is needed for HybUCB-AR when the following condition holds:

$$\frac{N_i N_j}{N_i + N_j} \geq \frac{16\log(1/\delta_t)(\max\{\Delta_i, \Delta_j\} - \omega_{i,j})}{\max\{\Delta_i, \Delta_j\}(2\omega_{i,j} - \max\{\Delta_i, \Delta_j\})^2} \quad \text{and} \quad 2\omega_{i,j} < \max\{\Delta_i, \Delta_j\}.$$

**Lemma 6.** *Suppose $\omega' < \Delta'$ and*

$$N' \geq \frac{4\delta'(2\Delta' - \omega')}{\Delta'(\Delta' - \omega')^2}.$$

*Then, under the good event, no online exploration is needed for this suboptimal arm.*

*Proof.* Continuing from Equation (23), we aim to identify the condition under which the following expression becomes non-positive:

$$\frac{2\delta' + N'\Delta'(\omega' - \Delta') + \sqrt{4\delta'^2 + 4\delta'N'\omega'\Delta'}}{\Delta'^2} \leq 0.$$

It is clear that this inequality can only hold when $\omega' < \Delta'$; otherwise, all terms in the numerator would be positive and the entire expression strictly positive.

Under the substitution from Equation (23), we consider the equivalent condition:

$$N'\Delta'(\Delta' - \omega') - 2\delta' \geq \sqrt{4\delta'^2 + 4\delta'N'\omega'\Delta'}.$$

Squaring both sides yields:

$$N'^2\Delta'^2(\Delta' - \omega')^2 - 4\delta'N'\Delta'(\Delta' - \omega') \geq 4\delta'N'\omega'\Delta'.$$

Rearranging terms, we obtain:

$$N' \geq \frac{4\delta'(2\Delta' - \omega')}{\Delta'(\Delta' - \omega')^2},$$

as desired. $\qquad\square$

**Remark 8.** *Specifically, when $\omega' = 0$, the condition simplifies to $N' \geq 8\delta'/\Delta'^2$, which degenerate to the algorithm property of vanilla UCB. In contrast, under the Saving condition in Lemma 5 $(2\Delta' > 2\omega' > \Delta')$, Saving begin when the offline data size $N_i N_j/(N_i + N_j)$ meets a certain threshold, but fully replacing online exploration requires a larger $N_i N_j/(N_i + N_j)$. The difference in $N_i N_j/(N_i + N_j)$ between these two thresholds is given by:*

$$\frac{4\delta'(2\Delta' - \omega')}{\Delta'(\Delta' - \omega')^2} - \frac{4\delta'(2\omega' - \Delta')}{\Delta'(\omega' - \Delta')^2} = \frac{4\delta'(3\Delta' - 3\omega')}{\Delta'(\Delta' - \omega')^2}.$$

## C.7 Proof of Instance Independent Regret Upper Bounds

As shown in Theorem 1, we provide two instance-independent regret upper bounds, and the final bound is obtained by taking the minimum of the two. The details of these bounds are presented in the following two Sections (C.7.1 and C.7.2).

### C.7.1 Analysis 1

*Proof.* We derive the instance independent upper bound through modification of the instance dependent regret upper bound. Let $\Delta$ be a variable between $(0,1)$. For $(a_i, a_j) \in \mathcal{A} \times \mathcal{A}$ such that $\max\{\Delta_i, \Delta_j\} \geq \Delta$, take $\delta_t = \frac{1}{2K(K+1)t^2}$, the core term in the instance regret upper bound (Equation (32), e.g., the leading $\Delta$-dependent term) is bounded by:

$$\frac{4(\Delta_i + \Delta_j)\log(1/\delta_t)}{\max\{\Delta_i^2, \Delta_j^2\}} \leq \frac{8\log(1/\delta_t)}{\max\{\Delta_i, \Delta_j\}} < \frac{8\log(1/\delta_T)}{\Delta} = \frac{16\log(\sqrt{2K(K+1)}T)}{\Delta},$$

where the first inequality comes by taking $\Delta_i + \Delta_j \leq 2\max\{\Delta_i, \Delta_j\}$.

For any arm pair $(a_i, a_j) \in \mathcal{A} \times \mathcal{A}$ satisfying $\max\{\Delta_i, \Delta_j\} < \Delta$, the total regret incurred by these sub-optimal pairs is upper bounded by $T\Delta$. Let $\mathrm{Saving}(a_i, a_j) = \frac{\Delta_i + \Delta_j}{2} \cdot \frac{N_i N_j}{N_i + N_j} \cdot \frac{\max\{\max\{\Delta_i, \Delta_j\} - 4\omega_{i,j}, 0\}}{\max\{\Delta_i, \Delta_j\}}$, and $\mathrm{Saving}'(a_i, a_j) = \min\left\{ \frac{\Delta_i + \Delta_j}{2} \cdot \frac{N_i N_j}{N_i + N_j} \cdot \frac{\max\{\max\{\Delta_i, \Delta_j\} - 4\omega_{i,j}, 0\}}{\max\{\Delta_i, \Delta_j\}}, \frac{8(\Delta_i + \Delta_j)\log(\sqrt{2K(K+1)}T)}{\max\{\Delta_i^2, \Delta_j^2\}} \right\}$ for all $i, j \in [K]$. Combining these results yields:

$$\begin{aligned}
\mathrm{Reg}(T) &\leq \frac{\pi^2}{6}\Delta_{\max} + \sum_{i \leq j} \max\left\{ \frac{16\log(\sqrt{2K(K+1)}T)}{\max\{\Delta_i^2, \Delta_j^2\}} - \mathrm{Saving}(a_i, a_j), 0 \right\} \\
&\leq \frac{\pi^2}{6}\Delta_{\max} + T\Delta + \frac{8K(K+1) \cdot \log(\sqrt{2K(K+1)}T)}{\Delta} - \sum_{i \leq j} \mathrm{Saving}'(a_i, a_j) \\
&\leq \frac{\pi^2}{6}\Delta_{\max} + 4\sqrt{2K(K+1) \cdot \log(\sqrt{2K(K+1)}T)} - \sum_{i \leq j} \mathrm{Saving}'(a_i, a_j),
\end{aligned}$$

where the last inequality follows by taking $\Delta = \sqrt{\frac{8K(K+1) \cdot \log(\sqrt{2K(K+1)}T)}{T}}$, as desired. $\qquad\square$

### C.7.2 Analysis 2

This theoretical analysis extends the framework of Appendix B.3 in Cheung and Lyu (2024), generalizing its regret analysis from the classical multi-armed bandit (MAB) setting to a preference-based model.

**Lemma 7.** *Let $\{T_{i,j}^*\}_{i \leq j \in [K]}$ is the optimal solution to the following maximization problem:*

$$\begin{aligned}
\max_{(T_{i,j})_{i \leq j}} &\sum_{i \leq j} \sum_{t=1}^{T_{i,j}} \sqrt{\frac{1}{t + \frac{N_i N_j}{N_i + N_j}}}, \\
\text{s.t.} &\sum_{i \leq j} T_{i,j} = T, \\
&T_{i,j} \in \mathbb{N}_{\geq 0}, \forall i \leq j \in [K].
\end{aligned}$$

*then it must hold that $T_{i,j}^* \leq \max\{\lceil \tau_* \rceil - \frac{N_i N_j}{N_i + N_j}, 0\}$ for all $a_i, a_j \in \mathcal{A}$.*

*Proof.* We proof this lemma by contradiction. Suppose there's an arm pair $(a_i, a_j)$ such that $T_{i,j}^* \geq \max\{\lceil \tau_* \rceil - \frac{N_i N_j}{N_i + N_j}, 0\} + 1$, then it must hold that there exist at least one arm pair $(a_{i'}, a_{j'})$ such that $T_{i'j'}^* \leq \max\{\lceil \tau_* \rceil - \frac{N_{i'} N_{j'}}{N_{i'} + N_{j'}}, 0\} - 1$, otherwise we have: $T_{i''j''}^* \geq \max\{\lceil \tau_* \rceil - N_{i''j''}, 0\}$ for all $i'', j'' \in \mathcal{A}$. Particularly, $T_{i,j}^* \geq \max\{\lceil \tau_* \rceil - \frac{N_i N_j}{N_i + N_j}, 0\} + 1$, this implies:

$$\sum_{i \leq j} T_{i,j}^* > \sum_{i \leq j} \max\{\lceil \tau_* \rceil - \frac{N_i N_j}{N_i + N_j}, 0\} \geq \sum_{i \leq j} \max\{\tau_* - \frac{N_i N_j}{N_i + N_j}, 0\} = \sum_{i \leq j} t_{i,j}^* = T, \qquad (33)$$

where the penultimate equality comes by the property of the optimal solution of the Linear Programming problem in Theorem 1: Since by the Linear Programming problem, $\forall k, l \in [K]$, $k \leq l$, let $\epsilon_{k,l} = t_{k,l}^* - \max\{\tau_* - \frac{N_k N_l}{N_k + N_l}, 0\} \geq 0$. If $\exists k', l' \in [K]$ such that $\epsilon_{k',l'} > 0$, then the solution $(\tilde{\tau}, \{\tilde{t}_{k,l}\}_{k,l \in [K]})$ defined as $\tilde{\tau} = \tau_* + \frac{2\epsilon_{k',l'}}{K(K+1)}$, $\tilde{t}_{k,l} = t_{k,l}^* + \frac{2\epsilon_{k',l'}}{K(K+1)}$ for all $(k,l) \in [K] \times [K] \backslash \{(k', l'), (l', k')\}$, and $\tilde{t}_{k',l'} = t_{k',l'}^* - \frac{K(K+1)-2}{K(K+1)}\epsilon_{k',l'}$ could also be a feasible solution to the linear programming problem. This implies $\tilde{\tau} > \tau^*$, hence the optimal case could only have $\epsilon = 0$, implying the establishment of the equality.

Hence, Equation (33) violated the feasible region of the Linear Programming problem in Lemma 7. Thereby, we have two distinct arm pairs $(a_i, a_j)$ and $(a_{i'}, a_{j'})$ such that:

$$T_{i,j}^* + \frac{N_i N_j}{N_i + N_j} \geq \max\{\lceil \tau_* \rceil, \frac{N_i N_j}{N_i + N_j}\} + 1, \text{ in particular } T_{i,j}^* \geq 1,$$

$$T_{i'j'}^* + \frac{N_{i'} N_{j'}}{N_{i'} + N_{j'}} \leq \max\{\lceil \tau_* \rceil, \frac{N_{i'} N_{j'}}{N_{i'} + N_{j'}}\} - 1 = \lceil \tau_* \rceil - 1.$$

To establish the contradiction argument, let $k \leq l$ with $k, l \in [K] \times [K]$, consider another feasible solution:

$$\tilde{T}_{k,l} = \begin{cases} \tilde{T}_{k,l}^* - 1 & \text{if } (k,l) = (i,j), \\ \tilde{T}_{k,l}^* + 1 & \text{if } (k,l) = (i',j'), \\ \tilde{T}_{k,l}^* & \text{otherwise.} \end{cases}$$

but then we have:

$$\sum_{k \leq l} \sum_{t_{k,l}=1}^{\tilde{T}_{k,l}} \sqrt{\frac{1}{t_{k,l} + \frac{N_k N_l}{N_k + N_l}}} - \sum_{k \leq l} \sum_{t_{k,l}=1}^{T_{k,l}^*} \sqrt{\frac{1}{t_{k,l} + \frac{N_k N_l}{N_k + N_l}}}$$

$$= \sqrt{\frac{1}{T_{i'j'}^* + \frac{N_{i'} N_{j'}}{N_{i'} + N_{j'}} + 1}} - \sqrt{\frac{1}{T_{i,j}^* + \frac{N_i N_j}{N_i + N_j}}}$$

$$\geq \sqrt{\frac{1}{\lceil \tau_* \rceil}} - \sqrt{\frac{1}{\max\{\lceil \tau_* \rceil, \frac{N_i N_j}{N_i + N_j}\} + 1}} > 0,$$

this implies $T_{i,j}^*$ is not the optimal solution, thereby $T_{i,j}^* \leq \max\{\lceil \tau_* \rceil - \frac{N_i N_j}{N_i + N_j}, 0\}$, as desired. $\qquad\square$

*Proof.* (Complete Proof of Instance Independent Upper Bound) By Lemma 4, arm pair $(a_i, a_j)$ has the chance to be selected only if:

$$\max\{\Delta_i, \Delta_j\} \leq \min\left\{ 4\sqrt{\frac{\log(1/\delta_t)}{2T_{i,j}}}, 4\sqrt{\frac{1}{2\left(T_{i,j} + \frac{N_i N_j}{N_i + N_j}\right)} \log\left((1/\delta_t)\right)} + 2\frac{\frac{N_i N_j}{N_i + N_j}}{T_{i,j} + \frac{N_i N_j}{N_i + N_j}} \omega_{i,j} \right\}.$$

Denote $\text{rad}_{i,j} = \sqrt{\frac{\log(1/\delta_t)}{2T_{i,j}}}$ and $\text{rad}_{i,j}^{\text{hyb}} = \sqrt{\frac{\log(1/\delta_t)}{2\left(T_{i,j} + \frac{N_i N_j}{N_i + N_j}\right)}} + \frac{\frac{N_i N_j}{N_i + N_j}}{T_{i,j} + \frac{N_i N_j}{N_i + N_j}} V_{i,j}$, hence we have:

$$\text{Reg}(T) = \sum_{t=1}^{T} \frac{\Delta_{A_1(t)} + \Delta_{A_2(t)}}{2}$$

$$\leq \sum_{t=1}^{T} \max\{\Delta_{A_1(t)}, \Delta_{A_2(t)}\}$$

$$\overset{(*)}{\leq} \sum_{t=1}^{T} \min\{4\text{rad}_{A_1(t), A_2(t)}, 4\text{rad}_{A_1(t), A_2(t)}^{\text{hyb}}\}$$

$$\leq \min\left\{ 4\sum_{i \leq j} \sum_{t=1}^{T_{i,j}} \text{rad}_{i,j}, 4\sum_{i \leq j} \sum_{t=1}^{T_{i,j}} \text{rad}_{i,j}^{\text{hyb}} \right\}.$$

The Inequality (*) established due to Lemma 4 and the fact that $\omega_{i,j} = V_{i,j} + p_{i,j}^{\text{off}} - p_{i,j} \leq 2V_{i,j}$. For the first term $4\sum_{i \leq j} \sum_{t=1}^{T_{i,j}} \text{rad}_{i,j}$, we have:

$$4\sum_{i \leq j} \sum_{t=1}^{T_{i,j}} \sqrt{\frac{\log(1/\delta_t)}{2T_{i,j}}} \leq 8\sum_{i \leq j} \sqrt{T_{i,j} \log(1/\delta_T)} = 8\sqrt{\log(1/\delta_T)} \sum_{i \leq j} 1 \cdot \sqrt{T_{i,j}}$$

$$\leq 8\sqrt{\log(1/\delta_T)} \sqrt{\sum_{i \leq j} 1^2} \sqrt{\sum_{i \leq j} T_{i,j}} = 8\sqrt{\frac{K(K+1)}{2} T \log(1/\delta_T)}. \qquad (34)$$

The last inequality follows from applying the Cauchy–Schwarz inequality. Since this bound corresponds to the vanilla UCB setting without offline data, it is already covered by the result provided in Section C.7.1, and is therefore omitted from the final regret bound.

For the Second term $4 \sum_{i \leq j} \sum_{t=1}^{T_{i,j}} \mathrm{rad}_{i,j}^{\mathrm{hyb}}$, we have:

$$\sum_{i \leq j} \sum_{t=1}^{T_{i,j}} 4 \sqrt{\frac{\log\left(1/\delta_T\right)}{2\left(t + \frac{N_i N_j}{N_i + N_j}\right)}} + 4 \frac{\frac{N_i N_j}{N_i + N_j}}{t + \frac{N_i N_j}{N_i + N_j}} V_{i,j}$$

$$\leq 4 \max_{i \leq j} V_{i,j} \cdot T + \sqrt{\log\frac{1}{\delta_T}} \sum_{i \leq j} \sum_{t=1}^{T_{i,j}} 4 \sqrt{\frac{1}{2\left(t + \frac{N_i N_j}{N_i + N_j}\right)}}. \tag{35}$$

Let $T_{i,j}^*$ be the optimal solution of the LP problem in Lemma 7. By applying Lemma 7, we obtain:

$$\sum_{i \leq j} \sum_{t=1}^{T_{i,j}} \sqrt{\frac{1}{t + \frac{N_i N_j}{N_i + N_j}}} \leq \sum_{i \leq j} \sum_{t=1}^{T_{i,j}^*} \sqrt{\frac{1}{t + \frac{N_i N_j}{N_i + N_j}}} \leq \sum_{i \leq j} \sum_{t=1}^{\max\{\lceil \tau_* \rceil - \frac{N_i N_j}{N_i + N_j}, 0\}} \sqrt{\frac{1}{t + \frac{N_i N_j}{N_i + N_j}}},$$

thus:

$$\sum_{i \leq j} \sum_{t=1}^{\max\{\lceil \tau_* \rceil - \frac{N_i N_j}{N_i + N_j}, 0\}} \sqrt{\frac{1}{t + \frac{N_i N_j}{N_i + N_j}}} \leq \sum_{i \leq j} \frac{\max\{\lceil \tau_* \rceil - \frac{N_i N_j}{N_i + N_j}, 0\}}{\lceil \tau_* \rceil} \sum_{t=1}^{\lceil \tau_* \rceil} \sqrt{\frac{1}{t}}$$

$$\leq \sum_{i \leq j} \max\{\lceil \tau_* \rceil - \frac{N_i N_j}{N_i + N_j}, 0\} \cdot \frac{2}{\sqrt{\tau_*}}$$

$$\leq \sum_{i \leq j} \frac{4 t_{ij}^*}{\sqrt{\tau_*}} = \frac{4T}{\sqrt{\tau_*}}.$$

where the last inequality follows by applying the feasibility of the LP problem in Theorem 2.

Replacing term $\sum_{i \leq j} \sum_{t=1}^{T_{i,j}} 4 \sqrt{\frac{1}{2\left(t + \frac{N_i N_j}{N_i + N_j}\right)}}$ with $\frac{16T}{\sqrt{\tau^*}}$ in Equation (35) leads to the desired result. $\qquad \square$

## D  Proof of Theorem 2

The proof of the regret lower bound builds upon the approach in Cheung and Lyu (2024), which we have extended to accommodate the hybrid setting.

**Theorem 4.** *Let $\mathbb{P}, \mathbb{Q}$ be probability distributions on $(\Omega, \mathcal{F})$. For an event $E \in \sigma(\mathcal{F})$, it holds that*

$$\Pr_{\mathbb{P}}(E) + \Pr_{\mathbb{Q}}(E^c) \geq \frac{1}{2} \exp\left(-KL(\mathbb{P}, \mathbb{Q})\right).$$

**Theorem 5.** *Consider two instances $\mathcal{I}_P$ and $\mathcal{I}_Q$ that share the same arm set $\mathcal{A}$, online phase horizon $T$, and offline sample size $\{N_i\}_{i \in [K]}$, but have different reward distributions $P = (P_i, P_i^{off})_{i \in [K]}$ and $Q = (Q_a, Q_a^{off})_{i \in [K]}$. For any non-anticipatory policy $\pi$, it holds that*

$$KL(P, Q) = \sum_{i \leq j} \mathbb{E}_{P,\pi}[T_{i,j}(T)] \cdot KL(P_{i,j}, Q_{i,j}) + \sum_{i \in [K]} N_i \cdot KL(P_i^{off}, Q_i^{off}).$$

Theorem 4 is a direct restatement of Lemma 15.1 from Lattimore and Szepesvári (2020), while Theorem 5 is adapted from Theorem C.2 in Cheung and Lyu (2024). The only difference in our setting is that the relative feedback data follows a discrete distribution; nonetheless, by following the original proof strategy, we are able to derive Theorem 5 accordingly.

**Claim 1.** *For $P_i = \mathcal{N}(\mu_i, \sigma^2)$, where $i \in \{1, 2\}$, we have*

$$KL(P_1, P_2) = \frac{(\mu_1 - \mu_2)^2}{2\sigma^2}.$$

**Claim 2.** *For $P_i = Bernoulli(p_i)$, where $i \in \{1, 2\}$, we have*

$$KL(P_1, P_2) = p_1 \log \frac{p_1}{p_2} + (1 - p_1) \log \left(\frac{1 - p_1}{1 - p_2}\right) \leq \frac{(p_1 - p_2)^2}{p_2(1 - p_2)},$$

where the last inequality of Claim 2 comes by applying $\log x \leq x - 1$.

### D.1 Instance dependent lower bound

*Proof.* We begin by denoting $\mu_i^{\text{off}}$ and $\mu_i$ as the means of the Gaussian distributions of $P_i^{\text{off}}$ and $P_i$, respectively, for each arm $a \in \mathcal{A}$. Assume $\mu_1$ corresponds to the optimal arm, while the remaining arms are suboptimal, with $\mu_i = \mu_j$ for all $i, j \neq 1$. The gap between the optimal arm $a_1$ and the suboptimal arms $(a_2, \ldots, a_K)$ is denoted by $\Delta$, where $\Delta = \mu_1 - \mu_i \in (0, \frac{1}{2})$.

Next, we introduce an alternative distribution $Q$. Let $k$ be a fixed arm in $\{2, \cdots, K\}$ and $T_i(t)$ represents the number of times arm $i$ is selected in pairs and pulled up to time $t$. For all $i \in [K] \setminus \{k\}$, define $Q_i^{\text{off}} = P_i^{\text{off}}$ and $Q_i = P_i$. For arm $k$, let $Q_k = \mathcal{N}(\mu_k + 2\Delta, 1)$. Define $Q_k^{\text{off}}$ as follows:

$$Q_k^{\text{off}} = \begin{cases} \mathcal{N}(\mu_k^{\text{off}}, 1) & \text{if } \mu_k^{\text{off}} \geq \mu_k + 2\Delta - V_k, \\ \mathcal{N}(\mu_k + 2\Delta - V_k, 1) & \text{if } \mu_k^{\text{off}} < \mu_k + 2\Delta - V_k, \end{cases} \tag{36}$$

where $V_k \geq |\mu_k^{\text{off}} - \mu_k|$. The construction of $Q_k^{\text{off}}$ in Equation (36) is governed by $V_k$. The high-level idea behind it is to maximize the regret lower bound by minimizing the KL divergence between $Q_k^{\text{off}}$ and $P_k^{\text{off}}$. However, directly setting $Q_k^{\text{off}} = P_k^{\text{off}}$ may violate the constraint $Q \in \mathcal{I}_V$. When the constraint is violated (e.g. $\mu_k^{\text{off}} < \mu_k + 2\Delta - V_k$), we instead define $Q_k^{\text{off}} = \mathcal{N}(\mu_k + 2\Delta - V_k, 1)$, which yields the smallest KL divergence possible while preserving $Q \in \mathcal{I}_V$.

Under the dueling bandit setting with the Bradley–Terry model, the pairwise preference gaps under distribution $P$ is given by:

$$\Delta_i = p_{1,i} - \frac{1}{2} = \sigma(\Delta) - \frac{1}{2}, \forall i \in [K].$$

Under the perturbed distribution Q, the pairwise gaps are defined as

$$\Delta_{k,i} = p_{k,i} - \frac{1}{2} = \sigma(2\Delta) - \frac{1}{2}, \forall i \in [K] \setminus \{1, k\}, \quad \text{and} \quad \Delta_{k,1} = \sigma(\Delta) - \frac{1}{2}.$$

Using the definition of regret under distribution $P$, we derive:

$$\text{Reg}_P(T) = T\left(\sigma(\Delta) - \frac{1}{2}\right) - \frac{\left(\sigma(\Delta) - \frac{1}{2}\right)}{2}\mathbb{E}_P\left[T_1(T)\mathbf{1}_{T_1(T) \leq T} + T_1(T)\mathbf{1}_{T_1(T) > T}\right]$$

$$\geq T\left(\sigma(\Delta) - \frac{1}{2}\right) - \frac{\left(\sigma(\Delta) - \frac{1}{2}\right)}{2}\mathbb{E}_P\left[T_1(T)\mathbf{1}_{T_1(T) \leq T}\right] - \frac{\left(\sigma(\Delta) - \frac{1}{2}\right)}{2}\mathbb{E}_P\left[T_1(T)\mathbf{1}_{T_1(T) > T}\right]$$

$$= T\left(\sigma(\Delta) - \frac{1}{2}\right) - \frac{T\left(\sigma(\Delta) - \frac{1}{2}\right)}{2}\Pr_P(T_1(T) \leq T) - T\left(\sigma(\Delta) - \frac{1}{2}\right)\Pr_P(T_1(T) > T)$$

$$= \frac{T\left(\sigma(\Delta) - \frac{1}{2}\right)}{2}\Pr_P(T_1(T) \leq T).$$

For distribution $Q$, the regret is lower bounded as:

$$\text{Reg}_Q(T) = \sum_{i=1}^{K} \frac{\Delta_i}{2}\mathbb{E}_Q(T_i(T)) \geq \frac{\sigma(\Delta) - \frac{1}{2}}{2}\mathbb{E}_Q(T_1(T)) \geq \frac{T\left(\sigma(\Delta) - \frac{1}{2}\right)}{2}\Pr_Q(T_1(T) > T).$$

Combining these, we obtain:

$$2CT^p \geq \text{Reg}_P(T) + \text{Reg}_Q(T)$$

$$\geq \frac{T\left(\sigma(\Delta) - \frac{1}{2}\right)}{2}\left(\Pr_P[T_1(T) \leq T] + \Pr_Q[T_1(T) > T]\right)$$

$$\geq \frac{T\left(\sigma(\Delta) - \frac{1}{2}\right)}{4}\exp(-KL(P, Q))$$

$$\geq \frac{T\Delta}{32}\exp(-KL(P, Q)). \tag{37}$$

where the first inequality follows from the definition of Cp-consistent, the penult inequality applies Theorem 4, and the final inequality comes by our restriction on $\Delta$: Since $\Delta \in (0, \frac{1}{2})$, we have $\sigma(\Delta) \geq \frac{1}{8}\Delta + \frac{1}{2}$.

By Theorem 5, the KL divergence is:

$$KL(P, Q) = \sum_{i \leq j} \mathbb{E}_P[T_{i,j}(T)] \cdot KL(P_{i,j}, Q_{i,j}) + \sum_{i \in [K]} N_i \cdot KL(P_i^{\text{off}}, Q_i^{\text{off}}).$$

For the first term, using $d(p,q) = p\log\left(\frac{p}{q}\right) + (1-p)\log\left(\frac{1-p}{1-q}\right) \leq \frac{(p-q)^2}{q(1-q)}$, we have:

$$\sum_{i,j} \mathbb{E}_P[T_{i,j}(T)]\mathrm{KL}(P_{i,j}, Q_{i,j}) = \sum_{i=2}^{K} \mathbb{E}_P[T_{k,i}(T)]d\left(\frac{1}{2}, \sigma(2\Delta)\right) + \mathbb{E}_P[T_{k,1}(T)]d\left(1 - \sigma(\Delta), \sigma(\Delta)\right)$$

$$\overset{(a)}{\leq} \sum_{i=2}^{K} \mathbb{E}_P[T_{k,i}(T)]d\left(\frac{1}{2}, \frac{1}{2}\Delta + \frac{1}{2}\right) + \mathbb{E}_P[T_{k,1}(T)]d\left(\frac{1}{2} - \frac{1}{4}\Delta, \frac{1}{2} + \frac{1}{4}\Delta\right)$$

$$\leq \sum_{i=2}^{K} \mathbb{E}_P[T_{k,i}(T)]\frac{\Delta^2}{1 - \Delta^2} + \mathbb{E}_P T_{k,1}(T)\frac{\Delta^2}{1 - \frac{1}{4}\Delta^2}$$

$$\leq \mathbb{E}_P[T_k(T)]\frac{\Delta^2}{1 - \Delta^2}$$

$$\leq \mathbb{E}_P[T_k(T)]2\Delta^2, \tag{38}$$

where inequality (a) follows by $\sigma(\Delta) \leq \frac{1}{4}\Delta + \frac{1}{2}$, this implies:

$$d\left(\frac{1}{2}, \sigma(\Delta)\right) \leq d\left(\frac{1}{2}, \frac{1}{4}\Delta + \frac{1}{2}\right) \quad \text{and} \quad d\left(1 - \sigma(\Delta), \sigma(\Delta)\right) \leq d\left(\frac{1}{2} - \frac{1}{4}\Delta, \frac{1}{2} + \frac{1}{4}\Delta\right).$$

For the second term, by our setup, we have:

$$\sum_{i \in [K]} N_i \cdot KL(P_i^{\mathrm{off}}, Q_i^{\mathrm{off}}) = N_k \frac{\max\{2\Delta - \omega_k, 0\}^2}{2}, \tag{39}$$

where $\omega_k = V_k + \mu_k^{\mathrm{off}} - \mu_k$. Combining Inequality (37), (38) and Equation (39), we get:

$$2CT^p \geq \frac{T\Delta}{32} \exp\left\{-\mathbb{E}_P[T_k(T)]2\Delta^2 - \frac{\max\{2\Delta - \omega_k, 0\}^2}{2}N_k\right\}.$$

To isolate $\mathbb{E}_P[T_k(T)]$, we take the natural logarithm of both sides and rearrange the inequality, yielding:

$$\mathbb{E}[T_k(T)] \geq \frac{1}{2\Delta^2}\left((1-p)\log T + \log\frac{\Delta}{64C} - \frac{\max\{2\Delta - \omega_k, 0\}^2}{2}N_k\right).$$

Summing over all arms $i \in [K]$, the total regret lower bound becomes:

$$\sum_{i \in [K]} \frac{\Delta}{2}\mathbb{E}[T_i(T)] \geq \frac{K}{4\Delta}\left((1-p)\ln T + \ln\frac{\Delta}{64C}\right) - \frac{1}{4\Delta}\sum_{i \in [K]} \frac{\max\{2\Delta - \omega_i, 0\}^2}{2}N_i,$$

as desired. $\qquad \square$

**Discussion about upper and lower bounds.** Consider the special case where $N_i = n$ for all $i \in [K]$ and $\Delta_i = \Delta$ for all sub-optimal arms. In this case, the upper and lower bounds differ by at most a factor of $K$.

*Proof.* We compare the upper and lower bounds separately for the instance-independent and instance-dependent cases. For the instance-independent bounds, we first compare the first term in the upper bound (Theorem 1(b)) and the lower bound (Theorem 2(b)). Specifically, the first term in the upper bound is $O(\sqrt{K^2T\log T})$, while the corresponding term in the lower bound is $\Omega(\sqrt{KT})$. This implies a gap of $\sqrt{K\log T}$ between the upper and lower bounds for this term. We then compare the second terms. The second term in the upper bound can be written as $O((\sqrt{\log T/\tau_*} + V_{\max}) \cdot T)$, and the corresponding term in the lower bound is $\Omega((\sqrt{1/\tau_*'} + V_{\max}) \cdot T)$. To compare the different parameter $\tau_*$ and $\tau_*'$, we consider a special case where $N_i = n$ for all $i \in [K]$. In this case,

$$\tau_* = \frac{n}{2} + \frac{2T}{K(K+1)}, \quad \tau_*' = n + \frac{2T}{K}.$$

Under this setting, it can be shown that the gap between the upper and lower bounds for the second term is again at most a factor of $\sqrt{K\log T}$.

For the instance-dependent case, to better compare the upper and lower bounds, we consider the special case where $N_i = n$ for all $i \in [K]$ and $\Delta_i = \Delta$ for all sub-optimal arms. Then the upper bound in Theorem 2(a) simplifies to

$$O\left(\frac{K^2\log T}{\Delta} - \frac{K^2n}{4}\sum_{i \leq j}\max\{\Delta - 4\omega_{i,j}, 0\}\right),$$

and the lower bound is in the form of:

$$\Omega\left(\frac{K\log T}{\Delta} + \frac{K}{4\Delta}\log\frac{\Delta}{64C} - Kn\sum_{i\in[K]}\max\{\Delta - \omega_i/2, 0\}^2\right).$$

From this comparison, we observe that the upper bound is approximately a factor of $K$ worse than the lower bound in this case. $\qquad\square$

## D.2 Instance independent regret lower bound

*Proof.* The divide the proof into three distinct cases:

**Case 1:** $2\sqrt{KT} > T \cdot (V_{\max} + \sqrt{2/\tau'_*})$, **and** $V_{\max} \leq 1/\sqrt{\tau'_*}$, we derive a regret lower bound of:

$$\Omega\left(\min\left\{\sqrt{KT}, \left(\sqrt{\frac{1}{\tau'_*}} + V_{\max}\right)\cdot T\right\}\right) = \Omega\left(\frac{T}{\sqrt{\tau'_*}}\right).$$

At this point, we consider a setting where the bias between offline and online feedback is negligible. We define the Gaussian reward distributions as follows, assuming no discrepancy between offline and online feedback:

$$P_i^{\text{off}} = P_i = \begin{cases} N(\Delta, 1) & \text{if } i = 1, \\ N(0, 1) & \text{if } i \neq 1. \end{cases} \qquad Q_i^{\text{off}} = Q_i = \begin{cases} N(\Delta, 1) & \text{if } i = 1, \\ N(2\Delta, 1) & \text{if } i = k, \\ N(0, 1) & \text{if } i \in [K]\backslash\{k, 1\}. \end{cases}$$

where $\Delta = 1/\sqrt{\tau'_*}$ and $k = \arg\min_{i\in[K]} N_i + \mathbb{E}_P(T_i)$.

Obviously $P, Q \in \mathcal{I}_V$, since $|\mu_i^{\text{off}} - \mu_i| = 0 \leq V_{\max}$. Without loss of generality, we assume that arm 1 has the largest offline sample size, that is, $1 = \arg\max_{i\in[K]} N_i$.

**Subcase 1.1:** $k = 1$.

In this case, we have:

$$\mathbb{E}_P[T_1] + N_1 \leq \mathbb{E}_P[T_i] + N_i \quad \text{for all } i \in [K]\setminus\{1\}.$$

Since we assume $1 = \arg\max_{i\in[K]} N_i$, it follows that:

$$\mathbb{E}_P[T_1] \leq \mathbb{E}_P[T_i] + N_i - N_1 \leq \mathbb{E}_P[T_i], \quad \text{for all } i \in [K]\setminus\{1\}.$$

Therefore, arm 1 has both the largest offline sample size and the smallest expected number of online pulls among all arms. By the pigeonhole principle, this implies that the optimal arm $i \neq 1$ is pulled at most $2T/K$ times. Thereby, the expected cumulative regret can then be bounded as:

$$\text{Reg}(T) \geq \left(2T - \frac{2T}{K}\right)\cdot\frac{\Delta}{2} = \frac{(K-1)\Delta T}{K} = \frac{(K-1)T}{K\sqrt{\tau'_*}}.$$

**Subcase 1.2:** $k \neq 1$ :

$$\begin{aligned} \text{Reg}_P(T) + \text{Reg}_Q(T) &\geq \frac{1}{2}\left[\sigma(\Delta) - \frac{1}{2}\right]\left(\Pr_P(\mathbb{E}[T_1] \leq T) + \Pr_Q(\mathbb{E}[T_1] > T)\right) \\ &\geq \frac{T\left[\sigma(\Delta) - \frac{1}{2}\right]}{4}\cdot\exp\left(-\sum_{i=1}^{K}\mathbb{E}[T_i]\cdot\text{KL}(P_{k,i}, Q_{k,i}) - N_k\cdot\text{KL}\left(P_i^{\text{off}}, Q_i^{\text{off}}\right)\right) \\ &\overset{(1)}{\geq} \frac{T\Delta}{32}\exp\left(-2\mathbb{E}[T_k]\Delta^2 - 2N_k\Delta^2\right) \\ &\overset{(2)}{\geq} \frac{T\Delta}{32}\exp\left(-2\tau'_*\Delta^2\right) \\ &\overset{(3)}{=} \frac{1}{32e^2}\cdot\frac{T}{\sqrt{\tau'_*}}, \end{aligned}$$

where inequality (1) follows by applying the same technique in D.1, inequality (2) hold since $k = \arg\min_{i\in[K]} N_i + \mathbb{E}_P[T_i] \leq \tau'_*$, and the final equality (3) comes by applying $\Delta = 1/\sqrt{\tau'_*}$.

**Case 2:** $2\sqrt{KT} > T \cdot (V_{\max} + \sqrt{2/\tau'_*})$, **and** $V_{\max} \geq 1/\sqrt{\tau'_*}$, we derive a regret lower bound of:

$$\Omega\left(\min\left\{\sqrt{KT}, \left(\sqrt{\frac{1}{\tau'_*}} + V_{\max}\right)\cdot T\right\}\right) = \Omega\left(T\cdot V_{\max}\right).$$

We construct another instance such that:

$$P_i^{\text{off}} = \mathcal{N}(0,1), \; \forall i \in [K], \quad P_i = \begin{cases} \mathcal{N}(\Delta, 1) & \text{if i} = 1, \\ \mathcal{N}(0,1) & \text{if i} \neq 1. \end{cases}$$

$$Q_i^{\text{off}} = \mathcal{N}(0,1), \; \forall i \in [K], \quad Q_i = \begin{cases} \mathcal{N}(\Delta, 1) & \text{if i} = 1, \\ \mathcal{N}(2\Delta, 1) & \text{if i} = k, \\ \mathcal{N}(0,1) & \text{if i} \in [K] \backslash \{k, 1\}. \end{cases}$$

where $k = \arg\min_{i \in [K]} \mathbb{E}[T_i]$, and take $\Delta = 1/2 \cdot V_{\max}$.

Since $|\mu_i^{\text{off}} - \mu_i| \leq 2\Delta = V_{\max} = V_i$ for all $i \in [K]$, this implies $P, Q \in \mathcal{I}_V$. Furthermore:

$$\begin{aligned} \text{Reg}_P(T) + \text{Reg}_Q(T) &\geq \frac{1}{2}\left[\sigma(\Delta) - \frac{1}{2}\right]\left(\Pr_P(\mathbb{E}[T_1] \leq T) + \Pr_Q(\mathbb{E}[T_1] > T)\right) \\ &\geq \frac{T\left[\sigma(\Delta) - \frac{1}{2}\right]}{4} \cdot \exp\left(-\sum_{i=1}^{K} \mathbb{E}[T_i] \cdot \text{KL}(P_{k,i}, Q_{k,i}) - N_k \cdot \text{KL}(P_i^{\text{off}}, Q_i^{\text{off}})\right) \\ &\geq \frac{T\Delta}{32}\exp\left(-2\mathbb{E}[T_k]\Delta^2\right) \\ &\geq \frac{TV_{\max}}{64}\exp\left(-\frac{TV_{\max}^2}{K-1}\right) \\ &\geq \frac{1}{64e^8}TV_{\max}, \end{aligned}$$

where the last inequality comes by the condition $2\sqrt{KT} > TV_{max} + \sqrt{2/\tau_*'} > TV_{max}$, this implies $TV_{\max}^2/(K-1) \leq 4K/(K-1) \leq 8$.

**Case 3: When** $2\sqrt{KT} \leq T \cdot (V_{\max} + \sqrt{2/\tau_*'})$, we derive a lower bound of

$$\Omega\left(\min\left\{\sqrt{KT}, \left(\sqrt{\frac{1}{\tau_*'}} + V_{\max}\right) \cdot T\right\}\right) = \Omega\left(\sqrt{KT}\right).$$

The analysis for this case largely follows that of **Case 2**. We use the same construction of reward distributions $P$, $Q$, but now take $\Delta = \sqrt{(K-1)/(4T)}$.

From the properties of the associated Linear Program problem in Theorem 2, we know that $\tau_*' \geq 2T/K$, this implies: $\sqrt{2T}/\sqrt{\tau_*'} \leq \sqrt{KT}$, them by the condition under this case, we get:

$$\sqrt{KT} \leq TV_{\max} + \sqrt{2T}/\sqrt{\tau_*'} - \sqrt{KT} \leq TV_{\max},$$

which further leads to:

$$|\mu_i^{\text{off}} - \mu_i| \leq 2\Delta = \sqrt{\frac{K-1}{T}} \leq \frac{1}{T} \cdot \sqrt{KT} \leq V_{\max}, \quad \forall i \in [K].$$

Hence, the constructed distributions $P, Q$ satisfy the bias constraint and belong to the class $\mathcal{I}_V$.

Following the same procedure in Case 2, we have:

$$\begin{aligned} \text{Reg}_P(T) + \text{Reg}_Q(T) &\geq \frac{1}{2}\left[\sigma(\Delta) - \frac{1}{2}\right]\left(\Pr_P(\mathbb{E}[T_1] \leq T) + \Pr_Q(\mathbb{E}[T_1] > T)\right) \\ &\geq \frac{T\Delta}{32}\exp\left(-2\mathbb{E}[T_k]\Delta^2\right) \\ &\geq \frac{1}{64e}\sqrt{(K-1)T}. \end{aligned}$$

The Theorem is proved. $\qquad\qquad\square$

# E  Proof of Theorem 3 and Supplementary Regret Bounds

## E.1  Instance Dependent Regret Upper Bound

The analysis of this algorithm is similar to Algorithm 1, the only key change here is the position of the relative data and the stochastic data, therefore, the $\text{UCB}^{\text{hyb}}$, UCB and good event $\mathcal{E}$ changed to the structure below. Note

that elimination based algorithm shuold consider all cases from $t = 1, \cdots, T$, first we define good events for both hybrid and online UCBs, let

$$\mathcal{E}_t^{\text{hyb}}(p_{i,j}) = \left\{ p_{i,j} \leq \text{UCB}^{\text{hyb}}(a_i, a_j) \leq p_{i,j} + 2\sqrt{\frac{\log(1/\delta_t)}{2(N_{i,j} + \frac{T_i(t)T_j(t)}{T_i(t)+T_j(t)})}} + \frac{N_{i,j}}{N_{i,j} + \frac{T_i(t)T_j(t)}{T_i(t)+T_j(t)}}\omega_{i,j} \right\},$$

$$\mathcal{E}_t(p_{i,j}) = \left\{ \mu_i \leq \text{UCB}(a_i, a_j) \leq p_{i,j} + 2\sqrt{\frac{\log(1/\delta_t)}{2} \cdot \frac{T_i + T_j}{T_i T_j}} \right\}.$$

with

$$\mathcal{E} = \bigcap_{t \in [T]} \bigcap_{i,j \in [K]} \left( \mathcal{E}_t(p_{i,j}) \cap \mathcal{E}_t^{\text{hyb}}(p_{i,j}) \right).$$

Following the proof of Lemma 3 will derive $\Pr(\mathcal{E}) \geq 1 - 2K(K+1)\sum_{t=1}^{T} \delta_t$.

Then we move to the analysis of the property of this algorithm, which is to detect the point of which an arm is eliminated at time $t$. we claim the result as follows:

**Claim 3.** *For Algorithm 2 under good event $\mathcal{E}$, arm $a_i$ will be eliminated if it satisfies the following inequality:*

$$\Delta_i > \min \left\{ 2\sqrt{\frac{\log(1/\delta_t)}{2N_{1,i} + T_i(t)}} + \frac{2N_{1,i}}{2N_{1,i} + T_i(t)}\omega_{1,i}, \ 2\sqrt{\frac{\log(1/\delta_t)}{T_i(t)}} \right\}.$$

*Proof of the claim.* According to the elimination criterion of HybUCB-RA (Line 6), arm $a_i$ will be eliminated if either of the following inequalities holds:

1. $\text{UCB}(a_i, a_j) < \frac{1}{2}, \forall a_j \in \mathcal{A}$,

2. $\text{UCB}^{\text{hyb}}(a_i, a_j) < \frac{1}{2}, \forall a_j \in \mathcal{A}$.

This also includes:

$$\text{UCB}(a_i, a_1) < \frac{1}{2} \quad \text{and} \quad \text{UCB}^{\text{hyb}}(a_i, a_1) < \frac{1}{2}.$$

When $\Delta_i > 2\sqrt{\frac{\log(1/\delta_t)}{T_i(t)}}$, and the event $\mathcal{E}$ holds, we derive

$$\text{UCB}(a_i, a_1) \leq p_{i,1} + 2\sqrt{\frac{\log(1/\delta_t)}{2} \cdot \frac{T_i + T_j}{T_i T_j}} \approx p_{i,1} + 2\sqrt{\frac{\log(1/\delta_t)}{T_i(t)}} < \frac{1}{2},$$

where the first inequality follows from the definition of event $\mathcal{E}$, and the final inequality results from the condition $\Delta_i > 2\sqrt{\frac{\log(1/\delta_t)}{T_i(t)}}$. The approximation holds because when arm $a_i$ remains active in round $t$, we have $|T_i(t) - T_1(t)| \leq 1$, making $\frac{T_i(t)T_1(t)}{T_i(t)+T_1(t)} = \frac{T_i(t)}{2} + O(1)$. The case for $\text{UCB}^{\text{hyb}}(a_1, a_i)$ follows symmetrically, we derive when $\Delta_i > 2\sqrt{\frac{\log(1/\delta_t)}{2N_{1,i} + T_i(t)}} + \frac{2N_{1,i}}{2N_{1,i} + T_i(t)}\omega_{i,1}$, the following condition holds:

$$\text{UCB}^{\text{hyb}}(a_i, a_1) \lessapprox 2\sqrt{\frac{\log(1/\delta_t)}{2N_{1,i} + T_i(t)}} + \frac{2N_{1,i}}{2N_{1,i} + T_i(t)}\omega_{i,1} < \frac{1}{2}.$$

Combining them together completes the proof. $\square$

Now we move on to the complete proof of this algorithm.

*Proof.* (Complete Proof of Algorithm 2) Define

$$g(a_i) = \max \left\{ \frac{1}{\Delta_i^2} \left[ 2\log(1/\delta_t) + 2N_{i,1}\Delta_i(\omega_{i,1} - \Delta_i) + \sqrt{4\log^2(1/\delta_t) + 8\log(1/\delta_t)N_{i,1}\omega_{i,1}\Delta_i} \right], \ 0 \right\},$$

$$f(a_i) = \begin{cases} -1 & \text{if } 2\omega_{i,1} \leq \Delta_i, \\ \frac{2\log(1/\delta_t)(2\omega_{i,1} - \Delta_i)}{\Delta_i(\omega_{i,1} - \Delta_i)} - N_{i,1} & \text{if } 2\Delta_i > 2\omega_{i,1} > \Delta_i, \\ 1 & \text{otherwise.} \end{cases}$$

By Lemma 5 and Claim 3, taking $\Delta' = \Delta_i$, $\omega' = \omega_{i,1}$, $N' = 2N_{i,1}$, $T' = T_i$, $\delta' = 1/\delta_t$ we could derive:

$$T_i(t) \leq \begin{cases} \dfrac{4\log(1/\delta_t)}{\Delta_i^2} & \text{if } f(a_i) \geq 0, \\ g(a_i) & \text{otherwise.} \end{cases}$$

Take $\delta_t = \frac{1}{2K(K+1)T^2}$, the expected regret for bad event becomes:

$$T\Delta_{\max} \times \Pr(\mathcal{E}^c) \leq T\Delta_{\max} \cdot 2K(K+1)T \frac{1}{2K(K+1)T^2} = \Delta_{\max}.$$

Combining the good event together, the regret is:

$$\text{Reg}_T(T) = \sum_{t=1}^{T} \mathbb{E}\left[\Delta_i \cdot [\mathbf{1}(\mathcal{E}_t) + \mathbf{1}(\mathcal{E}_t^c)]\right]$$

$$\leq \sum_{t=1}^{T} \mathbb{E}\left[\Delta_i \cdot \mathbf{1}(\mathcal{E}_t)\right] + \Delta_{\max} \sum_{t=1}^{T} \mathbb{E}\left[\mathbf{1}(\mathcal{E}_t^c)\right]$$

$$\leq \Delta_{\max} + \sum_{f(a_i) \leq 0} g(a_i)\Delta_i + \sum_{f(a_i) > 0} \frac{8\log(\sqrt{2K(K+1)}T)}{\Delta_i}.$$

Applying $\sqrt{x^2 + 2ax} < a + x$ to the final regret leads to the simplified version, which is:

$$\text{Reg(T)} = \Delta_{\max} + \sum_{i \in [K]} \max\left\{ \frac{8\log(\sqrt{2K(K+1)}T)}{\Delta_i} - 2N_{i,1}\max\{\Delta_i - 2\omega_{i,1}, 0\}, 0\right\}.$$

$\square$

## E.2 Instance Independent Regret Upper Bound

**Theorem 6.** *Choosing $\delta_t = \frac{1}{2K(K+1)T^2}$, the gap independent bound of HybElimUCB-RA satisfies:*

$$O\left(\min\left\{ \sqrt{KT\log(T)} - \sum_{i \in [K]} Saving(a_i), \left(\sqrt{\frac{\log(T)}{\tau_*}} + V_{\max}\right) \cdot T\right\}\right).$$

*where $Saving(a_i) = \min\left\{ \frac{8\log(\sqrt{2K(K+1)}T)}{\Delta_i}, 2N_{i,1}\max\{\Delta_i - 2\omega_{i,1}, 0\}\right\}$, and $\tau_*$ is the optimal solution to the below linear programming problem:*

$$\begin{aligned} \max_{\tau, t_i} \quad & \tau, \\ s.t. \quad & \tau \leq t_i/2 + N_{i,1} \quad \forall i \in [K], \\ & \sum_{i \in [K]} t_i = T, \\ & \tau \geq 0, t_i \geq 0 \quad \forall i, j \in [K]. \end{aligned}$$

*Proof.* The proof of this instance regret lower bound is symmetric to the proof of the instance dependent bound of HybUCB-AR, which is in C.7.1 and C.7.2. By the definition of regret bound and Claim 3, the expected regret is upper bounded by:

$$\text{Reg}(T) = \sum_{t=1}^{T} \Delta_{(A_t)}$$

$$\leq \sum_{t=1}^{T} \min\left\{ 2\sqrt{\frac{1}{2N_{1,A(t)} + T_{A_t}(t)} \log\left(\frac{1}{\delta_t}\right)} + \frac{2N_{1,A(t)}}{2N_{1,A(t)} + T_{A_t}(t)}\omega_{1,A(t)}, 2\sqrt{\frac{\log(1/\delta_t)}{T_{A_t}(t)}}\right\}$$

$$\leq \sum_{t=1}^{T} \min\left\{ 2\sqrt{\frac{1}{2N_{1,A(t)} + T_{A_t}(t)} \log\left(\frac{1}{\delta_t}\right)} + \frac{4N_{1,A(t)}}{2N_{1,A(t)} + T_{A_t}(t)}V_{1,A(t)}, 2\sqrt{\frac{\log(1/\delta_t)}{T_{A_t}(t)}}\right\}.$$
(40)

The second term of Equation (40) follows the conventional regret analysis, as provided in Appendix C.7.2, Equation (34). The upper bound for the first term can be derived similarly by following the same technique in Section C.7.2, where we replace the term $N_i N_j / (N_i + N_j)$ with $N_{1,i}$ and $t_{i,j}$ with $t_i/2$. For the sake of conciseness, we omit the detailed of the proof here. $\square$

### E.3 Regret Lower Bound

**Theorem 7.** *Let $\Delta \in (0, \frac{1}{2})$ be the gap between the optimal arm and all suboptimal arms. The regret lower bounds of HybElimUCB-RA satisfy:*

*(a) Instance-dependent bound:*

$$\Omega \left( \frac{8K}{\Delta} \left( (1-p)\log T + \log \frac{\Delta}{64C} - \sum_{i \in [K]} \frac{\max\{2\Delta - \omega_i, 0\}^2}{3} N_i \right) \right),$$

*where $C, p$ are constants from Definition 1, and $\omega_i = V_i + \mu_i^{\text{off}} - \mu_i$ with $V_i \geq |\mu_i^{\text{off}} - \mu_i|$.*

*(b) Instance-independent bound:*

$$\Omega \left( \min \left\{ \sqrt{KT}, \left( \sqrt{\frac{1}{\tau_*'}} + V_{\max} \right) \cdot T \right\} \right).$$

*where $\tau_*'$ is the optimal solution to the following Linear Programming problem:*

$$\max_{\tau', t_{i,j}} \quad \tau',$$
$$\text{subject to} \quad \tau' \leq t_i + N_i, \quad \forall i \in [K],$$
$$\sum_{i \in [K]} t_i = T, \quad \tau' \geq 0, \ t_i \geq 0.$$

*Proof.* This proof follows a similar structure to the lower bound argument presented in Appendix D. Specifically, let $k$ be a fixed arm in $\{2, \cdots, K\}$, we construct two instances $P$ and $Q$ such that

$$P_i^{\text{off}} = N(\mu^{\text{off}}, 1) \ \forall i \in [K], \quad P_i = \begin{cases} N(\mu + \Delta, 1) & \text{if } i = 1, \\ N(\mu, 1) & \text{if } i \neq 1. \end{cases}$$

$$Q_i^{\text{off}} = \begin{cases} \mathcal{N}(\mu + 2\Delta - V_k, 1) & \text{if } i = k \text{ and } \mu^{\text{off}} < \mu + 2\Delta - V_k, \\ \mathcal{N}(\mu^{\text{off}}, 1) & \text{otherwise,} \end{cases} \quad Q_i = \begin{cases} \mathcal{N}(\mu + \Delta, 1) & \text{if } i = 1, \\ \mathcal{N}(\mu + 2\Delta, 1) & \text{if } i = k, \\ \mathcal{N}(\mu, 1) & \text{if } i \in [K] \backslash \{1, k\}. \end{cases}$$

where $\mu^{\text{off}}$ and $\mu$ could be any values such that $|\mu^{\text{off}} - \mu| \leq V_i$ for all $i \in [K] \backslash \{1\}$ and $|\mu^{\text{off}} - (\mu + \Delta)| \leq V_1$. In addition, $P \in \mathcal{I}_V$ implies $Q \in \mathcal{I}_V$. Following the conventional analysis, we have:

$$\text{Reg}_P(T) = T \left( \sigma(\Delta) - \frac{1}{2} \right) - \left( \sigma(\Delta) - \frac{1}{2} \right) \mathbb{E}_P \left[ T_1(T) \mathbf{1}_{T_1(T) \leq T/2} + T_1(T) \mathbf{1}_{T_1(T) > T/2} \right]$$

$$\geq T \left( \sigma(\Delta) - \frac{1}{2} \right) - \left( \sigma(\Delta) - \frac{1}{2} \right) \mathbb{E}_P \left[ T_1(T) \mathbf{1}_{T_1(T) \leq T/2} \right] - \left( \sigma(\Delta) - \frac{1}{2} \right) \mathbb{E}_P \left[ T_1(T) \mathbf{1}_{T_1(T) > T/2} \right]$$

$$= T \left( \sigma(\Delta) - \frac{1}{2} \right) - \frac{T}{2} \left( \sigma(\Delta) - \frac{1}{2} \right) \Pr_P(T_1(T) \leq T/2) - T \left( \sigma(\Delta) - \frac{1}{2} \right) \Pr_P(T_1 > T/2)$$

$$= \frac{T \left( \sigma(\Delta) - \frac{1}{2} \right)}{2} \Pr_P (T_1(T) \leq T/2).$$

and

$$\text{Reg}_Q(T) = \sum_{i=1}^K \Delta_i \mathbb{E}_Q[T_i(T)] \geq (\sigma(\Delta) - \frac{1}{2}) \mathbb{E}_Q[T_1(T)] \geq \frac{T \left( \sigma(\Delta) - \frac{1}{2} \right)}{2} \Pr_Q(T_1(T) > T/2).$$

Following the same procedure we obtain:

$$2CT^p \geq \text{Reg}_P(T) + \text{Reg}_Q(T) \geq \frac{T\Delta}{32} \exp(-KL(P, Q)), \tag{41}$$

where

$$KL(P, Q) = \sum_{i \in [K]} \mathbb{E}_P[T_i(T)] \cdot KL(P_i, Q_i) + \sum_{i \leq j} N_{i,j} \cdot KL(P_{i,j}^{\text{off}}, Q_{i,j}^{\text{off}}).$$

For the first term, we have:

$$\sum_{i \in [K]} \mathbb{E}_P[T_i(T)] \cdot KL(P_i, Q_i) = \mathbb{E}_P[T_k(T)] \frac{(\sigma(2\Delta) - 1/2)^2}{2} \leq \frac{\mathbb{E}_P[T_k(T)]\Delta^2}{8}. \tag{42}$$

For the second term, let $N_k$ denotes the number of time we selected $a_k$ in each pair, we have:

$$\sum_{i \leq j} N_{i,j} \cdot \mathrm{KL}(P_{i,j}^{\mathrm{off}}, Q_{i,j}^{\mathrm{off}}) = \sum_{i=2}^{K} N_{k,i} \cdot d(\frac{1}{2}, \max\{\frac{1}{2}, \sigma(2\Delta - \omega_k)\})$$

$$\leq N_k \cdot d(\frac{1}{2}, \max\{\frac{1}{2}, \frac{2\Delta - \omega_k}{4} + \frac{1}{2}\})$$

$$\leq N_k \frac{\max\{2\Delta - \omega_k, 0\}^2}{4 - (\max\{2\Delta - \omega_k)^2, 0\}}$$

$$\leq \frac{1}{3} N_k \max\{2\Delta - \omega_k)^2, 0\}. \tag{43}$$

Putting Equation (42) and Inequality (43) and (41) together, we have:

$$2C^p \geq \mathrm{Reg}_P(T) + \mathrm{Reg}_Q(T) \geq \frac{T\Delta}{32} \exp\left(-\frac{\mathbb{E}_P[T_k(t)]\Delta^2}{8} - \frac{\max\{2\Delta - \omega_k, 0\}^2}{3} N_k\right).$$

reorganize the structure of the above equation following Appendix D.1 leads to the gap independent bound, which is:

$$\frac{8K}{\Delta}\left((1-p)\log T + \log \frac{\Delta}{64C} - \sum_{i \in [K]} \frac{\max\{2\Delta - \omega_i, 0\}^2}{3} N_i\right).$$

as desired. $\qquad\square$

*Gap-Independent Regret Bound.* The proof of the gap-independent lower bound largely follows the argument in Appendix D.2, with the only difference being the switch of the offline and online data. Therefore, we provide only a sketch of the modified proof. In **Case 1**, since the offline and online share the same distributions, the switch does not affect the analysis. In **Case 2** and **Case 3**, where the discrepancy between $P$ and $Q$ appears only in the online distributions, the KL divergence term is updated from the original expression to $\sum_{i=1}^{K} T_i \cdot \mathrm{KL}(P_i, Q_i) = 2T_k \cdot \Delta$, but the conclusion remains unchanged. The only modification lies in the boundary condition for each case, which now compares $2\sqrt{KT}$ with $T \cdot (V_{\max} + 1/\sqrt{T_*'})$. This adjustment arises because, in the online-stochastic setting, we have $\sum_{i=1}^{K} T_i = T$ rather than $2T$. $\qquad\square$

## F  Stochastic Bandits with Offline Relative Feedback: An Alternative Approach

As previously discussed, preference data typically provide less information compared to stochastic feedback. Estimating the underlying utility values $\mu_i$ for all $i \in [K]$ solely from preference data is extremely challenging, unless additional structural assumptions are imposed. For the sake of theoretical completeness, we still adopt a preference-based model in Algorithm 2 and build upon the elimination framework. However, this class of algorithms cannot match the performance of vanilla UCB when offline data is absent (i.e., when the offline dataset is empty) or provides no additional information.

---

**Algorithm 3** Offline Relative Online Stochastic: An Alternative Approach

---

**Require:** an arm set $\mathcal{A}$, offline dataset $\mathcal{D} = \{(A_i, A_j, Y_{i,j,k}), i, j \in [K], k \in [N_{i,j}]\}$ hyperparameter $\delta_t$ and estimated bias $V_{i,j}$ for all $(a_i, a_j)$ pairs.
  **Initialization:** Let $\mathcal{C} = \mathcal{A}$, $T_i = 0, \forall i \in [K]$, and for all $i, j \in [K]$, let $\mathrm{UCB}(a_i, a_j) = +\infty$, $\mathrm{UCB}^{\mathrm{hyb}}(a_i, a_j) = \hat{p}_{i,j}^{\mathrm{off}} + \sqrt{\frac{\log 1/\delta_t}{N_{i,j}}}$.
 1: **for** $t = 1, \cdots, T$ **do**
 2:     Select Action $A(t) = \arg\max_{a_i \in \mathcal{C}} \mathrm{UCB}(a_i)$.
 3:     Update selection times $T_{A(t)} = T_{A(t)} + 1$.
 4:     Record observation $X_{i,k}$ for $i = A(t)$, $k = T_{A(t)}$.
 5:     For all pairs $i, j \in [K]$, update UCB, $\mathrm{UCB}^{\mathrm{hyb}}$ according to equation (9), (10).
 6:     $\mathcal{C} = \mathcal{C} \setminus \{a_i \in \mathcal{C}, \exists a_j \in \mathcal{C} \setminus \{a_i\} \text{ s.t. } \mathrm{UCB}^{\mathrm{hyb}}(a_i, a_j) < \frac{1}{2}\}$.
 7: **end for**

---

In practice, more flexible algorithmic designs can be considered. In Algorithm 3, we present an alternative approach that aligns with vanilla UCB by utilizing UCB and $\mathrm{UCB}^{\mathrm{hyb}}$ separately. Specifically, we construct the

candidate set $\mathcal{C}_t^{\text{hyb}}$ using hybrid data, and then apply the standard UCB selection rule to choose $A(t)$ within this set.

While such variants may perform well empirically, they lack the strong theoretical guarantees provided by Algorithm 2. In particular, these methods no longer ensure balanced exploration across arms during the online phase. As a result, the sample counts $T_i(t)$ for each arm $a_i \in \mathcal{C}_t$ can become highly imbalanced.

Moreover, since vanilla UCB does not guarantee a theoretical lower bound on $\min\{T_i(t)\}$ for all $i \in [K]$, the confidence radius in UCB$^{\text{hyb}}$—given by $\sqrt{1/\left(N_{i,j} + \frac{T_i(t)T_j(t)}{T_i(t)+T_j(t)}\right)}$—can degrade to $\sqrt{1/N_{i,j}}$ in the worst case. Thereby, the theoretical analysis falls back to a naive two-stage procedure: (i) using offline data to eliminate suboptimal arms, followed by (ii) applying vanilla UCB to the surviving set. Under this simplified strategy, the regret bound becomes:

$$\text{Reg}(T) = O\left(\sum_{a_i \in \mathcal{C}} \frac{\log T}{\Delta_i}\right),$$

where $\mathcal{C} \subseteq \mathcal{A}$ denotes the set of arms retained after offline filtering. Compared to Algorithm 1, such two-phase strategies lack a continuous and elegant theoretical guarantee throughout the learning process. Bridging this gap—by designing practical algorithms that maintain strong regret guarantees even under hybrid and heterogeneous feedback—remains an important direction for future work.

# G  Supplemental Experiments

We conduct a comprehensive evaluation of HybUCB-AR and HybElimUCB-RA in hybrid bandit settings with heterogeneous offline-online feedback. The experiments aim to demonstrate the algorithms' robustness and superiority over established baselines across diverse scenarios. Our evaluation encompasses two primary environments: (1) synthetic datasets, where we assess scalability with varying numbers of arms $K$ and sensitivity to key parameters, (2) real-world datasets, where we validate practical efficacy using authentic data. For all experiments, we set a set of random seeds to ensure the reproducibility of the experiments. Our experiments were implemented in Python.

## G.1  Synthetic Data Experiments

To evaluate the performance of HybUCB-AR and HybElimUCB-RA in controlled settings, we conduct experiments on synthetic datasets. In the first set of experiments, we assess the algorithms' scalability by varying the number of arms $K$, comparing their performance against established baselines to demonstrate their effectiveness across different scales. In the second set of experiments, we analyze the sensitivity of HybUCB-AR and HybElimUCB-RA to key parameters ($N_i$, $\Delta$, and $V_i$), focusing on how different parameter values affect their convergence behavior. The synthetic environment allows precise control over reward distributions and feedback mechanisms, enabling robust analysis of algorithmic behavior.

### G.1.1  Data Generation

We generate a synthetic $K$-armed bandit environment to simulate the heterogeneous feedback setting. For each arm $a_i \in \mathcal{A}$, we generate the offline and online reward means, $\mu_i^{\text{off}}$ and $\mu_i$, as follows. The offline mean of the best arm, $\mu_1^{\text{off}}$, is drawn uniformly from $[0.5 + \Delta, 1]$, where $\Delta \in (0, 0.5)$ denotes the gap between $\mu_1^{\text{off}}$ and $\mu_2^{\text{off}}$; we then set $\mu_2^{\text{off}} = \mu_1^{\text{off}} - \Delta$. For the remaining $K - 2$ arms, $\mu_i^{\text{off}}$ is drawn uniformly from $[0, \mu_2^{\text{off}}]$. The online reward mean is defined as

$$\mu_i = \mu_i^{\text{off}} + d_i \cdot \text{bias}, \quad \text{where } d_i \in \{-1, 1\} \text{ is chosen uniformly at random}, \quad \forall i \in [K].$$

All absolute feedback $\{i, X_i\}$ for each $i \in [K]$ is generated from a Gaussian distribution $\mathcal{N}(\mu_i, 1)$, and relative feedback $\{i, j, Y_{i,j}\}$ for all $i, j \in [K]$ is generated from a Bernoulli distribution Bernoulli$(p_{i,j})$ according to the Bradley-Terry model.

### G.1.2  Experiments for HybUCB-AR

HybUCB-AR leverages offline absolute rewards to enhance online dueling bandit learning. We compare it against three baselines: HybUCB-AR without offline data ($N_i = 0$), Relative Upper Confidence Bound (RUCB) (Zoghi et al., 2014), and Interleaved Filter 2 (IF2) (Yue et al., 2012).

**Performance Comparison and Scalability ($K = 8, 16, 24, 32$)  Setup.** We evaluate HybUCB-AR's scalability across $K = 8, 16, 24, 32$ arms. We set the sub-optimal gap $\Delta = 0.1$, the offline-online bias to 0.1, and the number of offline samples $N_i = 500$. Each algorithm runs for $T = 50,000$ rounds with 100

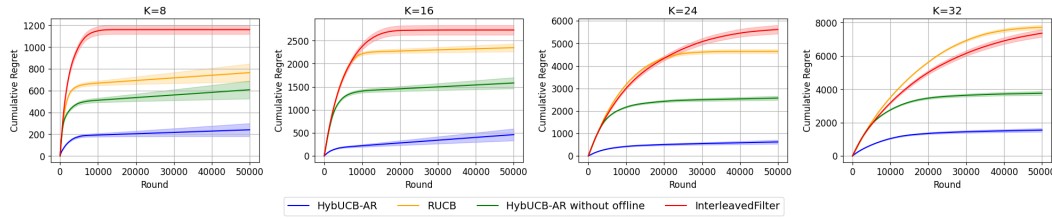

Figure 2: Average cumulative regret of HybUCB-AR for $K = 8, 16, 24, 32$ over 100 runs, with shaded area indicating standard deviation.

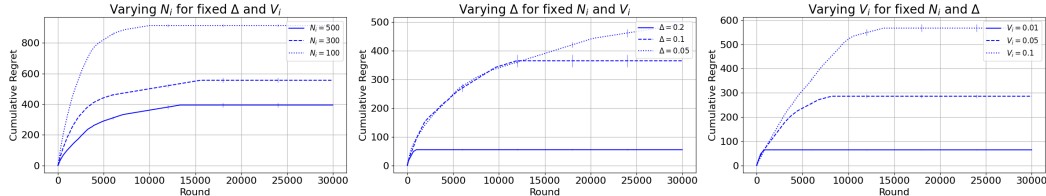

Figure 3: Parameter sensitivity of HybUCB-AR, showing effects of $N_i$, $\Delta$, and $V_i$ on cumulative regret, with vertical lines indicating standard deviation every 6,000 rounds.

independent trials. We set $\delta_t = 0.02$, for RUCB, we set the confidence parameter $\alpha = 0.51$, following Zoghi et al. (2014).

**Results.** Figure 2 shows the average cumulative regret over 100 trials, with shaded area indicating standard deviation. Without offline data, HybUCB-AR, by maximizing the informative pair in the confidence set, achieves a regret reduction of 15–40% compared to RUCB across all $K$. The performance advantage grows with larger $K$, highlighting its scalability. And across all $K$, HybUCB-AR surpass baseline when heterogeneous offline data is included.

**Parameter Sensitivity Analysis**  **Setup.** We analyze HybUCB-AR's sensitivity to three parameters:For all $i \in [K]$, we set the number of offline samples, $N_i \in \{100, 300, 500\}$ the sub-optimal gap $\Delta \in \{0.05, 0.1, 0.2\}$, and the offline-online bias, $V_i \in \{0.01, 0.05, 0.1\}$. We fix $K = 20$, run for 30,000 rounds with 100 trials, and use default values ($N_i = 500$, $\Delta = 0.1$, $V_i = 0$) for all parameters unless otherwise specified.

**Results.** Figure 3 shows the cumulative regret. As $N_i$ increases, the algorithm converges faster, leading to lower cumulative regret. Smaller $\Delta$ increases regret due to harder arm differentiation. Larger $V_i$ reduces the ability to utilize offline data, resulting in slower convergence. One key insight is that the accumulated regret does not decrease linearly with the size of the offline data, while the increase in regret caused by bias shift grows approximately linearly with the magnitude of the bias. This observation is consistent with our theoretical regret analysis in Theorem 1.

### G.1.3  Experiments for HybElimUCB-RA

HybElimUCB-RA uses offline relative preferences to improve online stochastic learning. We compare it against four baselines: HybElimUCB-RA without offline data (with $N_i = 0$), Explore-Then-Commit (ETC) (Robbins, 1952), Upper Confidence Bound (UCB) (Lai and Robbins, 1985), and Thompson Sampling (TS) (Agrawal and Goyal, 2013).

**Performance Comparison and Scalability ($K = 8$, 16, 24, 32)**  **Setup.** We evaluate HybElimUCB-RA's scalability across $K = 8$, 16, 24, 32 arms. The sub-optimal gap $\Delta$ is set to 0.1, the offline-online bias to 0.01, and the number of offline samples to $N_i = 500$. Each algorithm runs for $T = 30,000$ rounds with 100 independent trials. HybElimUCB-RA and UCB use $\delta_t = 0.05$, ETC uses an exploration phase of 500, and TS assumes a Gaussian prior with variance 1.

**Results.** Figure 4 shows the average cumulative regret over 100 trials, with shaded area indicating standard deviation. When leveraging offline relative preferences, HybElimUCB-RA outperforms all baselines. The performance advantage grows with larger $K$, highlighting its scalability. Without offline data, HybElimUCB-RA matches the performance of elimination-based UCB.

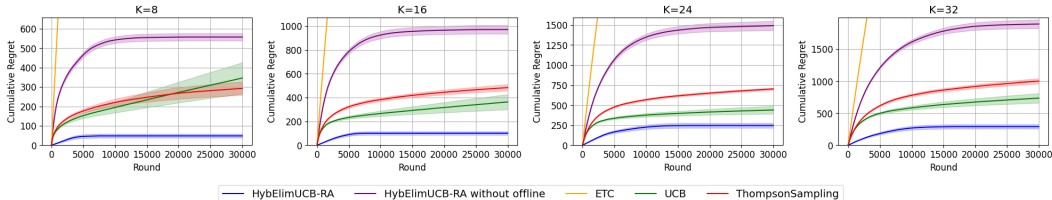

Figure 4: Average cumulative regret of HybElimUCB-RA for $K = 8, 16, 24, 32$ over 100 trials, with shaded area indicating standard deviation.

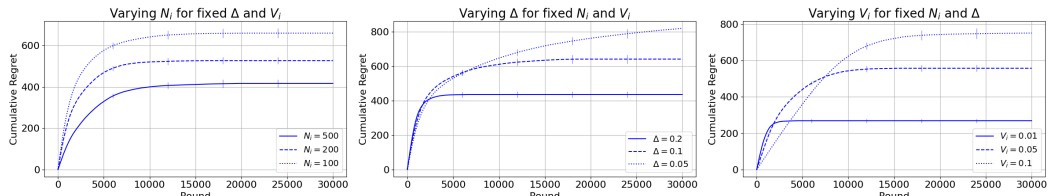

Figure 5: Parameter sensitivity of HybElimUCB-RA, showing effects of $N_i$, $\Delta$, and $V_i$ on cumulative regret, with vertical lines indicating standard deviation every 6,000 rounds.

**Parameter Sensitivity Analysis   Setup.** We analyze HybElimUCB-RA's sensitivity to three parameters: For all $i \in [K]$, we set the number of offline samples $N_i \in \{100, 200, 500\}$, the sub-optimal gap $\Delta \in \{0.05, 0.1, 0.2\}$, and the offline-online bias, $V_i \in \{0.01, 0.05, 0.1\}$. We fix $K = 10$, run for 25,000 rounds with 100 trials, and use default values ($N_i = 100$, $\Delta = 0.1$, $V_i = 0.01$) for all parameters unless otherwise specified.

**Results.** Figure 5 shows the cumulative regret, with vertical lines indicating standard deviation every 6,000 rounds. As $N_i$ increases, the algorithm converges faster, leading to lower cumulative regret. Smaller $\Delta$ increases regret due to harder arm differentiation. Larger $V_i$ reduces the ability to utilize offline data, resulting in slower convergence. This result is similar to the result we obtained in Figure 3, demonstrating the consistent of our algorithm.

## G.2   Real Data Experiments

We evaluate HybUCB-AR and HybElimUCB-RA on MovieLens-20M and Yelp datasets to validate their performance in real-world hybrid bandit settings. Experiments compare both algorithms against established baselines, leveraging offline data to enhance online learning. All experiments are implemented in Python.

### G.2.1   Data Preparation

The **MovieLens-20M** dataset contains 20,000,000 ratings (1–5) for movies. We normalize ratings to [0,1] by dividing by 5, select movies with at least 100 ratings, take the top 100 by rating count, and randomly sample $K = 10$ movies as arms. For HybUCB-AR, offline data consists of 100,000 normalized ratings. For HybElimUCB-RA, offline data includes 1,000 preference duels per arm pair, generated by sampling ten ratings per arm and comparing their means.

The **Yelp** dataset, sourced from the Yelp Academic Dataset, includes business reviews with star ratings (1–5). We normalize ratings to [0, 1], select businesses with at least 100 ratings, take the top 100 by rating count, and randomly sample $K = 10$ businesses. Offline data follows the same structure as MovieLens.

Online feedback is generated using the environment module designed by us. For HybUCB-AR, duels sample 3 ratings per arm, with the higher average rating winning (ties resolved randomly). For HybElimUCB-RA, rewards are the average of 30 sampled ratings per arm.

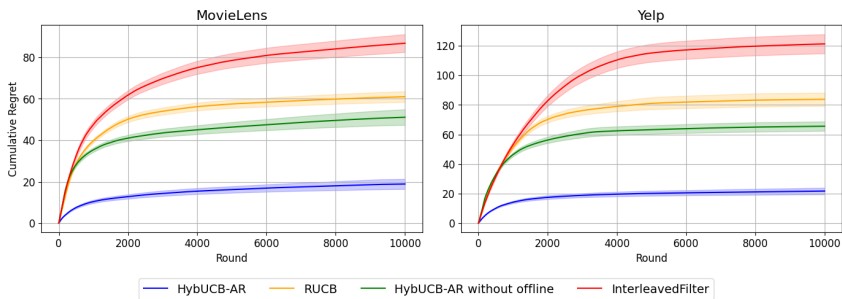

Figure 6: Performance of HybUCB-AR on MovieLens 20M ($K = 10$), with shaded areas showing standard deviation across 50 trials.

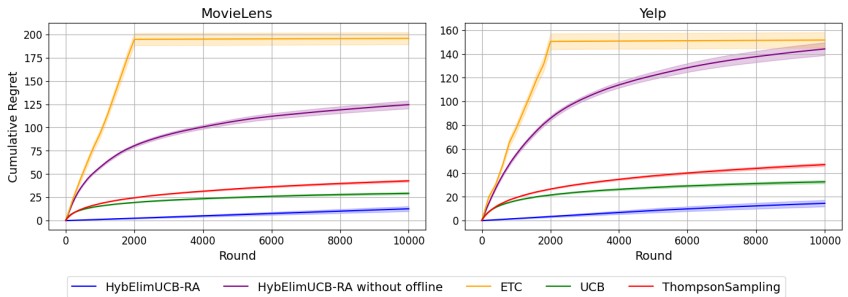

Figure 7: Performance of HybElimUCB-RA on MovieLens-20M and Yelp with ($K = 10$). Shaded areas showing standard deviation across 50 runs.

### G.2.2 Experiments for HybUCB-AR

HybUCB-AR leverages offline absolute rewards to improve online dueling bandit learning, using a V-matrix defined as $V = (V_{i,j})_{i,j \in [K]}$. We compare it against HybUCB-AR without offline data ($N_i = 0$), RUCB (Zoghi et al., 2014), and Interleaved Filter 2 (IF2) (Yue et al., 2012).

**Performance Comparison  Setup.** We set $K = 10$, run each algorithm for 10,000 rounds over 50 trials. The environment module samples 3 ratings per arm for duels. We set $\delta_t = 0.02$. Offline data comprises 100,000 normalized ratings.

**Results.** Figures 6 show the average cumulative regret, with shaded areas indicating standard deviation. Without offline data, HybUCB-AR outperforms RUCB by 15–25% on both datasets. HybUCB-AR shows the same good performance as in the synthetic experiments.

### G.2.3 Experiments for HybElimUCB-RA

HybElimUCB-RA uses offline relative preferences to enhance online absolute reward learning, employing the same V-matrix as HybUCB-AR. We compare it against HybElimUCB-RA without offline data ($N_i = 0$), Explore-Then-Commit (ETC) (Robbins, 1952), Upper Confidence Bound (UCB) (Lai and Robbins, 1985), and Thompson Sampling (TS) (Agrawal and Goyal, 2013).

**Performance Comparison  Setup.** We set $K = 10$, run each algorithm for 10,000 total arm pulls over 50 trials. The environment module samples 30 ratings per arm to compute rewards. HybElimUCB-RA and UCB use $\delta_t = 0.05$, ETC uses an exploration phase of 200 pulls per arm, and TS assumes a Gaussian prior with mean 0.5 and variance 1. Offline data consists of 1,000 relative preference duels per arm pair.

**Results.** Figure 7 shows the average cumulative regret. In two real data environments, HybElimUCB-RA combined with offline data can achieve good results and is better than the baseline.

# H  Useful Lemmas

**Lemma 8** (Chernoff Inequality, Theorem 5.3 in Lattimore and Szepesvári (2020)). *If $X$ is $\sigma$ sub-Gaussian, then for any $\epsilon > 0$,*

$$\Pr\left(X \geq \varepsilon\right) \leq \exp\left(-\frac{\varepsilon^2}{2\sigma^2}\right).$$

**Lemma 9** (Hoeffding's Lemma). *Let $X$ be any real-valued random variable such that $a \leq X \leq b$ almost surely, i.e., with probability one. Then, for all $\lambda \in \mathbb{R}$,*

$$\mathbb{E}\left[e^{\lambda X}\right] \leq \exp\left(\lambda\mathbb{E}[X] + \frac{\lambda^2(b-a)^2}{8}\right),$$

*or equivalently,*

$$\mathbb{E}\left[e^{\lambda(X-\mathbb{E}[X])}\right] \leq \exp\left(\frac{\lambda^2(b-a)^2}{8}\right).$$

**Lemma 10** (sub-Gaussian Property of Lipschitz Transformations). *Let $X$ be a random variable that is $\sigma$ sub-Gaussian, meaning $\mathbb{E}[e^{\lambda X}] \leq e^{\lambda^2\sigma^2/2}$ for all $\lambda \in \mathbb{R}$. If $f : \mathbb{R} \to \mathbb{R}$ is $L$-Lipschitz continuous, then $f(X)$ is $2L\sigma$ sub-Gaussian.*

*Proof.* A related result appears in Theorem 5.5 of Boucheron et al. (2013), which shows that $f(X)$ is $L\sigma$ sub-Gaussian when $X$ is strictly Gaussian. For a general $\sigma$ sub-Gaussian $X$, we establish that $f(X)$ is $2L\sigma$ sub-Gaussian.

Let $Y$ be an independent copy of $X$. We aim to bound the moment-generating function of $(f(X) - \mathbb{E}[f(X)])^2$. Consider a constant $c > 0$:

$$
\begin{aligned}
\mathbb{E}\left[\exp\left(\frac{(f(X) - \mathbb{E}[f(X)])^2}{c^2}\right)\right] &= \mathbb{E}\left[\exp\left(\frac{(f(X) - \mathbb{E}[f(Y)])^2}{c^2}\right)\right] \\
&= \int_{\mathbb{R}} P(X \in dx)\exp\left(\frac{(f(x) - \mathbb{E}[f(Y)])^2}{c^2}\right) \\
&\leq \int_{\mathbb{R}} P(X \in dx)\mathbb{E}\left[\exp\left(\frac{(f(x) - f(Y))^2}{c^2}\right)\right] \\
&= \mathbb{E}\left[\exp\left(\frac{(f(X) - f(Y))^2}{c^2}\right)\right] \\
&\leq \mathbb{E}\left[\exp\left(\frac{L^2(X - Y)^2}{c^2}\right)\right] \\
&\leq \mathbb{E}\left[\exp\left(\frac{2L^2X^2 + 2L^2Y^2}{c^2}\right)\right] \\
&= \left(\mathbb{E}\left[\exp\left(\frac{2L^2X^2}{c^2}\right)\right]\right)^2 \\
&\leq \mathbb{E}\left[\exp\left(\frac{4L^2X^2}{c^2}\right)\right] \leq 2.
\end{aligned}
$$

Here, the second equality uses the law of total expectation; the first inequality applies Jensen's inequality; the second inequality uses the $L$-Lipschitz property, $|f(x) - f(y)| \leq L|x - y|$; the third inequality uses $(X - Y)^2 \leq 2X^2 + 2Y^2$; and the final steps exploit the independence of $X$ and $Y$. The last inequality holds because $X$ is $\sigma$ sub-Gaussian, so $\mathbb{E}[e^{X^2/\sigma^2}] \leq 2$. Setting $4L^2/c^2 = 1/\sigma^2$, we get $c = 2L\sigma$. Thus, $f(X)$ is $2L\sigma$ sub-Gaussian, as desired. $\qquad\square$

**Lemma 11** (MVUE). *Let $\hat{X}_1, \hat{X}_2$ be 2 independent estimator of $\mu$, with variance $\sigma_1^2$ and $\sigma_2^2$, then the minimum variance unbiased estimator of $X$ is:*

$$\hat{X} = \frac{\sigma_2^2}{\sigma_1^2 + \sigma_2^2}\hat{X}_1 + \frac{\sigma_1^2}{\sigma_1^2 + \sigma_2^2}\hat{X}_2,$$

*with variance $\frac{\sigma_1^2\sigma_2^2}{\sigma_1^2 + \sigma_2^2}$.*

