# OpenReview forum: "Learning Across the Gap: Hybrid Multi-armed Bandits with Heterogeneous Offline and Online Data"
_NeurIPS.cc/2025/Conference — NeurIPS 2025 poster_

### Official Review · Reviewer_5U6k · 2025-06-04

**Clarity:** 3
**Significance:** 3
**Originality:** 3
**Rating:** 5
**Confidence:** 3

**Summary:**

This paper studies hybrid stochastic MAB. That is, warm starting the online interaction stage with offline dataset. this paper considers that the offline and online feedback are of different types, one is absolute signal and the other is pairwise comparison. The main challenge is how to fuse these two types of feedback

In both settings, there are algorithms proposed that leverege offline dataset whenever it is helpful, and does not suffer from it otherwise. Lower bounds are also provided. Lastly, experiments validate the effectiveness of the proposed algorithms.

**Questions:**

1. Does the offline data never hurt the online regret, even if they are not helpful at all?
2. "closely match" in line 279 is not precise enough. I expect more discussion on the gap between upper and lower bounds
3. Now two settings have different types of alg. design (UCB vs. elimination). What's the challenge to unify them?

typos

1. $\\mathcal{C}_t$ in Eq. (5) should be $\\mathcal{C}^{\\mathrm{on}}_t$?

**Ethical Concerns:**

["NO or VERY MINOR ethics concerns only"]

**Final Justification:**

my final recommendation would be 5: Accept. The studied problem is highly important, and technically I don't see major weakness.

**Limitations:**

yes

**Quality:**

4

**Strengths And Weaknesses:**

Strengths

1. It's a well-motivated problem due to RLHF
2. Algorithm design is not obvious or trivial
3. Remarks which interpret the results are very informative and sharp
4. Lower bound results complement upper bound results.

Weaknesses

1. The performance depends on hyper-parameter tuning, which further relies on some unknown knowledge.

---

> ### Author Rebuttal · Authors · 2025-07-30
>
> # Response to Reviewer 5U6k
>
> We thank the reviewer 5U6k for the valuable comments and suggestions. Please find our detailed response below.
>
> ## 1. Hyper-parameter tuning
>
>
> Our algorithm involves only two hyper-parameters: the confidence level and the bias bound $V$. The confidence level can be set using standard theoretical analysis, leaving $V$ as the only parameter requiring prior knowledge. We thus believe the reviewer is primarily referring to $V$.
>
> We agree that in the off2online learning with distribution bias setting, it would be desirable to eliminate the need for such a parameter. However, as rigorously shown in Section 3 of [1], no algorithm can outperform—or even match—the performance of a purely online approach without any prior knowledge of the bias. This lower bound fundamentally highlights the necessity of incorporating some prior information in order to leverage offline data effectively.
>
> Although our hyper-parameter $V$ encodes prior knowledge about the bias, it does not need to be exact—any valid upper bound suffices. When the learner has sufficient knowledge about the offline and online environments, the bound can be tight, allowing our algorithm to effectively leverage offline data for improved performance. In contrast, when the prior is loose or even unavailable, the algorithm gracefully degrades to the performance of vanilla UCB, thereby ensuring robustness.
>
> We fully acknowledge that further reducing reliance on this parameter is a valuable and promising research direction, and we leave this as an open problem for future work.
>
> [1] Cheung, W. C., & Lyu, L. (2024). Leveraging (Biased) Information: Multi-armed Bandits with Offline Data. In International Conference on Machine Learning.
>
> ## 2. Response to questions
>
> ### 2.1 Offline data does not hurt online regret
>
> Note that our algorithm incorporates a valid bias bound as an input. When the bias is either uncertain (i.e., not helpful) or substantially large (i.e., exceeding practical limits), we could set this valid bias bound to infinity. Consequently, due to the minimum UCB criterion implemented in our algorithm, both HybUCB-AR and HybElimUCB-RA will at least retain the performance of their pure online exploration form.
>
> ### 2.2 Discussion about upper and lower bounds
>
>
> We compare the upper and lower bounds separately for the instance-independent and instance-dependent cases.
>
> For the instance-independent bounds, we first compare the first term in the upper bound (Theorem 1(b)) and the lower bound (Theorem 2(b)). Specifically, the first term in the upper bound is $O(\sqrt{K^2 T \log T})$, while the corresponding term in the lower bound is $\Omega(\sqrt{KT})$. This implies a gap of $\sqrt{K \log T}$ between the upper and lower bounds for this term. We then compare the second terms. The second term in the upper bound can be written as $O((\sqrt{{\log T}/{\tau_*}} + V_{\max}) \cdot T)$, and the corresponding term in the lower bound is $\Omega( (\sqrt{{1}/{\tau_*'}} + V_{\max})\cdot T)$. To compare the different parameter $\tau_*$ and $\tau_*'$, we consider a special case where $N_i = n$ for all $i \in [K]$. In this case,
> $$
> \tau_* = \frac{n}{2} + \frac{2T}{K(K+1)}, \quad \tau_*' = n + \frac{2T}{K}.
> $$
> Under this setting, it can be shown that the gap between the upper and lower bounds for the second term is again at most a factor of $\sqrt{K \log T}$.
>
> For the instance-dependent case, to better compare the upper and lower bounds, we consider the special case where $N_i=n$ for all $i\in[K]$ and $\Delta_i = \Delta$ for all sub-optimal arms. Then the upper bound in Theorem 2(a) simplifies to
> $$
> O\left(\frac{K^2\log T}{\Delta}-\frac{K^2n}{4}\sum_{i\leq j}\max\\{\Delta-4\omega_{i,j},0\\} \right),
> $$
>  and the lower bound is in the form of:
> $$
> \Omega\left(\frac{K\log T}{\Delta}+\frac{K}{4\Delta}\log\frac{\Delta}{64C}-Kn\sum_{i\in[K]}{\max\\{\Delta-\omega_i/2,0\\}^2}\right).
> $$
> From this comparison, we observe that the upper bound is approximately a factor of $K$ worse than the lower bound in this case. We will include a discussion on the tightness of the bounds in the next version of the paper.
>
> ### 2.3 Challenges in unifying different types of algorithm
>
> Thanks for this good question. We invested significant effort in attempting to unify the two algorithms, particularly aiming to make the second setting (relative to absolute) align with vanilla UCB when offline data is unavailable. Before we formed HybElimUCB-RA, two ways have been proposed:
>
> - For HybUCB-AR (offline absolute with online relative feedback), we simultaneously utilize both offline and online data to maintain estimates $p_{i,j}^{\text{hyb}}$ for all $i, j \in [K]$. In contrast, when adapting this to the online absolute feedback setting, we need to maintain estimates $\mu_i$ instead. Here, a key challenge emerges: the offline relative feedback data, due to its pairwise nature, lacks absolute value information, making it intrinsically challenging to employ for maintaining absolute value estimates (Especially under the Condorcet winner assumption).
>
> - An alternative approach involves employing a hybrid method, which uses offline and online data to estimate $p_{i,j}^{\text{hyb}}$ and pure online feedback to estimate $\mu_i$. This algorithm can be divided into three parts:
>
>   **For** each time step $t$, **do**
>
>   1. Construct candidate set $C_t$ based on the hybrid estimator $p_{i,j}^{\text{hyb}}$ and its confidence radius.
>   2. Select and play an arm $a$ such that $a = \arg\max_{i \in [K]} \text{UCB}(a_i)$ (vanilla UCB).
>   3. Update the statistics $\mu_i$, $p_{i,j}^{\text{hyb}}$ and their upper confidence bound for all $i, j \in [K]$ based on the observed feedback.
>
>   This algorithm is detailed in Appendix F.
>
>   From a practical viewpoint, this algorithm is expected to perform well since: (1) it matches vanilla UCB in the worst case, and (2) the hybrid estimator utilizes offline and online data jointly in the elimination process. However, note that the upper confidence bound for $p_{i,j}^{\text{hyb}}$ is given by
>    $$\hat{p}\_{i,j}^{\text{hyb}} + \sqrt{\frac{\log(1/\delta\_t)}{2\left(\frac{T\_i T\_j}{T\_i + T\_j} + N\_{i,j}\right)}} + \frac{N\_{i,j}}{\frac{T\_i T\_j}{T\_i + T\_j} + N_{i,j}} V\_{i,j}$$
>   (Equation 10 in the paper), which is directly influenced by $\frac{T_i T_j}{T_i + T_j}$. Since vanilla UCB does not guarantee the minimum selection times for each arm, in the worst case—due to insufficient exploration for some arms $i, j \in [K]$, $\frac{T_i T_j}{T_i + T_j} = 0$—the theoretical analysis degenerates into a two-stage algorithm:
>
>   1. Eliminate some arms based on the offline dataset.
>   2. Perform vanilla UCB.
>
>   This approach lacks the robust theoretical guarantees provided by HybElimUCB-RA.
>
> Finally, we developed HybElimUCB-RA. Unifying the use of offline relative data for absolute online feedback remains an important and promising direction for future research.
>
> ## 3. Typos
>
> Yes, this should be a typo. Thanks for pointing it out!

---

> > ### Comment · Reviewer_5U6k · 2025-08-05
> >
> > I thank authors for the response. While the hyper-parameter tuning issue is a bit unpleasing, I still appreciate the importance and novelty of this paper and would like to recommend for acceptance.

---

> > > ### Author Response · Authors · 2025-08-07
> > >
> > > Thank you for your positive feedback and for appreciating the importance and novelty of our work. We sincerely appreciate your support and recommendation for acceptance.

---

### Official Review · Reviewer_qfCy · 2025-06-06

**Clarity:** 2
**Significance:** 3
**Originality:** 3
**Rating:** 5
**Confidence:** 2

**Summary:**

The paper analyzes the hybrid multi-armed bandit setting, where the learner has access to an offline dataset before interacting with an online learning environment. Interestingly, this setting considers the heterogeneous across the offline versus online: one is the preference relative data, the other is the typical reward data.

Even though Agnihotri et al., 2024 have studied a similar setting previously, in this paper, the authors also analyze the distribution shift problem, where the offline dataset may not be informative to the online phase. I think that the warm-start setting with heterogeneous data and distribution shift is a very interesting and practical problem.

The authors provide the instance-dependent and instance-independent bounds for both the lower bound and the upper bound of their proposed algorithms, for both of their settings. They also provided an experiment section to demonstrate the effectiveness of their algorithms.

**Questions:**

Already discussed above.

**Ethical Concerns:**

["NO or VERY MINOR ethics concerns only"]

**Final Justification:**

I will maintain my positive score. I wish you the best of luck with the final results.

**Limitations:**

The author should discuss the limitations clearly in the paper.

**Quality:**

3

**Strengths And Weaknesses:**

Quality: The submission is technically sound. The theorem's results make sense to me, although I haven't checked all the proofs.
The idea of using pessimism to address mis-specification (or distribution shift in this case) is valid and well-known on a high level, but its application to construct the confidence set in this paper is novel.

Clarity: The submission is clearly written and organized. Still, it would help a lot if the authors gave a high-level and intuitive explanation for their algorithms and analysis. For example, I still don't have a good high-level understanding of the exploration term in equations 4 and 10, as well as the significance of the terms in the upper and lower bounds.

Another suggestion on clarity is on the Remarks. I think they are all important discussions, but they may be written in a better way. For example, Theorem 1b bounds include the term Saving(a_i, a_j). I understand that keeping it like this helps generalize different amounts of saving, which is problem-dependent, but it makes the reader have a harder time evaluating whether the saving can make a difference to the bound or not. Instead of making a remark like the one in the paper, maybe you can just add a corollary that, assuming x and y, the upper bound is z.

Significance and Originality: This paper analyzes a practical setting, with novel analysis and algorithms. I think there's a significant contribution from this compared to the previous work. I think the related work section shows good and relevant previous and related work.

Notes:
- I'm not sure if the instance-dependent upper-bound and lower-bound in Theorems 1 and 2 "closely match" or are tight.
- I can't comment on the correctness of the proof of the significance (impact) of the bounds (about how much this paper improves on the previous results)

---

> ### Author Rebuttal · Authors · 2025-07-30
>
> # Response to Reviewer qfCy
>
> We thank the reviewer qfCy for the valuable comments and suggestions. Please find our detailed response below.
>
> ## 1. High level intuition
>
> ### 1.1 Intuition of the algorithm design
>
> The purpose of our algorithm is to effectively utilize offline heterogeneous data to enhance online multi-armed bandit (MAB) learning by extending the UCB framework. Intuitively, when there is no bias and the data is homogeneous, it is straightforward to directly use offline data to estimate $\mu_i$ or $p_{i,j}$, for all $i, j\in [K]$. However, under our setting, two questions would arise:
>
> 1. How to construct a unified statistic and an upper confidence bound for heterogeneous data?
> 2. How to deal with the potential bias in the offline data?
>
> To address the first challenge, we construct the unified statistic $p_{i,j}^{\text{hyb}}$ using the Bradley-Terry model and the sub-Gaussian property of the data (Lemma 1, Appendix C.1). The detailed construction of $\text{UCB}^{\text{hyb}}$ (Equations 4 and 10) is based on rigorous theoretical analysis, which is presented in the next section.
>
> For the second challenge, we introduce a valid bias bound that confines the distributional shift within a predefined region. Since the precise bias is unknown, our approach is to conservatively incorporate the bias term into the unified statistic, forming $ \text{UCB}^{\text{hyb}} $. However, naively applying $ \text{UCB}^{\text{hyb}} $ would yield suboptimal results—for some arms, the bound may be loose and it lacks information about the shift direction. To address this, we adopt a "minimum UCB criterion": When $ \text{UCB}^{\text{hyb}} < \text{UCB} $, offline data helps reduce online exploration; conversely, when $ \text{UCB}^{\text{hyb}} \geq \text{UCB} $, offline data provides no benefit, and we revert to the vanilla upper confidence bound.
>
> ### 1.2 Intuition of the algorithm analysis
>
> Similar to existing UCB algorithms, our theoretical analysis is mainly derived from two steps:
>
> 1. Define a good event, such that the real value of the UCB statistic lies within the estimated confidence region with high probability (e.g. $1-\delta$).
> 2. Under the good event, for a given time step $T$, find an upper bound on the number of times each arm (or arm pair) can be selected.
>
>
> To address the first issue, we introduce Equations (4) and (10). Given the presence of bias and heterogeneous data, their mathematical formulation is complex, as shown in the equations. The design of these equations is to ensure the probability of the good event (Equation (12) in our paper), with details provided in Lemma 3 (Appendix C.3).
>
> The intuition behind the second step stems from the following key observation: as sample sizes increase, confidence radius  contract. Once these radius  become sufficiently small, the UCB of any suboptimal arm will satisfy:
> $$
> \mathrm{UCB}(a_i)<\mu_1<\mathrm{UCB}(a_1),
> $$
> This inequality guarantees that suboptimal arms will no longer satisfy the algorithm's selection criterion. We formalize this intuition in Lemma 4 and Lemma 5 (Appendix C.4–C.5).
>
> Finally, the regret upper bound is derived by combining the regret under "bad" event (with low probability, bounded by a constant) and the sum of each arm's regret under good event, computed as the product of its selection frequency and $\Delta_i$.
>
> ## 2. Clarifying the "Saving" term with a corollary
>
> For Theorem 1(b), the Saving term follows the same formulation as in Theorem 1(a). The transition from an instance-dependent bound to an instance-independent bound involves purely mathematical transformations. The explicitly mathematical expression of the Saving term is:
> $$
> \text{Saving}(a_i,a_j)=\frac{\Delta_i+\Delta_j}{2}\cdot\frac{N_iN_j}{N_i+N_j}\cdot\frac{\max\\{\max\\{\Delta_i,\Delta_j\\}-4\omega_{i,j},0\\}}{\max\\{\Delta_i,\Delta_j\\}}.
> $$
> Regarding Remark 2, as you pointed out, since the "Saving" term is rather complex, providing a special case along with its regret upper bound could help readers better understand the effectiveness of offline data. Building on the discussed approach, we present the following corollary:
>
> **Corollary**: When $V_{i,j} = 0$ and offline data is uniformly distributed (i.e., $N_i = n$ for all $i \in [K]$), the regret upper bound reduces to:
> $$
> O\left(\sum_{i\leq j}\frac{\Delta_i+\Delta_j}{2}\left[\max \left\\{\frac{\log T}{\max\\{\Delta_i^2,\Delta_j^2\\}}-\frac{n}{2},0\right\\}\right]\right),$$
> and
> $$
> O\left(\sqrt{\frac{2K(K+1)T^2\log T}{nK(K+1)+4T}}\right)
> $$
> respectively.
> Under this setting, the contribution of offline data to regret reduction is captured by the terms $-\frac{n}{2}$ and $nK(K+1)$, respectively. As the amount of offline data becomes sufficiently large, the regret upper bound approaches a constant.
>
> This presentation will be refined in the next version of the paper.
>
>
>
> ## 3. Discussion about upper and lower bounds
>
> We compare the upper and lower bounds separately for the instance-independent and instance-dependent cases.
>
> For the instance-independent bounds, we first compare the first term in the upper bound (Theorem 1(b)) and the lower bound (Theorem 2(b)). Specifically, the first term in the upper bound is $O(\sqrt{K^2 T \log T})$, while the corresponding term in the lower bound is $\Omega(\sqrt{KT})$. This implies a gap of $\sqrt{K \log T}$ between the upper and lower bounds for this term. We then compare the second terms. The second term in the upper bound can be written as $O((\sqrt{{\log T}/{\tau_*}} + V_{\max}) \cdot T)$, and the corresponding term in the lower bound is $\Omega( (\sqrt{{1}/{\tau_*'}} + V_{\max})\cdot T)$. To compare the different parameter $\tau_*$ and $\tau_*'$, we consider a special case where $N_i = n$ for all $i \in [K]$. In this case,
> $$
> \tau_* = \frac{n}{2} + \frac{2T}{K(K+1)}, \quad \tau_*' = n + \frac{2T}{K}.
> $$
> Under this setting, it can be shown that the gap between the upper and lower bounds for the second term is again at most a factor of $\sqrt{K \log T}$.
>
> For the instance-dependent case, to better compare the upper and lower bounds, we consider the special case where $N_i=n$ for all $i\in[K]$ and $\Delta_i = \Delta$ for all sub-optimal arms. Then the upper bound in Theorem 2(a) simplifies to
> $$
> O\left(\frac{K^2\log T}{\Delta}-\frac{K^2n}{4}\sum_{i\leq j}\max\\{\Delta-4\omega_{i,j},0\\} \right),
> $$
>  and the lower bound is in the form of:
> $$
> \Omega\left(\frac{K\log T}{\Delta}+\frac{K}{4\Delta}\log\frac{\Delta}{64C}-Kn\sum_{i\in[K]}{\max\\{\Delta-\omega_i/2,0\\}^2}\right).
> $$
> From this comparison, we observe that the upper bound is approximately a factor of $K$ worse than the lower bound in this case. We will include a discussion on the tightness of the bounds in the next version of the paper.

---

### Official Review · Reviewer_Q1f6 · 2025-07-02

**Clarity:** 3
**Significance:** 4
**Originality:** 4
**Rating:** 5
**Confidence:** 3

**Summary:**

This paper studies how users can utilize two different types of heterogeneous data that arise within a single game. In particular, motivated by the recent emergence of RLHF, it investigates how two commonly encountered observation types—preference-based observations and conventional direct bandit rewards—can coexist and complement each other. The paper further explores how such offline data can be mutually beneficial. Moreover, it presents the results in a highly general setting that allows for heterogeneity not only in observation types but also in the underlying data distributions.

**Questions:**

Please check the Strength & Weakness section above.

**Ethical Concerns:**

["NO or VERY MINOR ethics concerns only"]

**Final Justification:**

The authors have responded sincerely to my questions, and I still believe that the paper’s value—providing a general approach to combining two different types of data—is sufficient. Based on the other reviewers’ comments, it appears they share a similar impression. Therefore, I have decided to maintain my current score.

**Limitations:**

No.

**Paper Formatting Concerns:**

No isssues found.

**Quality:**

3

**Strengths And Weaknesses:**

This is a personal opinion as a reviewer, but I found this paper to propose an interesting approach to fusing two types of data that, while intuitively complementary, are not straightforward to combine. The significance and novelty of the topic addressed in this work are, at least from my personal perspective, quite clear.

Moreover, the paper is formulated in a very general framework that can accommodate heterogeneous cases that are likely to arise in practice. By employing a "minimum UCB" approach, the authors propose a mechanism that naturally discards offline data when its quality is not sufficiently helpful for online learning. I found this to be a particularly interesting idea.

Regarding **soundness**, I unfortunately was not able to carefully verify all the proofs. However, assuming the results are correct, the fact that the authors provide a matching lower bound (Theorem 2) to their upper bound strongly supports the optimality of their results. The logical flow and the implications drawn by the authors seem reasonable to me, and while my confidence is not high due to limited verification, I intuitively believe there is no major issue with the soundness. I plan to examine the validity of the proofs more closely during the rebuttal period.

---

**Weaknesses**:

A few points remain unclear:

1. The authors appear to have made efforts to clearly delineate their contributions. However, the results themselves depend on many variables and are presented in quite a complex form (e.g., involving a prior upper bound referred to as $V$, or a quantity represented as $\tau^*$, which arises from some complicated optimization). While it is clear that the "saving" offers some degree of benefit, it is difficult to assess how impactful or innovative this actually is. It would be helpful if the authors could provide a carefully chosen, perhaps even "cherry-picked," example that most dramatically illustrates the benefit—preferably in an environment where offline data can dramatically help, such as a homogeneous setting where greedy algorithms suffice online, and then quantify how many offline samples are needed to guarantee such behavior.

2. Remark 4 is difficult to understand. Is the author claiming that there are situations in which preference-based feedback is more useful than direct reward observations? While I appreciate the authors' honesty in noting this, I think the explanation—that it is due to subgaussianity and the BT model—could be expanded. A mathematical example or deeper explanation of why preference feedback is more helpful in such cases would be much appreciated.

---

> ### Author Rebuttal · Authors · 2025-07-30
>
> # Response to Reviewer Q1f6
>
> We thank the reviewer Q1f6 for the valuable comments and suggestions. Please find our detailed response below.
>
> ## 1. Example Demonstrating Offline Data Effectiveness
>
> In Remark 3, we briefly discuss the conditions under which offline data becomes beneficial, with a comprehensive analysis deferred to Appendix C.6. Here, we present a special case under which the regret upper bound simplifies significantly, thereby helping readers better understand the effect of offline data.
>
> Consider the case where $V_{i,j} = 0$ and offline data is uniformly distributed (i.e., $N_i = n$ for all $i \in [K]$). Under this scenario, the regret upper bound simplifies to:
> $$
> O\left( \sum_{i \leq j} \frac{\Delta_i + \Delta_j}{2} \left[ \max \left\\{ \frac{16\operatorname{log}(\sqrt{2K(K+1)}T)}{\max\\{\Delta_i^2, \Delta_j^2\\}} - \frac{n}{2}, 0 \right\\} \right] \right)
> $$
>
>
> and
>
> $$
> O\left( \sqrt{\frac{2K(K+1)T^2 \log T}{nK(K+1) + 4T}} \right),
> $$
> respectively. For the instance-dependent bound, the effectiveness of offline data is captured by the $-\frac{n}{2}$ term, while for the instance-independent bound, it is reflected in the $nK(K+1)$ term. Both bounds demonstrate that regret approaches zero (a constant) as the amount of offline data grows sufficiently large. To determine when online exploration is unnecessary for arm pair $i, j \in [K]$, we compute:
>
> \begin{align*}
> \frac{16\log (\sqrt{2K(K+1)}T)}{\max\\{\Delta_i^2,\Delta_j^2\\}} - \frac{n}{2} &\leq 0, \\\\
> n &\geq \frac{32\log (\sqrt{2K(K+1)}T)}{\max\\{\Delta_i^2,\Delta_j^2\\}} .
> \end{align*}
> We will put these discussions in the next version for better clarity.
>
> **Note:** To help better understand how the savings depend on key parameters, we provide experimental illustrations in Appendix G (Figures 3 and 5), showing the effects of $V_{i,j}$, $\Delta_i$, and $N_i$ on regret convergence rate.
>
> ## 2. Comparisons with homogeneous offline data
>
> Remark 4 compares the utility of homogeneous (relative feedback) versus heterogeneous (absolute feedback) offline data in the HybUCB-AR algorithm. The logical flow proceeds as follows: (1) First we directly compare the mathematical form of the confidence radius, which suggests that relative feedback appears more efficient. (2) Further analysis of the derived result reveals that the preliminary conclusion lacks robustness (due to model assumptions). (3) Finally, we clarify absolute data is generally more effective. Here, we present our reasoning process in detail.
>
> To compare the effectiveness of samples, a key metric for evaluating data utility is the convergence rate of the confidence radius with respect to the number of samples. As shown in Equation (4) in our paper, when the bias is zero, the confidence radius for heterogeneous offline data is:
>
> \begin{align*}
> \sqrt{\frac{1}{2} \cdot \frac{\log(1/\delta_t)}{T_{i,j} + \frac{N_i N_j}{N_i + N_j}}}
> \end{align*}
>
> For homogeneous offline data (relative feedback), the confidence radius (While the paper focuses on the heterogeneous case, the homogeneous version can be derived analogously) is:
>
> \begin{align*}
> \sqrt{\frac{1}{2} \cdot \frac{\log(1/\delta_t)}{T_{i,j} + N_{i,j}}}.
> \end{align*}
>
> Consider the homogeneous case where $N_{i,j} = n$. To achieve an equivalent or tighter confidence radius using heterogeneous offline data, we solve:
>
> \begin{align*}
> \min \quad & N_i + N_j \\\\
> \text{s.t.} \quad & \frac{N_i N_j}{N_i + N_j} \geq n, \\quad N_i, N_j\geq 0
> \end{align*}
>
> This yields $N_i = N_j = 2n$, indicating that absolute feedback requires at least twice as many samples as relative feedback to achieve the same confidence radius. This result suggests that relative feedback is more efficient in this context, which may seem counterintuitive.
>
> At first glance, this appears anomalous since absolute feedback typically provides more information per sample. However, upon closer examination, we find this phenomenon is fundamentally tied to the Bradley-Terry model and our sub-Gaussian assumptions:
>
> - **Relative feedback** (modeled as Bernoulli comparisons) is inherently 1/2-sub-Gaussian (Lemma 1.1, Appendix C.1).
> - **Absolute feedback**, while initially assumed to be 1-sub-Gaussian, becomes 1/2-sub-Gaussian after transformation through the Bradley-Terry model (Lemma 10, Appendix G).
>
> Crucially, these results are model-dependent: Different modeling assumptions (e.g., alternative link functions or sub-Gaussian parameters) would alter the weighting between $ T_{i,j} $ and $ \frac{N_i N_j}{N_i + N_j} $ in Equation (\*). Thus, data utility depends not only on the data type (absolute vs. relative), but also on the underlying statistical assumptions.
>
> On the other hand (from the "In practice" sentence in Remark 4), absolute feedback inherently contains more information and thus offers greater practical utility. For example:
>
> - **Relative feedback** ($N_{i,j}$) only provide information about the relative performance between arms $i$ and $j$.
> - **Absolute feedback** ($N_i$) enables joint estimation of multiple probabilities ($p_{i,j}, p_{i,k}, \dots$) by combining data across all arms ($N_j, N_k, \dots$).
>
>
>
> This fundamental difference explains why absolute feedback often proves more valuable in real-world applications despite its theoretically higher sample requirements under specific model assumptions. We will refine the presentation in the next version of the paper.

---

> > ### Comment · Reviewer_Q1f6 · 2025-08-02
> >
> > Thanks for the authors' detailed response. I will keep my score. I wish you the best of luck with the final decision.

---

> > > ### Author Response · Authors · 2025-08-07
> > >
> > > We sincerely appreciate the reviewer's time and constructive feedback throughout the review process. Thank you for maintaining your positive evaluation of our work, and for your kind wishes.

---

### Official Review · Reviewer_eSEX · 2025-07-03

**Clarity:** 3
**Significance:** 2
**Originality:** 2
**Rating:** 3
**Confidence:** 2

**Summary:**

This paper is the first work to analyze hybrid MAB settings with heterogeneous offline-online feedback, addressing two scenarios:
- Offline preference data $\to$ online absolute rewards
- Offline absolute data $\to$ online dueling bandits

This paper proposes two algorithms HybUCB-AR and HybElimUCB-RA for Offline absolute data + online dueling bandits and Offline preference data + online stochastic bandits, respectively.

Empirical results show that both algorithms outperform state-of-the-art online methods when using heterogeneous offline data

**Questions:**

See weaknesses.

**Ethical Concerns:**

["NO or VERY MINOR ethics concerns only"]

**Final Justification:**

After reading the rebuttal from the authors, I still believe that this result is an extension of previous work[1], since handling offline-to-online conversion by taking min of UCB is not so interesting to me.

[1] Wang Chi Cheung, Lixing Lyu. Leveraging (Biased) Information: Multi-armed Bandits with Offline Data.

**Limitations:**

Yes

**Quality:**

2

**Strengths And Weaknesses:**

Strengths:
- Providing both theoretical foundations and practical performance improvements while maintaining safety guarantees.
- Although I didn't go through all the proof details, the theoretical part looks correct and well-organized.

Weaknesses:
- The results seem to be a heterogeneous version of [1]. While in practice considering heterogeneous data type is useful, the novelty in theoretical aspect is not enough.
- The algorithm follows the standard UCB-based exploration technique and incorporate the information from offline dataset by taking the minimum of two UCB rewards, which is not surprising. And the algorithms require knowledge of the bias for each data, which is not practical enough.

[1] Wang Chi Cheung, Lixing Lyu. Leveraging (Biased) Information: Multi-armed Bandits with Offline Data.

---

> ### Author Rebuttal · Authors · 2025-07-30
>
> # Response to Reviewer eSEX
>
> We thank the reviewer eSEX for the valuable comments and suggestions. Please find our detailed response below.
>
> ## 1 .Theoretical novelty
>
> We present our theoretical contributions from three distinct perspectives: (1) the analytical advancements over [1] in handling the off2online learning problem with reward bias; (2) the novelty of addressing heterogeneous feedback; and (3) the improvements over [2] in tackling the (pure online) dueling bandit problem.
>
> Although our algorithmic framework appears similar to that of [1] in handling reward bias, our theoretical analysis fundamentally differs. We point out a hidden issue in [1]: to achieve a regret bound with a "saving" term, their analysis doubles the coefficient of the standard $\log T / \Delta$ term of UCB. Specifically, traditional online UCB yields a regret of the form $C \log T / \Delta$ where $C$ is a constant term and $\Delta$ is the minimum preference gap. However, the regret upper bound for the hybrid setting in [1] is $2C \log T / \Delta - \mathrm{Saving}$. This means that when the $\mathrm{Saving}$ term is small, their regret can even exceed that of pure online UCB. Our analysis resolves this issue. Rather than first bounding $N_i$ (the number of offline samples for arm $i$) to derive a regret of the form $C \log T / \Delta^2 - \mathrm{Saving}$, and then artificially enlarging the threshold $T_i$ (number of online pulls) to $2C \log T / \Delta^2 - \mathrm{Saving}$ to leverage the properties of pure online UCB (Appendix B.2 Sub-case 1(b) in [1]), we avoid reasoning about the range of $N_i$ altogether, and instead directly analyze the required $T_i$ for eliminating suboptimal arms under hybrid UCB parameters (Appendix C.5 in our paper). This avoids unnecessary pessimism in the standard term and yields a regret of the form $C \log T / \Delta - \mathrm{Saving}$. While the asymptotic order remains the same, our analysis constitutes a fundamental advance: it retains the intrinsic performance of pure online UCB without incurring the additional overhead present in [1], demonstrating a deeper understanding of the algorithm's optimality for hybrid learning.
>
> Prior works on hybrid learning have considered absolute and relative feedback separately. However, combining the two remains a challenging and largely unexplored direction. Though work [3] considers to utilize different types of reward, their approach is to construct candidate sets independently for each feedback type. In contrast, our approach introduces a unified statistic $p_{i,j}^{\text{hyb}}$—along with a corresponding confidence radius—that simultaneously incorporates both absolute and relative feedback. This unified estimation strategy allows for more efficient use of all available data, particularly in scenarios where each individual feedback signal may be insufficiently informative on its own. We believe that this hybrid estimator is of independent interest beyond our specific setting, and can provide a useful foundation for addressing heterogeneous feedback in broader learning problems.
>
> In the reduced pure online setting with no offline data, our analysis for the dueling bandits also leads to an improved regret bound over [2]. Specifically, [2] establishes a regret upper bound of $O\left(\sum_{i \le j} \frac{\log T}{\min\set{\Delta_i, \Delta_j}} \right)$. By modifying the algorithm to select arm pairs that maximize information gain, we improve the regret bound to $O\left(\sum_{i \le j} \frac{\log T}{\max\set{\Delta_i, \Delta_j}} \right)$.
>
> [1] Cheung, W. C., & Lyu, L. (2024). Leveraging (Biased) Information: Multi-armed Bandits with Offline Data. In International Conference on Machine Learning.
>
> [2] Zoghi, M., Whiteson, S., Munos, R., & Rijke, M. (2014). Relative Upper Confidence Bound for the K-Armed Dueling Bandit Problem. In International Conference on Machine Learning.
>
> [3] Wang, X., Zeng, Q., Zuo, J., Liu, X., Hajiesmaili, M., Lui, J., & Wierman, A. (2025). Fusing Reward and Dueling Feedback in Stochastic Bandits. In International Conference on Machine Learning. In International Conference on Machine Learning.
>
>
>
> ## 2. The valid bias bound $V$
>
> As rigorously proven in Section 3 of [1] and [4], no algorithm can outperform, or even match, the performance of a purely online approach in the absence of any prior knowledge about bias. This fundamental lower bound underscores the necessity of incorporating some form of prior information when attempting to leverage offline data effectively.
>
> To the best of our knowledge, existing theoretical works that address distributional bias typically assume that an upper bound $V$ is known in advance [1], or require the bias to exceed a certain threshold to safely exclude misleading offline data [4]. Compared with [4], our algorithm makes no distributional assumptions about the offline or online data. It only requires a hyperparameter $V$, which does not need to be exact—any valid upper bound on the bias suffices. When this bound is tight, the offline data can be effectively utilized, yielding improved performance. Even under limited prior knowledge and loose bias bounds, our method still matches the performance of vanilla UCB, thus ensuring robustness.
>
> For practical use, we recommend estimating the bias through a small number of online interactions, which can provide a reasonably prior. We agree that further reducing the reliance on the parameter $V$ is an important and promising research direction, and we leave this as future work.
>
>
> [4] Qu, C., Shi, L., Panaganti, K., You, P., & Wierman, A. (2024). Hybrid Transfer Reinforcement Learning: Provable Sample Efficiency from Shifted-Dynamics Data. In Artificial Intelligence and Statistics.
>
>
>
> ## 3. UCB-based algorithm
>
> We agree that UCB is a well-established method in a broad range of online learning applications. For our considered hybrid learning problem, it is natural and intuitive to leverage offline data to update UCB-style confidence intervals while controlling for bias. The core challenge in our work does not lie in the use of UCB itself, but in designing principled UCB-style statistics that can effectively handle heterogeneous feedback and providing regret guarantees that are no weaker than their pure online counterparts. To the best of our knowledge, no prior work has addressed these challenges.

---

> > ### Author Response · Authors · 2025-08-07
> >
> > Thank you very much for your valuable feedback and thorough review of our manuscript. We were encouraged by your assessment and have addressed your questions in our rebuttal. We would be grateful to know if you have any remaining concerns that we could clarify.

---

### Decision · Program_Chairs · 2025-09-17

**Decision:**

Accept (poster)

**Comment:**

(a) Summary: This paper studies the problem of using offline data for online decision-making in the MAB problem, when the online and offline data belong to different feedback types. Specifically, it studies how offline, preference data can help with MAB with reward feedback, and how offline reward data can help with online dueling bandits. This work proposes algorithms for both settings, establishes their regret guarantees, and shows that the algorithm-specific guarantees are nearly tight. Finally, experiments show that the proposed algorithms outperform baselines.

(b) Strengths:
- The paper’s problem formulation is conceptually novel and well-motivated. Including both preference feedback and reward feedback makes the setting practically relevant.
- The proposed algorithms are technically sound and the theoretical analysis is solid.
- The paper is well-organized and mostly well-written.
- Some reviewers appreciate the technical significance and novelty of the proposed algorithms.

(c) Weaknesses:
- Practicality of the algorithms: the algorithms need to know the bias of the offline data, which is often difficult in practice. Tuning this hyperparameter can require many runs of the algorithm, making its applications less straightforward.
- Theoretical novelty: Reviewer eSEX raised the issue that the theoretical novelty is limited. The reviewer considers the work to be an extension of a previous paper that studies hybrid offline-online learning in multi-armed bandits under the homogeneous setting. The main difference here is the heterogeneous setting, but the algorithm design was found to be not sufficiently novel. The algorithms use standard UCB techniques, and incorporate the offline data by taking a minimum between the two UCB rewards (online and hybrid).
- Some reviewers (Q1f6, qfCy) found the clarity of the theoretical bounds can be improved.

(d) Reasons for decision: The primary reason for acceptance is that the paper studies a new and practically motivated setting, and gives a technically sound solution. While there were concerns about the degree of theoretical novelty and the practicality of the algorithm, the paper nevertheless makes a valuable contribution to the timely problem of leveraging heterogeneous offline data in online learning.

(e) Summary of rebuttal: During the rebuttal, the authors provided detailed responses to address the reviewers’ concerns. While 3 reviewers maintained their positive scores, one reviewer still found the work to be too incremental.
- On the bias hyperparameter: the authors explained that without any prior knowledge of offline data, there is an existing lower bound showing that one cannot improve upon vanilla UCB. They also mentioned that only an upper bound is required for their algorithms. This answer seems to address reviewer 5U6k’s concerns but not for reviewer eSEX.
- On theoretical novelty: reviewer eSEX expressed that the work is an extension of a previous work to the heterogeneous setting. The authors addressed this in detail in their rebuttal: they pointed out the differences between their analysis and prior work, and emphasized that this work was the first to introduce a "unified statistic" to handle heterogeneous feedback, distinct from prior work on homogeneous data. Reviewer eSEX was not convinced and maintained their score, but the other reviewers found the paper's originality and significance to be high.
- The authors also addressed clarity issues and the tightness of their upper / lower bounds.